# Demystifying Topological Message-Passing with Relational Structures: A Case Study on Oversquashing in Simplicial Message-Passing

[*]**Diaaeldin Taha**[1], [*]**James Chapman**[2], [†]**Marzieh Eidi**[1,3], [†]**Karel Devriendt**[4], **Guido Montúfar**[1,2]

[1]Max Planck Institute for Mathematics in the Sciences, Leipzig, Germany
[2]UCLA, CA, USA
[3]Center for Scalable Data Analytics and Artificial Intelligence, Leipzig, Germany
[4]University of Oxford, Oxford, UK

`taha@mis.mpg.de`, `chapman20j@math.ucla.edu`, `meidi@mis.mpg.de`,
`karel.devriendt@maths.ox.ac.uk`, `montufar@math.ucla.edu`

## Abstract

Topological deep learning (TDL) has emerged as a powerful tool for modeling higher-order interactions in relational data. However, phenomena such as oversquashing in topological message-passing remain understudied and lack theoretical analysis. We propose a unifying axiomatic framework that bridges graph and topological message-passing by viewing simplicial and cellular complexes and their message-passing schemes through the lens of relational structures. This approach extends graph-theoretic results and algorithms to higher-order structures, facilitating the analysis and mitigation of oversquashing in topological message-passing networks. Through theoretical analysis and empirical studies on simplicial networks, we demonstrate the potential of this framework to advance TDL.

## 1 Introduction

Recent years have witnessed a growing recognition that traditional machine learning, rooted in Euclidean spaces, often fails to capture the complex structure and relationships present in real-world data. This shortcoming has driven the development of geometric deep learning (GDL) (Bronstein et al., 2021) and, more recently, topological deep learning (TDL) (Hajij et al., 2023), for handling non-Euclidean and relational data. TDL, in particular, has emerged as a promising frontier for relational learning, that extends beyond graph neural networks (GNNs). TDL offers tools to capture and analyze higher-order interactions and topological features in complex data and higher-order structures, such as simplicial complexes, cell complexes, and sheaves (Hajij et al., 2023). However, the TDL field is young, and the TDL community has yet many open theoretical and practical questions relating to, e.g., oversquashing and rewiring (research directions 2 and 9 of Papamarkou et al., 2024).

Oversquashing is a challenging failure mode in GNNs, where information struggles to propagate across long paths due to the compression of an exponentially growing number of messages into fixed-size vectors (Alon & Yahav, 2021). This phenomenon has been examined through various perspectives, including curvature (Topping et al., 2022), graph expansion (Banerjee et al., 2022), effective resistance (Black et al., 2023), and spectral properties (Karhadkar et al., 2023). Despite the potential of higher-order message passing architectures—such as simplicial neural networks (Ebli et al., 2020), message passing simplicial networks (Bodnar et al., 2021b), and CW networks (Bodnar et al., 2021a)—there remains a lack of unified frameworks for analyzing and mitigating oversquashing in these settings.

In this paper, we take a first step toward studying oversquashing in TDL by showing that simplicial complexes and their message passing schemes can be interpreted as relational structures, making it

---

[*]Equal contribution.
[†]Equal contribution.

possible to extend key GNN insights and tools to higher-order message passing architectures. The conceptual framework and theoretical results developed in this paper address pressing questions in the TDL community (e.g., research directions 2 and 9 of Papamarkou et al., 2024).

**Contributions.** Our contributions are threefold:

- **Axiomatic**: We provide a unifying view of simplicial complexes and their message passing schemes through the lens of relational structures.
- **Theoretical**: We introduce *influence graphs* which enable novel extensions of prior graph analyses to higher-order structures, where existing methods for analysis do not apply. We extend graph-theoretic concepts and results on oversquashing to relational structures, analyzing network sensitivity (Lemma 3.2), local geometry (Proposition 3.4), and the impact of network depth (Theorem 3.5) and hidden dimensions (Section 3.4).
- **Practical**: We propose a heuristic to extend oversquashing-mitigation techniques from graph-based models to relational structures.

**Related Work.** Our work sits at the intersection of graph neural networks, topological deep learning, relational learning, and the study of oversquashing and graph rewiring in graph neural networks. We review related work in Appendix A.

The rest of this paper is organized as follows: Section 2 provides the axiomatic groundwork for relating simplicial and relational message passing. Section 3 presents our theoretical analysis of oversquashing in this context. Section 4 introduces a heuristic rewiring strategy to mitigate oversquashing in relational message passing. Section 5 presents our experimental results, followed by a discussion and conclusions in Section 6.

## 2 SIMPLICIAL COMPLEXES ARE RELATIONAL STRUCTURES

In this section, we reinterpret simplicial complexes and message passing through the lens of *relational structures*. We begin by recalling simplicial complexes and a representative simplicial message passing scheme, then reframe these notions within a relational framework. We illustrate the definitions in this section with a small worked example in Appendx H. We note that the connection we establish here extends to other higher-order structures, such as cellular complexes (Hansen & Ghrist, 2019; Bodnar et al., 2021a; Giusti et al., 2024).

### 2.1 SIMPLICIAL COMPLEXES AND MESSAGE PASSING

Simplicial complexes are mathematical structures that generalize graphs to higher dimensions, capturing relationships among vertices, edges, triangles, and higher-dimensional objects.

**Definition 2.1** (Simplicial Complex, Nanda, 2021). *Let $V$ be a non-empty set. A simplicial complex $\mathcal{K}$ is a collection of non-empty subsets of $V$ that contains all the singleton subsets of $V$ and is closed under the operation of taking non-empty subsets.*

A member $\sigma = \{v_0, v_1, \ldots, v_d\} \in \mathcal{K}$ with cardinality $|\sigma| = d + 1$ is called a *$d$-simplex*. Geometrically, 0-simplices are vertices, 1-simplices are edges, 2-simplices are triangles, and so on.

**Definition 2.2** (Boundary Incidence Relation). *We say that $\tau$ covers $\sigma$, written $\sigma \prec \tau$, iff $\sigma \subsetneq \tau$ and there is no $\delta \in \mathcal{K}$ such that $\sigma \subsetneq \delta \subsetneq \tau$.*

The incidence relations from Definition 2.2 can be used to construct four types of (local) adjacencies.

**Definition 2.3.** *Consider a simplex $\sigma \in \mathcal{K}$. Four types of adjacent simplices can be defined:*

1. *Boundary adjacency: $\mathcal{B}(\sigma) = \{\tau : \tau \prec \sigma\}$;*
2. *Co-boundary adjacency: $\mathcal{C}(\sigma) = \{\tau : \sigma \prec \tau\}$;*
3. *Lower adjacency: $\mathcal{N}_\downarrow(\sigma) = \{\tau : \exists \delta \text{ such that } \delta \prec \tau \text{ and } \delta \prec \sigma\}$;*
4. *Upper adjacency: $\mathcal{N}_\uparrow(\sigma) = \{\tau : \exists \delta \text{ such that } \tau \prec \delta \text{ and } \sigma \prec \delta\}$.*

In Figure 1, we illustrate an example of a simplicial complex and its adjacency relations.

We now, following Bodnar et al. (2021b, Section 4), review a general scheme for message passing on simplicial complexes. In Appendix A, we provide references for topological message passing architectures that fit this scheme. We refer readers to Appendix F.5 for specific instantiations of this scheme in our graph and topological message passing models.

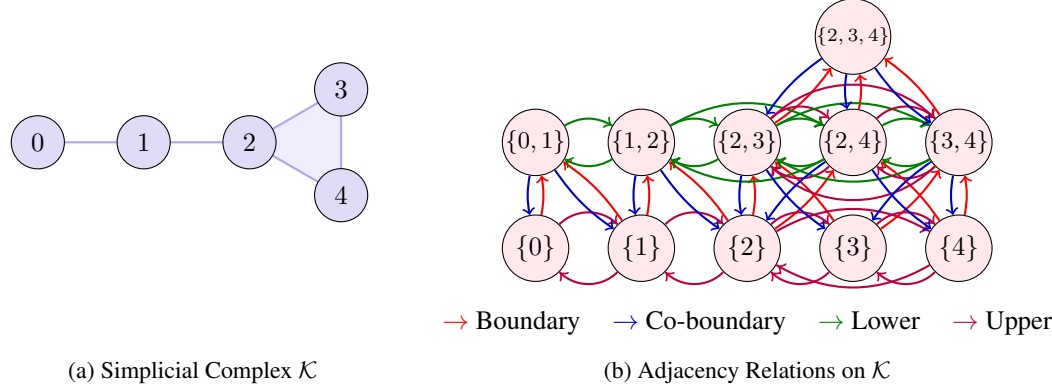

(a) Simplicial Complex $\mathcal{K}$         (b) Adjacency Relations on $\mathcal{K}$

Figure 1: The left panel shows a simplicial complex $\mathcal{K}$ consisting of five vertices, five edges, and one triangle. The right panel shows the corresponding adjacency relations depicted as arrows to each simplex $\sigma \in \mathcal{K}$ emanating from each of its adjacent simplices in $\mathcal{B}(\sigma), \mathcal{C}(\sigma), \mathcal{N}_\downarrow(\sigma), \mathcal{N}_\uparrow(\sigma)$.

Simplicial message passing extends graph message passing from pairwise node connections to higher-dimensional adjacency connections between simplices. Message passing schemes on simplicial complexes iteratively update feature vectors assigned to simplices by exchanging messages between adjacent simplices. For a simplicial complex $\mathcal{K}$, we denote the feature vector of a simplex $\sigma \in \mathcal{K}$ as $\mathbf{h}_\sigma \in \mathbb{R}^p$. At each iteration (layer) $t$, the feature vectors $\mathbf{h}_\sigma^{(t)}$ of simplices $\sigma \in \mathcal{K}$ are updated by aggregating messages from adjacent simplices. For a simplex $\sigma \in \mathcal{K}$, the messages passed from adjacent simplices are defined as follows:

$$
\begin{aligned}
\mathbf{m}_\mathcal{B}^{(t+1)}(\sigma) &= \mathrm{AGG}_{\tau \in \mathcal{B}(\sigma)} \left( M_\mathcal{B}(\mathbf{h}_\sigma^{(t)}, \mathbf{h}_\tau^{(t)}) \right), \\
\mathbf{m}_\mathcal{C}^{(t+1)}(\sigma) &= \mathrm{AGG}_{\tau \in \mathcal{C}(\sigma)} \left( M_\mathcal{C}(\mathbf{h}_\sigma^{(t)}, \mathbf{h}_\tau^{(t)}) \right), \\
\mathbf{m}_\downarrow^{(t+1)}(\sigma) &= \mathrm{AGG}_{\tau \in \mathcal{N}_\downarrow(\sigma)} \left( M_\downarrow(\mathbf{h}_\sigma^{(t)}, \mathbf{h}_\tau^{(t)}, \mathbf{h}_{\sigma \cap \tau}^{(t)}) \right), \\
\mathbf{m}_\uparrow^{(t+1)}(\sigma) &= \mathrm{AGG}_{\tau \in \mathcal{N}_\uparrow(\sigma)} \left( M_\uparrow(\mathbf{h}_\sigma^{(t)}, \mathbf{h}_\tau^{(t)}, \mathbf{h}_{\sigma \cup \tau}^{(t)}) \right).
\end{aligned}
\tag{1}
$$

Here AGG is an aggregation function (e.g., sum or mean), and $M_\mathcal{B}, M_\mathcal{C}, M_\downarrow, M_\uparrow$ are message functions (e.g., linear or MLP). Then, an update operation UPDATE (e.g., MLP) incorporates these four different types of incoming messages:

$$
\mathbf{h}_\sigma^{(t+1)} = \mathrm{UPDATE} \left( \mathbf{h}_\sigma^{(t)}, \mathbf{m}_\mathcal{B}^{(t+1)}(\sigma), \mathbf{m}_\mathcal{C}^{(t+1)}(\sigma), \mathbf{m}_\downarrow^{(t+1)}(\sigma), \mathbf{m}_\uparrow^{(t+1)}(\sigma) \right).
\tag{2}
$$

Finally, after the last iteration, a read-out function is applied to process the features to perform a desired task, such as classification or regression.

## 2.2 RELATIONAL STRUCTURES AND MESSAGE PASSING

We model simplicial complexes and the above message passing scheme using *relational structures*.

**Definition 2.4** (Relational Structure, Hodges, 1993). *A relational structure* $\mathcal{R} = (\mathcal{S}, R_1, \ldots, R_k)$ *consists of a finite set $\mathcal{S}$ of* entities, *and relations* $R_i \subseteq \mathcal{S}^{n_i}$, *where $n_i$ is the arity of $R_i$.*

We note that modeling simplicial complexes as relational structures generalizes a powerful perspective in which simplicial complexes and similar constructs are treated as augmented Hasse diagrams as demonstrated, for example, by Hajij et al. (2023), Eitan et al. (2024), and Papillon et al. (2024).

We now introduce a general scheme for message passing on relational structures which encompasses the simplicial message passing scheme from Section 2.1. This scheme is an extension of the relational graph convolution model from Schlichtkrull et al. (2018) which allows for relations of different arities, not just binary relations.

**Definition 2.5** (Relational Message Passing Model). *A relational message passing model on a relational structure* $\mathcal{R} = (\mathcal{S}, R_1, \ldots, R_k)$ *consists of:*

- *Feature vectors:* $\mathbf{h}_\sigma^{(t)} \in \mathbb{R}^{p_t}$ *for each $\sigma \in \mathcal{S}$ at layer $t \geq 0$, initialized as $\mathbf{h}_\sigma^{(0)} = \mathbf{x}_\sigma$ (input features). Here, $p_t$ denotes the dimensionality of the feature vectors at layer $t$.*

- Message functions $\psi_i^{(t)} \colon \mathbb{R}^{p_t} \times \cdots \times \mathbb{R}^{p_t} \to \mathbb{R}^{p_{i,t}}$ *(with $n_i$ arguments) for each relation $R_i$, where $i = 1, \ldots, k$. Each message function takes $n_i$ input feature vectors (corresponding to the target simplex and its $n_i - 1$ related simplices) and outputs a message vector of dimension $p_{i,t}$. The parameter $n_i$ represents the arity of the relation $R_i$.*
- Update function $\phi^{(t)} \colon \mathbb{R}^{p_{1,t}} \times \cdots \times \mathbb{R}^{p_{k,t}} \to \mathbb{R}^{p_{t+1}}$. *The output dimension $p_{t+1}$ specifies the dimensionality of the feature vectors at layer $t + 1$.*
- Shift operators $\mathbf{A}^{R_i} \in \mathbb{R}_{\geq 0}^{|\mathcal{S}|^{n_i}}$ *associated with each relation $R_i$, for $i = 1, \ldots, k$. For each $\sigma \in \mathcal{S}$ and $\boldsymbol{\xi} = (\xi_1, \ldots, \xi_{n_i-1}) \in \mathcal{S}^{n_i-1}$ with $(\sigma, \boldsymbol{\xi}) \in R_i$, the element $\mathbf{A}_{\sigma,\boldsymbol{\xi}}^{R_i}$ represents the strength of the signal passed from $\boldsymbol{\xi}$ to $\sigma$. More specifically, for any combination of entities $(\zeta_1, \zeta_2, \ldots, \zeta_{n_i}) \in \mathcal{S}^{n_i}$ where the relation $R_i$ does not hold among the entities $\zeta_1, \zeta_2, \ldots, \zeta_{n_i}$ (i.e., $(\zeta_1, \zeta_2, \ldots, \zeta_{n_i}) \notin R_i$), the tensor $\mathbf{A}^{R_i}$ satisfies $\mathbf{A}_{\zeta_1, \zeta_2, \ldots, \zeta_{n_i}}^{R_i} = 0$.*

*The update rule is given by:*

$$\mathbf{h}_\sigma^{(t+1)} = \phi^{(t)} \left( \mathbf{m}_{\sigma,1}^{(t)}, \ldots, \mathbf{m}_{\sigma,k}^{(t)} \right), \tag{3}$$

*where for each $i = 1, \ldots, k$, the message $\mathbf{m}_{\sigma,i}^{(t)}$ received by $\sigma$ over $R_i$ is computed as:*

$$\mathbf{m}_{\sigma,i}^{(t)} = \sum_{\boldsymbol{\xi} \in \mathcal{S}^{n_i-1}} \mathbf{A}_{\sigma,\boldsymbol{\xi}}^{R_i} \, \psi_i^{(t)} \left( \mathbf{h}_\sigma^{(t)}, \mathbf{h}_{\xi_1}^{(t)}, \ldots, \mathbf{h}_{\xi_{n_i-1}}^{(t)} \right). \tag{4}$$

**Remark 2.6.** *The shift operators in Definition 2.5 extend the definition of graph shift operators (Mateos et al., 2019; Gama et al., 2020; Dasoulas et al., 2021) from graphs to relational structures. Whereas relations indicate whether entities are connected, shift operators numerically encode these connections with weights.*

In the context of message passing, a simplicial complex $\mathcal{K}$ can be viewed as a relational structure $\mathcal{R}(\mathcal{K}) = (\mathcal{S}, R_1, \ldots, R_5)$, where $\mathcal{S} = \mathcal{K}$ are the entities, and $R_i$ are relations defined as follows: $R_1 = \{(\sigma) : \sigma \in \mathcal{K}\}$ (identity), $R_2 = \{(\sigma, \tau) : \sigma \in \mathcal{K}, \tau \in \mathcal{B}(\sigma)\}$ (boundary), $R_3 = \{(\sigma, \tau) : \sigma \in \mathcal{K}\tau \in \mathcal{C}(\sigma)\}$ (co-boundary), $R_4 = \{(\sigma, \tau, \delta) : \sigma \in \mathcal{K}, \tau \in \mathcal{N}_\downarrow(\sigma), \delta = \sigma \cap \tau\}$ (lower adjacency), $R_5 = \{(\sigma, \tau, \delta) : \sigma \in \mathcal{K}, \tau \in \mathcal{N}_\uparrow(\sigma), \delta = \sigma \cup \tau\}$ (upper adjacency). The message functions $\psi_i^{(t)}$ correspond to $M_\mathcal{B}, M_\mathcal{C}, M_\downarrow, M_\uparrow$, the update function $\phi^{(t)}$ to UPDATE, and aggregation uses shift operators $\mathbf{A}^{R_i}$. This establishes an equivalence between message passing on the simplicial complex $\mathcal{K}$ and the relational structure $\mathcal{R}(\mathcal{K})$.

**Remark 2.7.** *The relational message passing scheme in Definition 2.5 encompasses relational graph neural networks (Schlichtkrull et al., 2018), simplicial neural networks (Bodnar et al., 2021b), higher-order graph neural networks (Morris et al., 2019), and cellular complex neural networks (Bodnar et al., 2021a). We demonstrate how higher-order graphs fit the relational framework in Appendix G.*

---

**Takeaway Message 1 (Axiomatic)**

Simplicial complexes can be represented as *relational structures*, where the entities are simplices, and the relations capture the adjacency among simplices of different dimensions. Simplicial message passing is an instance of relational message passing on these structures.

---

## 3 OVERSQUASHING IN RELATIONAL MESSAGE-PASSING

The existing literature on oversquashing in GNNs does not directly address relational message passing. In this section, we address that gap by deriving new extensions of key results on oversquashing in GNNs to relational message passing. We illustrate the definitions in this section with a small worked example in Appendx H.

In our analysis of relational structures and message passing schemes, we naturally encounter matrices and graphs that capture the aggregated influence of the underlying shift operators. For convenience, we introduce notation for these matrices and graphs. For each relation $R_i$ of arity $n_i$ with

shift operator $\mathbf{A}^{R_i}$, we define the matrix $\tilde{\mathbf{A}}^{R_i} \in \mathbb{R}_{\geq 0}^{|\mathcal{S}| \times |\mathcal{S}|}$ as:

$$\tilde{\mathbf{A}}_{\sigma,\tau}^{R_i} = \sum_{j=1}^{n_i-1} \sum_{\boldsymbol{\xi} \in \mathcal{S}^{n_i-2}} \mathbf{A}_{\sigma,\xi_1,\ldots,\xi_{j-1},\tau,\xi_j,\ldots,\xi_{n_i-2}}^{R_i}, \quad \sigma, \tau \in \mathcal{S}. \tag{5}$$

This matrix captures all possible ways an entity $\tau$ can influence entity $\sigma$ via the relation $R_i$. Specifically, it sums over all positions $j$ where $\tau$ can appear among the arguments of the shift operator $\mathbf{A}^{R_i}$, and over all possible combinations of the other entities $\boldsymbol{\xi}$.

We aggregate these matrices over all relations to form the *aggregated influence matrix* $\tilde{\mathbf{A}} \in \mathbb{R}_{\geq 0}^{|\mathcal{S}| \times |\mathcal{S}|}$:

$$\tilde{\mathbf{A}} = \sum_{i=1}^{k} \tilde{\mathbf{A}}^{R_i}. \tag{6}$$

Next, we define the *augmented influence matrix* $\mathbf{B}$, which plays the role of an augmented adjacency matrix in our analysis:

$$\mathbf{B} = \gamma \mathbf{I} + \tilde{\mathbf{A}}, \tag{7}$$

where $\gamma = \max_\sigma \sum_{\boldsymbol{\xi} \in \mathcal{S}^{n_i-1}} \tilde{\mathbf{A}}_{\sigma,\boldsymbol{\xi}}$ is the maximum row sum of $\tilde{\mathbf{A}}$.

Lastly, we introduce graphs that capture the aggregated message passing dynamical structure implied by the relational structure and the message passing scheme.

**Definition 3.1** (Influence Graph). *Given a relational structure $\mathcal{R} = (\mathcal{S}, R_1, \ldots, R_k)$ and a relational message passing scheme with update rule given by Equation 3, and given $\mathbf{Q} \in \{\tilde{\mathbf{A}}, \mathbf{B}\}$, where $\tilde{\mathbf{A}}$ and $\mathbf{B}$ are defined by Equations 6 and 7 respectively, we define the* influence graph *$\mathcal{G}(\mathcal{S}, \mathbf{Q}) = (\mathcal{S}, \mathcal{E}, w)$ as follows: The set of entities (i.e., nodes) is $\mathcal{S}$. The set of edges $\mathcal{E}$ consists of directed edges from entity $\tau$ to entity $\sigma$ for each pair $(\sigma, \tau) \in \mathcal{S} \times \mathcal{S}$ with $\mathbf{Q}_{\sigma,\tau} > 0$. Each edge from $\tau$ to $\sigma$ is assigned a weight $w_{\tau \to \sigma} = \mathbf{Q}_{\sigma,\tau}$.*

As we will see next, these graphs make it possible to leverage and extend graph-theoretic concepts, results, and intuition to understand and analyze the behavior of relational message passing schemes.

## 3.1 SENSITIVITY ANALYSIS

We now analyze the sensitivity of relational message passing to changes in the input features. This analysis is crucial for understanding how information propagates through the network and for identifying potential bottlenecks or oversquashing effects. We begin with a standard assumption about the boundedness of the Jacobians of the message and update functions.

**Assumption 1** (Bounded Jacobians). *All message functions $\psi_i^{(\ell)}$ and update functions $\phi^{(\ell)}$ are differentiable with bounded Jacobians: There exist constants $\beta_i^{(\ell)}$ and $\alpha^{(\ell)}$ such that $\|\partial \psi_i^{(\ell)} / \partial \mathbf{h}_\sigma\|_1 \leq \beta_i^{(\ell)}$ for any input feature vector $\mathbf{h}_\sigma$, and $\|\partial \phi^{(\ell)} / \partial \mathbf{m}_j\|_1 \leq \alpha^{(\ell)}$ for any message input $\mathbf{m}_j$. We write $\beta^{(\ell)} = \max_i \beta_i^{(\ell)}$.*

Our main result on sensitivity is the following, which is a novel extension of GNN sensitivity analysis results (e.g., Topping et al., 2022, Lemma 1 and Di Giovanni et al., 2023, Theorem 3.2) to relational (and topological) message passing. We provide the proof in Appendix C.1.

**Lemma 3.2** (Sensitivity Bound for Relational Message Passing). *Consider a relational structure $\mathcal{R} = (\mathcal{S}, R_1, \ldots, R_k)$ with update rule given by Equation 3 and satisfying Assumption 1. Then, for any $\sigma, \tau \in \mathcal{S}$ and $t > 0$, the Jacobian at layer $t$ with respect to the input features $(t = 0)$ satisfies*

$$\left\| \frac{\partial \mathbf{h}_\sigma^{(t)}}{\partial \mathbf{h}_\tau^{(0)}} \right\|_1 \leq \left( \prod_{\ell=0}^{t-1} \alpha^{(\ell)} \beta^{(\ell)} \right) \left( \mathbf{B}^t \right)_{\sigma,\tau}. \tag{8}$$

Thus, the bound on the Jacobian of the $\sigma$-feature with respect to the input $\tau$-feature depends on the $(\sigma, \tau)$-entry of the $t$-th matrix power $\mathbf{B}^t$, which reflects the number and strength of $t$-length paths from $\tau$ to $\sigma$ in the graph $\mathcal{G}(\mathcal{S}, \mathbf{B})$. Structural properties of $\mathcal{G}(\mathcal{S}, \mathbf{B})$ that lead to small values of $(\mathbf{B}^t)_{\sigma,\tau}$, such as bottlenecks or long distances between nodes, therefore contribute to the phenomenon of oversquashing, where the influence of distant entities is diminished.

As demonstrated throughout this work, our result offers a systematic framework for extending other theoretical findings on oversquashing in graphs, which do not directly apply to simplicial complexes and similar relational structures. This includes the influential works by Topping et al. (2022), Di Giovanni et al. (2023), and Fesser & Weber (2023). By leveraging our axiomatic framework, we derive principled extensions on the impact of local geometry (Section 3.2), depth (Section 3.3), and hidden dimensions (Section 3.4) in higher-order message passing, addressing settings where prior results are not applicable. Additionally, this result offers a clear approach for deriving analogs of key quantities such as curvature (Definition 3.3), and can serve as a guide for future work.

## 3.2 THE IMPACT OF LOCAL GEOMETRY

Lemma 3.2 shows that the entries of the matrix $\mathbf{B}^t$, which encode the number and strength of connections in a relational message passing scheme, control feature sensitivity. Prior works relate similar bounds to notions of discrete curvature for unweighted undirected graphs, such as balanced Forman curvature (Topping et al., 2022), Ollivier-Ricci curvature (Nguyen et al., 2023), and augmented Forman curvature (Fesser & Weber, 2023), via counting local motifs, such as triangles and squares, in the underlying graphs. Following this approach, we derive a result analogous to Fesser & Weber (2023, Proposition 3.4), introducing a motif-counting quantity inspired by the augmented Forman curvature, but adapted for the particular weighted directed graphs arising in our setting.

**Definition 3.3.** *Let $\mathcal{G} = (\mathcal{S}, \mathcal{E}, w)$ be a weighted directed graph with entities (nodes) $\mathcal{S}$, edges $\mathcal{E}$, and edge weights $w \colon \mathcal{E} \to \mathbb{R}_{\geq 0}$. For each entity $\tau \in \mathcal{S}$, define the* weighted out-degree $w_\tau^{\text{out}} = \sum_{(\tau \to \sigma) \in \mathcal{E}} w_{\tau \to \sigma}$ *and the* weighted in-degree $w_\tau^{\text{in}} = \sum_{(\sigma \to \tau) \in \mathcal{E}} w_{\sigma \to \tau}$. *For an edge $(\tau \to \sigma) \in \mathcal{E}$, define the* weighted triangle count $w_T = \sum_{\xi \in \mathcal{S}} w_{\tau \to \xi} \cdot w_{\xi \to \sigma}$ *and the* weighted quadrangle count $w_F = \sum_{\xi_1, \xi_2 \in \mathcal{S}} w_{\tau \to \xi_1} \cdot w_{\xi_1 \to \xi_2} \cdot w_{\xi_2 \to \sigma}$. *Then, the* extended Forman curvature *of the edge $(\tau \to \sigma)$ is defined as:*

$$\text{EFC}_{\mathcal{G}}(\tau, \sigma) = 4 - w_\tau^{\text{out}} - w_\sigma^{\text{in}} + 3w_T + 2w_F. \tag{9}$$

We immediately get the following result, inspired by Nguyen et al. (2023, Proposition 4.4) and Fesser & Weber (2023, Proposition 3.4), and which is exemplary of results connecting sensitivity analysis to notions of discrete curvature. We provide the proof in Appendix C.2.

**Proposition 3.4.** *Consider a relational structure $\mathcal{R} = (\mathcal{S}, R_1, \dots, R_k)$ with update rule given by Equation 3 and satisfying Assumption 1. Denote $\mathcal{G} = \mathcal{G}(\mathcal{S}, \mathbf{B})$. Then, for any $\sigma, \tau \in \mathcal{S}$ with an edge $(\tau \to \sigma) \in \mathcal{G}$, the following holds:*

$$\left\| \frac{\partial \mathbf{h}_\sigma^{(2)}}{\partial \mathbf{h}_\tau^{(0)}} \right\|_1 \leq \frac{1}{3} \left( \prod_{\ell=0}^{1} \alpha^{(\ell)} \beta^{(\ell)} \right) \left[ \text{EFC}_{\mathcal{G}}(\tau, \sigma) + w_\tau^{\text{out}} + w_\sigma^{\text{in}} - 4 \right]. \tag{10}$$

In principle, a similar result using balanced Forman curvature (as in Topping et al., 2022, Theorem 4) is possible using our framework, and we leave that extension for future work. Connections to Ollivier-Ricci curvature are discussed in Appendix B.1.

We present experimental analyses related to curvature on relational structures in Section D.3 (edge curvature distribution) and Appendices D.1 (edge curvature visualization) and D.2 (weighted curvature). We further propose a relational extension of curvature-based rewiring techniques in Section 4 and empirically analyze the impact of relational rewiring using real-world and synthetic benchmarks in Sections 5.1 and 5.2, respectively.

## 3.3 THE IMPACT OF DEPTH

To facilitate our analysis of depth, we make the following non-restrictive assumptions:

**Assumption 2** (Row-Normalized Shift Operators)**.** *Each shift operator $\mathbf{A}^{R_i}$ associated with relation $R_i$ is row-normalized, such that for all $\sigma \in \mathcal{S}$,*

$$\sum_{\boldsymbol{\xi} \in \mathcal{S}^{n_i-1}} A_{\sigma, \boldsymbol{\xi}}^{R_i} = \begin{cases} 1, & \text{if } \sum_{\boldsymbol{\xi}} A_{\sigma, \boldsymbol{\xi}}^{R_i} \neq 0, \\ 0, & \text{if } \sum_{\boldsymbol{\xi}} A_{\sigma, \boldsymbol{\xi}}^{R_i} = 0. \end{cases} \tag{11}$$

**Assumption 3** (Bounded $\alpha^{(\ell)}$ and $\beta^{(\ell)}$)**.** *There exist constants $\alpha_{\max} > 0$ and $\beta_{\max} > 0$ such that for all layers $\ell$, $\alpha^{(\ell)} \leq \alpha_{\max}$ and $\beta^{(\ell)} \leq \beta_{\max}$.*

We now present our main result on the impact of depth in relational message passing, extending a previous result by Di Giovanni et al. (2023, Theorem 4.1) to our setting. We provide the proof in Appendix C.3. By *the combinatorial distance* from $\tau$ to $\sigma$ in the graph $\mathcal{G}(\mathcal{S}, \tilde{\mathbf{A}})$, we mean the smallest number of edges in a directed path from $\tau$ to $\sigma$ in the graph. Similarly, by *combinatorial length* of a directed path, we mean the number of edges in the path.

**Theorem 3.5** (Impact of Depth on Relational Message Passing). *Consider a relational structure $\mathcal{R} = (\mathcal{S}, R_1, \ldots, R_k)$ with update rule given by Equation 3 and satisfying Assumptions 1, 2, and 3. Let $\sigma, \tau \in \mathcal{S}$ be entities such that the combinatorial distance from $\tau$ to $\sigma$ in the graph $\mathcal{G}(\mathcal{S}, \tilde{\mathbf{A}})$ is $r$. Denote by $\omega_\ell(\sigma, \tau)$ the number of directed paths from $\tau$ to $\sigma$ of combinatorial length at most $\ell$ in $\mathcal{G}(\mathcal{S}, \tilde{\mathbf{A}})$. Then, for any $0 \le m < r$, there exists a constant $C > 0$, depending only on $\alpha_{\max}$, $\beta_{\max}$, $k$, and $m$, but not on $r$ nor the specific relations in $\mathcal{R}$, such that*

$$\left\| \frac{\partial \mathbf{h}_\sigma^{(r+m)}}{\partial \mathbf{h}_\tau^{(0)}} \right\|_1 \le C \omega_{r+m}(\sigma, \tau)(2\alpha_{\max}\beta_{\max}M)^r, \tag{12}$$

*where $M = \max_{\sigma,\tau} \tilde{\mathbf{A}}_{\sigma,\tau}$.*

This result indicates that the sensitivity can decay exponentially with depth when $M < 1/(2\alpha_{\max}\beta_{\max})$, particularly when the number of walks $\omega_t(\sigma, \tau)$ is limited by the structure of $\mathcal{G}(\mathcal{S}, \tilde{\mathbf{A}})$. Such exponential decay is a characteristic of the oversquashing phenomenon, where information from distant nodes becomes increasingly compressed, reducing its influence on the output.

We present experimental validation of this result in Section 5.2.

### 3.4 THE IMPACT OF HIDDEN DIMENSIONS

In situations where the Lipschitz constants of the message and update functions from Assumption 1 are affected by hyperparameters, such as the widths of neural networks implementing said functions, one can have $\beta_i^{(\ell)} = O(p_{i,\ell})$ and $\alpha^{(\ell)} = O(p_{\ell+1})$. This is the case, for instance, when the message and update functions are shallow neural networks (see Appendix B.2 and Appendix C.4).

Writing $p_\ell' = \max_i p_{i,\ell}$ and substituting $\beta^{(\ell)} = O(p_\ell')$ and $\alpha^{(\ell)} = O(p_{\ell+1})$ into the bound from Lemma 3.2, one gets:

$$\left\| \frac{\partial \mathbf{h}_\sigma^{(t)}}{\partial \mathbf{h}_\tau^{(0)}} \right\|_1 \le C \cdot \left( \prod_{\ell=0}^{t-1} p_{\ell+1} \cdot p_\ell' \right) \left( \mathbf{B}^t \right)_{\sigma,\tau}, \tag{13}$$

where $C$ is a constant independent of the layer widths, $p_\ell'$ is the maximum dimension of the message vectors at layer $\ell$, and $p_{\ell+1}$ is the output dimension of the update function at layer $\ell$.

This implies that low hidden dimensions in the message and update functions contribute to a low sensitivity upper bound, which can exacerbate the oversquashing problem. Increasing the hidden dimensions will raise the upper bound, which can help improve the model's ability to propagate information effectively and enhance performance on tasks. However, increasing the hidden dimensions risks overfitting due to the increased model complexity (Bartlett et al., 2017).

We present experimental validation of this result in Section 5.2.

> **Takeaway Message 2 (Theoretical)**
>
> By reformulating higher order structures as relational structures, key results on oversquashing in graph neural networks extend to relational message passing schemes through the *aggregated influence matrix* and the *influence graph*. This conceptual framework enables analysis of the impact of local geometry, depth, and hidden dimensions in relational message passing schemes, just as in graph neural networks.

## 4 REWIRING HEURISTICS FOR RELATIONAL STRUCTURES

Inspired by First-Order Spectral Rewiring (FoSR) (Karhadkar et al., 2023), we propose a rewiring heuristic that integrates additional connections into a relational structure without altering its original

connections. To capture the overall connectivity of a relational structure, we define the collapsed adjacency matrix, which counts the number of direct connections between entities.

**Definition 4.1** (Collapsed Adjacency Matrix). *Given a relational structure $\mathcal{R} = (\mathcal{S}, R_1, \ldots, R_k)$, the* collapsed adjacency matrix $\mathbf{A}^{\text{col}}$ *for the structure $\mathcal{R}$ is defined by:*

$$\mathbf{A}^{\text{col}}_{\sigma,\tau} = \sum_{i=1}^{k} \sum_{\boldsymbol{\xi} \in \mathcal{S}^{n_i-1}} \mathbf{1}_{\{(\sigma,\boldsymbol{\xi}) \in R_i, \tau = \xi_j \text{ for some } j \in \{1,\ldots,n_i\}\}}, \quad \sigma, \tau \in \mathcal{S}. \tag{14}$$

This matrix captures direct connections between entities through any relation, effectively collapsing the relational structure into a graph. Our proposed relational rewiring algorithm is as follows.

---

**Algorithm 1** Relational Rewiring Algorithm

---

**Require:** Relational structure $\mathcal{R} = (\mathcal{S}, R_1, \ldots, R_k)$; graph rewiring algorithm REWIREALGO
1: Construct the collapsed adjacency matrix $\mathbf{A}^{\text{col}}$ (Definition 4.1)
2: Build the graph $\mathcal{G}^{\text{col}} = (\mathcal{S}, \mathbf{A}^{\text{col}})$
3: Apply REWIREALGO to $\mathcal{G}^{\text{col}}$ to obtain additional edges $E_{\text{new}}$
4: Define a new relation $R_{k+1} = E_{\text{new}}$
5: Update the relational structure: $\mathcal{R}' = (\mathcal{S}, R_1, \ldots, R_k, R_{k+1})$

---

Adding new connections ($E_{\text{new}}$) without removing existing ones improves the model capacity to capture long-range dependencies while preserving the original relational structure. We experimentally analyze the impact of relational rewiring using real-world and synthetic benchmarks in Sections 5.1 and 5.2, respectively. Future work could explore rewiring algorithms that remove or reclassify edges, such as spectral pruning (Jamadandi et al., 2024), either by re-labeling the edges as new relations or by deletion. We report preliminary empirical results on pruning in Appendix D.

> **Takeaway Message 3 (Practical)**
>
> Graph rewiring techniques for improving information flow and mitigating oversquashing can be adapted to relational structures. This approach improves long-range connectivity and enhances message propagation while maintaining the integrity of the original connections.

## 5 EXPERIMENTS AND RESULTS

### 5.1 REAL-WORLD BENCHMARK: GRAPH CLASSIFICATION

We run an empirical analysis with real-world datasets to compare the performance of different graph, relational graph, and simplicial message passing models, and the impact of relational rewiring on said models. We provide more details in Appendix E.

**Task and Datasets.** We use graph classification tasks ENZYMES, IMDB-B, MUTAG, NCI1, and PROTEINS from the TUDataset (Morris et al., 2020) for evaluation.

**Graph Lifting.** For the topological message passing models, graphs are separately treated as complexes with only graph nodes (0-dimensional simplices) and upper adjacencies, and also lifted into clique and ring complexes (see appendix).

**Models.** We evaluate three types of models: a) *Graph message passing models*: SGC (Wu et al., 2019), GCN (Kipf & Welling, 2017), and GIN (Xu et al., 2019); b) *Relational graph message passing models*: RGCN (Schlichtkrull et al., 2018) and RGIN; c) *Topological message passing models*: SIN (Bodnar et al., 2021b), CIN (Bodnar et al., 2021a), and CIN++ (Giusti et al., 2024).

**Relational Rewiring.** We apply relational rewiring for 40 iterations (Section 4) using three choices for REWIREALGO: SDRF (Topping et al., 2022), FoSR (Karhadkar et al., 2023), and AFRC (Fesser & Weber, 2023). Due to computational constraints, we run the three choices on all datasets except IMDB with Clique graph lifting, where we only run FoSR. We use fixed, dataset- and model-agnostic hyperparameters, diverging from prior work where hyperparameter sweeps are carried out. It is

important to note that hyperparameter tuning can significantly impact performance on downstream tasks, as highlighted by, e.g., Tori et al. (2024).

**Training.** Models are trained for up to 500 epochs, with early stopping and learning rate decay based on a validation set. Additional details can be found in Appendix E. Results are reported as *mean ± standard error* over 10 trials.

**Results.** Table 1 shows test accuracies for the TUDataset experiments. Rewiring generally boosts performance for base graphs across models and datasets, and the impact of rewiring with our dataset- and model-agnostic choice of hyperparameters varies across datasets, with relational and topological models performance responding to rewiring similarly to graph models.

| Lift | Model | ENZYMES No Rew. | ENZYMES Best Rew. | IMDB-B No Rew. | IMDB-B FoSR | MUTAG No Rew. | MUTAG Best Rew. | NCI1 No Rew. | NCI1 Best Rew. | PROTEINS No Rew. | PROTEINS Best Rew. |
|---|---|---|---|---|---|---|---|---|---|---|---|
| None | SGC | $18.3 \pm 1.2$ | $21.5 \pm 1.6$ | $49.5 \pm 1.5$ | $50.0 \pm 1.8$ | $64.5 \pm 5.8$ | $70.0 \pm 2.6$ | $55.2 \pm 1.0$ | $54.4 \pm 0.7$ | $62.2 \pm 1.4$ | $65.0 \pm 1.5$ |
| | GCN | $32.2 \pm 2.0$ | $30.7 \pm 1.5$ | $49.1 \pm 1.4$ | $47.9 \pm 1.0$ | $71.0 \pm 3.8$ | $83.0 \pm 1.5$ | $48.3 \pm 0.6$ | $49.1 \pm 0.8$ | $73.1 \pm 1.2$ | $75.8 \pm 1.7$ |
| | GIN | $47.2 \pm 1.9$ | $50.0 \pm 2.7$ | $71.7 \pm 1.5$ | $67.1 \pm 1.5$ | $83.0 \pm 3.1$ | $88.0 \pm 2.4$ | $77.2 \pm 0.5$ | $77.9 \pm 0.5$ | $70.6 \pm 1.4$ | $72.2 \pm 0.8$ |
| | RGCN | $33.8 \pm 1.6$ | $42.5 \pm 1.3$ | $47.6 \pm 1.4$ | $68.0 \pm 1.3$ | $72.5 \pm 2.5$ | $83.5 \pm 1.8$ | $53.2 \pm 0.7$ | $63.4 \pm 1.1$ | $71.9 \pm 1.6$ | $75.6 \pm 1.2$ |
| | RGIN | $46.8 \pm 1.8$ | $49.8 \pm 2.0$ | $69.6 \pm 1.6$ | $48.9 \pm 2.9$ | $81.5 \pm 1.7$ | $85.5 \pm 2.0$ | $76.8 \pm 1.1$ | $77.0 \pm 0.7$ | $70.8 \pm 1.2$ | $72.4 \pm 1.4$ |
| | SIN | $47.5 \pm 2.3$ | $46.8 \pm 2.1$ | $70.0 \pm 1.4$ | $63.0 \pm 2.7$ | $88.5 \pm 3.0$ | $85.5 \pm 1.7$ | $77.0 \pm 0.6$ | $76.4 \pm 0.4$ | $70.2 \pm 1.3$ | $73.2 \pm 1.5$ |
| | CIN | $50.0 \pm 1.9$ | $49.0 \pm 2.0$ | $58.1 \pm 4.0$ | $58.4 \pm 2.7$ | $86.5 \pm 1.8$ | $87.0 \pm 2.4$ | $51.4 \pm 2.5$ | $66.2 \pm 2.0$ | $70.7 \pm 1.0$ | $71.0 \pm 1.4$ |
| | CIN++ | $48.5 \pm 1.9$ | $51.0 \pm 1.5$ | $66.6 \pm 3.7$ | $56.0 \pm 3.9$ | $85.0 \pm 3.4$ | $91.0 \pm 2.3$ | $60.8 \pm 3.8$ | $64.8 \pm 3.1$ | $67.9 \pm 1.9$ | $71.4 \pm 1.4$ |
| Clique | SGC | $14.5 \pm 1.4$ | $16.8 \pm 0.9$ | $48.7 \pm 2.2$ | $47.8 \pm 1.6$ | $70.0 \pm 3.3$ | $69.5 \pm 2.6$ | $50.0 \pm 1.3$ | $56.8 \pm 0.8$ | $59.9 \pm 1.8$ | $59.1 \pm 1.4$ |
| | GCN | $30.7 \pm 1.2$ | $30.2 \pm 2.4$ | $64.0 \pm 3.1$ | $65.5 \pm 3.1$ | $67.0 \pm 3.5$ | $81.5 \pm 2.9$ | $48.4 \pm 0.4$ | $49.6 \pm 0.6$ | $69.9 \pm 0.6$ | $75.0 \pm 1.4$ |
| | GIN | $44.0 \pm 1.7$ | $48.5 \pm 2.2$ | $69.1 \pm 1.2$ | $70.8 \pm 1.1$ | $83.0 \pm 2.8$ | $82.5 \pm 2.6$ | $78.8 \pm 0.7$ | $78.2 \pm 0.6$ | $68.7 \pm 1.4$ | $72.8 \pm 1.2$ |
| | RGCN | $48.8 \pm 1.2$ | $45.2 \pm 1.5$ | $71.0 \pm 1.0$ | $69.7 \pm 1.5$ | $79.5 \pm 1.7$ | $81.5 \pm 3.8$ | $78.8 \pm 0.7$ | $75.0 \pm 0.9$ | $72.4 \pm 1.6$ | $74.2 \pm 1.2$ |
| | RGIN | $50.8 \pm 1.5$ | $55.8 \pm 2.5$ | $71.6 \pm 0.9$ | $69.0 \pm 1.4$ | $86.0 \pm 2.3$ | $85.0 \pm 2.4$ | $79.2 \pm 0.6$ | $79.5 \pm 0.4$ | $71.5 \pm 1.5$ | $71.8 \pm 1.7$ |
| | SIN | $51.0 \pm 2.4$ | $46.5 \pm 1.2$ | $53.0 \pm 1.9$ | $64.0 \pm 2.3$ | $87.0 \pm 3.2$ | $83.5 \pm 1.7$ | $76.6 \pm 1.3$ | $75.4 \pm 0.7$ | $66.9 \pm 1.3$ | $70.4 \pm 1.2$ |
| | CIN | $49.8 \pm 1.9$ | $46.7 \pm 1.3$ | $52.6 \pm 2.4$ | $68.1 \pm 1.6$ | $85.5 \pm 2.8$ | $86.5 \pm 2.8$ | $51.8 \pm 2.3$ | $72.5 \pm 0.8$ | $70.7 \pm 1.2$ | $70.3 \pm 0.8$ |
| | CIN++ | $50.5 \pm 2.1$ | $52.7 \pm 1.6$ | $62.8 \pm 3.8$ | $64.7 \pm 1.5$ | $90.5 \pm 2.2$ | $84.5 \pm 3.3$ | $61.5 \pm 4.6$ | $76.8 \pm 0.4$ | $68.3 \pm 1.3$ | $71.9 \pm 1.0$ |
| Ring | SGC | $16.5 \pm 1.6$ | $19.3 \pm 1.2$ | $50.1 \pm 1.9$ | $49.9 \pm 1.9$ | $65.5 \pm 3.6$ | $75.0 \pm 5.7$ | $51.5 \pm 1.2$ | $51.4 \pm 0.4$ | $44.8 \pm 2.3$ | $49.3 \pm 3.6$ |
| | GCN | $34.8 \pm 1.3$ | $32.0 \pm 1.4$ | $46.9 \pm 1.4$ | $48.0 \pm 1.2$ | $72.0 \pm 2.7$ | $77.5 \pm 2.4$ | $49.3 \pm 0.9$ | $49.4 \pm 0.6$ | $72.2 \pm 1.3$ | $72.7 \pm 1.2$ |
| | GIN | $46.7 \pm 2.4$ | $47.0 \pm 1.6$ | $70.1 \pm 1.7$ | $73.1 \pm 1.1$ | $88.0 \pm 2.1$ | $89.0 \pm 1.9$ | $78.9 \pm 0.6$ | $77.5 \pm 0.9$ | $69.8 \pm 1.4$ | $72.1 \pm 0.9$ |
| | RGCN | $35.2 \pm 1.7$ | $45.7 \pm 1.5$ | $71.1 \pm 1.4$ | $70.0 \pm 1.6$ | $83.5 \pm 2.7$ | $84.0 \pm 2.1$ | $73.9 \pm 0.5$ | $73.5 \pm 0.5$ | $70.7 \pm 1.6$ | $71.3 \pm 1.2$ |
| | RGIN | $45.3 \pm 1.3$ | $49.2 \pm 1.5$ | $68.6 \pm 1.2$ | $67.2 \pm 1.8$ | $87.0 \pm 2.9$ | $87.5 \pm 2.4$ | $78.4 \pm 0.7$ | $79.8 \pm 0.7$ | $68.8 \pm 1.5$ | $71.3 \pm 1.5$ |
| | SIN | $40.3 \pm 2.2$ | $48.0 \pm 2.0$ | $50.6 \pm 1.9$ | $60.9 \pm 2.1$ | $85.0 \pm 2.1$ | $88.5 \pm 2.5$ | $80.0 \pm 0.8$ | $79.1 \pm 0.9$ | $70.6 \pm 1.1$ | $72.1 \pm 0.7$ |
| | CIN | $47.5 \pm 2.0$ | $49.5 \pm 2.0$ | $48.6 \pm 1.6$ | $66.1 \pm 2.0$ | $93.5 \pm 2.1$ | $95.0 \pm 1.3$ | $51.6 \pm 3.2$ | $76.5 \pm 0.5$ | $68.7 \pm 1.4$ | $68.5 \pm 1.6$ |
| | CIN++ | $47.5 \pm 1.7$ | $46.3 \pm 1.8$ | $66.0 \pm 1.4$ | $67.8 \pm 1.3$ | $85.5 \pm 2.0$ | $90.0 \pm 2.7$ | $56.8 \pm 4.5$ | $76.0 \pm 0.6$ | $68.1 \pm 1.2$ | $70.1 \pm 1.2$ |

Table 1: Test accuracy for TUDataset experiments. Each value is presented as the mean ± standard error across ten trials. The best-performing result for each dataset is highlighted in gold, while the second-best is in silver. The results after rewiring are shown with green text if the mean increased and red text if the mean decreased.

## 5.2 SYNTHETIC BENCHMARK: RINGTRANSFER

We confirm the theoretical results from Section 3 using the RINGTRANSFER benchmark, a graph feature transfer task designed to tease out the effect of long-range dependencies in message-passing models using rings of growing size. We follow the experimental setup of Karhadkar et al. (2023) and Di Giovanni et al. (2023), and provide more details in Appendix E.2. We test the impact of neural network hidden dimensions (Section 3.4), relational structure depth (Section 3.3), and relational structure local geometry (Sections 3.4 and 4) on task performance by varying the hidden dimensions, ring sizes, and rewiring iterations. The results, consistent with the theory, demonstrate that increasing network hidden dimensions improves performance up to a point, after which it declines, potentially due to overfitting. Larger ring sizes lead to performance deterioration, as the effects of long-range dependencies and bottlenecks start to take over. At the same time, rewiring improves performance by facilitating communication between distant nodes and mitigating oversquashing. As illustrated in Figure 2, message passing on graphs and simplicial complexes demonstrate similar trends, consistent with our theoretical predictions.

## 5.3 ADDITIONAL EXPERIMENTS AND ANALYSES

We report additional analyses in Appendix D.1 and Appendix D.2. There, we visualize the curvature of relational structures for dumbbell graphs and their corresponding clique complexes. We also reports a statistically significant linear relationship between the weighted curvature of graphs and their lifted clique complexes. These interesting patterns merit further investigation. We also present the following additional experiments: (1) neighbors match for path of cliques and tree datasets in

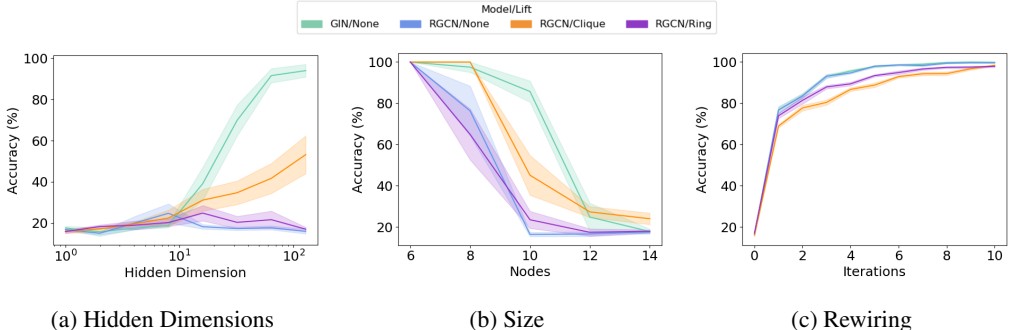

Figure 2: Performance on RINGTRANSFER obtained by varying model hidden dimensions (left), ring size (middle), and number of rewiring iterations (right).

Appendix D.4, (2) graph regression for ZINC in Appendix D.5, (3) node classification for COR-NELL, WISCONSIN, TEXAS, CORA, and CITESEER in Appendix D.6, (4) simplex pruning on the MUTAG dataset in Appendix D.8, and (5) full TUDataset results in Appendix E.1.

## 6   DISCUSSION AND CONCLUSIONS

This work addresses pressing questions about oversquashing in topological networks and higher-order generalizations of rewiring algorithms raised by the TDL community (Questions 2 and 9 of Papamarkou et al., 2024). We introduce a theoretical framework for unifying graph and topological message passing via relational structures, extending key graph-theoretic results on oversquashing and rewiring strategies to higher-order networks such as simplicial complexes via *influence graphs* that capture the aggregated message passing dynamical structure on relational structures. Our approach applies broadly to other message-passing schemes, including relational GNNs, high-order GNNs, and CW networks, providing a foundation for future theoretical and empirical research. Empirical results on real-world datasets show that simplicial networks respond to rewiring similarly to graph networks, and synthetic benchmarks further confirm our theoretical findings.

Certain aspects are worthy of further investigation. In particular, we compare message passing on graphs and their clique complexes through proxies (e.g., performance on tasks), as the significant differences in size and structure make direct empirical comparisons, e.g., of curvature, less theoretically rigorous. While we observe statistically significant patterns when comparing weighted curvatures, further theoretical and empirical investigation is needed. Furthermore, the rewiring algorithms we applied our relational rewiring heuristic to were not originally designed with weighted directed influence graphs in mind. Potentially, further improvements could be obtained by implementing algorithms specifically tailored for rewiring weighted directed graphs.

For future work, exploring global geometric properties of relational structures, studying oversmoothing, and empirically analyzing more relational message-passing schemes are promising directions. Developing theoretical tools and tailored rewiring heuristics for weighted directed graphs will be crucial, as will be tools for direct comparisons of message-passing across different relational structures. Furthermore, systematically and empirically assessing our framework's higher-order extensions of more state-of-the-art (SoTA) graph rewiring solutions is essential. By unifying topological message passing into message passing on relational structures and generalizing graph-based analysis to this setting, we hope that the present work can aid in both rigorous analysis and direct comparison between different higher-order message passing schemes.

Lastly, for practitioners, we recommend topological message passing as yet another relational learning tool with relational rewiring as a preprocessing step.

**Reproducibility Statement**  The code for replicating our experiments is available at `https://github.com/chapman20j/Simplicial-Oversquashing`. Experimental settings and implementation details are described in Section 5, and Appendices E and F.

ACKNOWLEDGMENTS

GM and JC have been supported by NSF CCF-2212520. GM also acknowledges support from NSF DMS-2145630, DFG SPP 2298 project 464109215, and BMBF in DAAD project 57616814.

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

# Appendix

## Table of Contents

## A   RELATED WORK

**Topological Networks**    Topological deep learning (TDL) integrates algebraic topology with neural networks to create message-passing schemes that are more expressive than graph neural networks. An early contribution was the simplicial Weisfeiler-Lehman (SWL) test introduced by Bodnar et al. (2021a), which extends the Weisfeiler-Lehman (WL) test from graphs to simplicial complexes. Simplicial neural networks (SNNs) built on SWL generalize graph isomorphism networks

(GIN) (Xu et al., 2019), offering provably stronger expressiveness. CW networks (CWNs) (Bodnar et al., 2021b) extend message passing to cell complexes, achieving greater power than the traditional WL test and exceeding the 3-WL test. These hierarchical, geometrically-grounded representations enable effective handling of higher-order interactions. Hajij et al. (2020) proposed a general message-passing scheme for cell complexes, though it lacks formal analysis of expressiveness and complexity. In contrast, Bodnar et al. (2022) and Suk et al. (2022) introduced neural sheaf diffusion models, which learn sheaf structures over graphs, particularly excelling in heterophilic graph tasks. Attention mechanisms have been integrated into topological message passing in works such as Goh et al. (2022) for simplicial complexes, Barbero et al. (2022) for sheaves, and Giusti et al. (2023) for cellular complexes. Additionally, Hajij et al. (2022) extends message-passing to combinatorial complexes. Our work represents a first step toward extending the theory of oversquashing and oversmoothing, widely studied in graph neural networks, to these topological networks.

We note that our relational structures approach generalizes a powerful perspective in which simplicial complexes and similar constructs are treated as augmented Hasse diagrams—a viewpoint that has shown both practical and theoretical advantages. Recent works have leveraged this perspective: Hajij et al. (2023) provided a general description of topological message passing schemes, Eitan et al. (2024) explored the expressivity limits of topological message passing, and Papillon et al. (2024) introduced a framework for systematically transforming any graph neural network into a topological analog. Our work aligns with and contributes to this growing body of research.

For detailed surveys of topological deep learning architectures, we refer to Papillon et al. (2023b) and Giusti (2024). We also refer the reader to the recent position paper by Papamarkou et al. (2024) on open problems in TDL.

**Oversquashing, Oversmoothing, and Graph Rewiring**  Oversquashing refers to the phenomenon where information from distant nodes is compressed into fixed-size vectors during message passing, limiting the ability of a model to capture long-range dependencies. In GNNs, this has been extensively studied, with works such as those of Alon & Yahav (2021), Topping et al. (2022), and Di Giovanni et al. (2023) naming and relating the phenomenon to the geometric properties of graphs. To address this issue, numerous techniques have been proposed, including spatial and curvature-based rewiring methods such as SDRF (Topping et al., 2022), BORF (Nguyen et al., 2023), and AFRC (Fesser & Weber, 2023), which modify the graph structure to improve connectivity and alleviate oversquashing. Spectral rewiring approaches, such as FOSR (Karhadkar et al., 2023), optimize the graph spectral gap to enhance long-range message propagation, while implicit methods such as Graph Beltrami Diffusion (Chamberlain et al., 2021) and Graph Transformers (Dwivedi & Bresson, 2020) allow information flow across a fully connected graph without explicitly modifying the topology.

Related to oversquashing, another significant challenge in GNNs is oversmoothing, where node features become indistinguishable as they are excessively aggregated through layers of message passing. Li et al. (2018), Oono & Suzuki (2020), and Nt & Maehara (2019) have identified and theoretically analyzed oversmoothing, showing that it limits the effectiveness of deep GNNs. Generally, there is a trade-off between mitigating oversquashing and avoiding oversmoothing, and various rewiring techniques aim to balance these competing objectives.

Following foundational work on understanding and mitigating oversquashing and oversmoothing in graph neural networks, the field has rapidly evolved with various innovative and sophisticated approaches in recent years. For example, Liu et al. (2023) introduced CurvDrop, a Ricci curvature-based, topology-aware method to address both failure modes of message passing, while Sun et al. (2023) proposed DeepRicci, a self-supervised Riemannian model designed to alleviate oversquashing. Qian et al. (2023) developed a probabilistic framework leveraging differentiable $k$-subset sampling, and Shen et al. (2024a) introduced a technique based on maximization operations in graph convolution. Shen et al. (2024b) used effective resistance for graph rewiring and preprocessing. Li et al. (2024) combined graph rewiring with ordered neurons, while Stanovic et al. (2024) proposed maximal independent set-based pooling to mitigate both oversquashing and oversmoothing. Shi et al. (2024) presented a physics-informed, agnostic method targeting both failure modes, and Pei et al. (2024) addressed them through multi-track message passing. Attali et al. (2024) employed Delaunay triangulation of features, while Huang et al. (2024a) proposed particular spectral filters

to improve spectral GNNs. Collectively, these works demonstrate the breadth and depth of recent advancements, highlighting a vibrant and rapidly evolving research landscape.

However, the study of oversquashing and oversmoothing in topological message passing remains largely unexplored. While works such as those of Bodnar et al. (2021b), Bodnar et al. (2021a), and Giusti et al. (2024) have alluded to the potential of topological message passing to capture group interactions and long-range dependencies, a theoretical analysis of oversquashing and oversmoothing in higher-order structures like simplicial or cellular complexes is, to the best of our knowledge, absent from the literature. Our work aims to fill this gap, providing a rigorous first step in studying oversquashing in topological message passing. Moreover, our framework for extending graph rewiring to relational rewiring (Section 4), along with future innovations from the topological deep learning community, promises a wealth of topological and relational rewiring techniques for further assessment and refinement.

For more exhaustive expositions on oversquashing and oversmoothing, we refer the reader to the excellent recent surveys of Shi et al. (2023) and Rusch et al. (2023).

**Relational Learning**  Relational graph neural networks (R-GNNs) extend traditional GNNs to handle multi-relational data, particularly in the context of knowledge graphs, where nodes represent entities and edges capture diverse types of relationships between them. Knowledge graphs, such as those used in knowledge representation and reasoning tasks (Nickel et al., 2015), are inherently multi-relational and benefit significantly from RGNNs, which model the varying nature of relations explicitly. The relational graph convolutional network (R-GCN) (Schlichtkrull et al., 2018) is a foundational approach that assigns distinct transformations to each relation type, allowing for efficient representation learning on multi-relational graphs. Extensions such as Relational Graph Attention Networks (RGAT) (Veličković et al., 2018) incorporate attention mechanisms, enabling the model to focus on the most important relations during message passing. Additionally, Composition-based Graph Convolutional Networks (CompGCN) (Vashishth et al., 2019) apply compositional operators to better capture interactions in knowledge graphs.

Recent works in relational learning have introduced various extensions for modeling multi-relational and higher-order interactions. Hypergraph-based approaches, such as Fatemi et al. (2023) and Huang et al. (2024b), unify relational reasoning with hypergraph structures, enabling the modeling of higher-order relationships. Message-passing frameworks like the one proposed by Yadati (2020) extend traditional GNN paradigms to ordered and recursive hypergraphs. Additionally, tensor decomposition methods such as GETD (Liu et al., 2020) represent hypergraphs as high-dimensional tensors, allowing efficient encoding of hyper-relational data. We refer the reader to the excellent survey by Antelmi et al. (2023) for a comprehensive overview of relational hypergraphs and their representation learning techniques. Robinson et al. (2024) recently extended relational learning to relational databases consisting of data laid out across multiple tables. Lastly, topological deep learning, which is one focus of this work, is a new frontier in relational learning (Papamarkou et al., 2024).

Our work unifies relational graph neural networks and topological neural networks by viewing complexes as relational structures, bridging the gap between the two fields.

**Graph Lifting**  Graph lifting transforms a graph into a higher-dimensional structure to enable more expressive message-passing schemes. For example, higher-order graph neural networks ($k$-GNNs) (Morris et al., 2019) lift graphs by representing $k$-node subgraphs as entities, capturing more complex relationships between nodes. Similarly, Chen et al. (2019) and Maron et al. (2019) proposed lifting graphs to higher-order structures to improve graph isomorphism testing. In topological message passing, graphs are lifted into structures like clique complexes to capture interactions beyond pairwise relationships. Recently, this topic has gained increased attention, as highlighted by the ICML Topological Deep Learning Challenge 2024 (Papillon et al., 2023a), which emphasized the development of topological lifting techniques across various data structures, including graphs, hypergraphs, and simplicial complexes. Among these, high-order graphs, simplicial complexes, and cellular complexes fit naturally within the relational structure framework and can be analyzed uniformly through this lens.

## B  ADDITIONAL REMARKS

### B.1  REMARKS FOR SECTION 3.2

**Remark B.1** (Ollivier-Ricci Curvature for Weighted Directed Graphs). *We recall the definition of Ollivier-Ricci curvature (ORC) for weighted directed graphs from Eidi & Jost (2020). Consider a weighted directed graph $\mathcal{G} = (\mathcal{S}, \mathcal{E}, w)$, where $\mathcal{S}$ is the set of entities (nodes), $\mathcal{E}$ is the set of directed edges, and $w : \mathcal{E} \to \mathbb{R}_{\geq 0}$ assigns non-negative weights to edges. We abuse notation, and for any two entities $\xi, \eta \in \mathcal{S}$ without an edge $(\xi \to \eta) \in \mathcal{E}$, we write $w_{\xi \to \eta} = 0$. For each entity $\sigma \in \mathcal{S}$, denote the* weighted out-degree *and* weighted in-degree*, as in Definition 3.3, by:*

$$w_\sigma^{\text{out}} = \sum_{(\sigma \to \eta) \in \mathcal{E}} w_{\sigma \to \eta}, \quad w_\sigma^{\text{in}} = \sum_{(\xi \to \sigma) \in \mathcal{E}} w_{\xi \to \sigma}.$$

*For a directed edge $(\sigma \to \tau) \in \mathcal{E}$ with non-zero weight $w_{\sigma \to \tau} > 0$, we define the probability measures:*

$$\mu_\tau^{\text{out}}(\xi) = \frac{w_{\tau \to \xi}}{w_\tau^{\text{out}}}, \quad \mu_\sigma^{\text{in}}(\xi) = \frac{w_{\xi \to \sigma}}{w_\sigma^{\text{in}}}, \quad \xi \in \mathcal{S}.$$

*The* Ollivier-Ricci curvature *of an edge $(\sigma \to \tau)$ with non-zero weight $w_{\sigma \to \tau}$ is then defined as:*

$$k(\sigma, \tau) = 1 - \frac{W\left(\mu_\sigma^{\text{in}}, \mu_\tau^{\text{out}}\right)}{w_{\sigma \to \tau}},$$

*where $W\left(\mu_\sigma^{\text{in}}, \mu_\tau^{\text{out}}\right)$ is the (directed) Wasserstein distance between the measures $\mu_\sigma^{\text{in}}$ and $\mu_\tau^{\text{out}}$, defined by:*

$$W\left(\mu_\sigma^{\text{in}}, \mu_\tau^{\text{out}}\right) = \inf_{\pi \in \Pi(\mu_\sigma^{\text{in}}, \mu_\tau^{\text{out}})} \sum_{\xi, \eta \in \mathcal{S}} \pi(\xi, \eta) d(\xi, \eta),$$

*where $\Pi(\mu_\sigma^{\text{in}}, \mu_\tau^{\text{out}})$ is the set of all joint probability measures on $\mathcal{S} \times \mathcal{S}$ with marginals $\mu_\sigma^{\text{in}}$ and $\mu_\tau^{\text{out}}$, and $d(\xi, \eta)$ is the distance from $\xi$ (incoming neighbor of $\sigma$) to $\eta$ (outgoing neighbor of $\tau$) in the graph $\mathcal{G}$ (the shortest directed path distance based on edge weights).*

**Remark B.2.** *Assume that the edge $(\sigma \to \tau) \in \mathcal{E}$ exists and has non-zero weight $w_{\sigma \to \tau} > 0$ in the influence graph $\mathcal{G}(\mathcal{S}, \mathbf{B})$. Then, the sensitivity bound for information flow from $\tau$ to $\sigma$ from Lemma 3.2 can be related to the Ollivier-Ricci curvature $k(\sigma, \tau)$ of the edge $(\sigma \to \tau)$ as follows:*

$$\left\| \frac{\partial \mathbf{h}_\sigma^{(2)}}{\partial \mathbf{h}_\tau^{(0)}} \right\|_1 \leq \left( \prod_{\ell=0}^{1} \alpha^{(\ell)} \beta^{(\ell)} \right) w_\tau^{\text{out}} w_\sigma^{\text{in}} \left( 1 - \frac{w_{\sigma \to \tau}}{w_{\max}^3} (1 - k(\sigma, \tau)) \right),$$

*where $w_{\max}^3$ is the maximum weighted 3-step path from incoming neighbors of $\sigma$ to outgoing neighbors of $\tau$. This result indicates that lower Ollivier-Ricci curvature (smaller $k(\sigma, \tau)$) leads to reduced sensitivity, thereby contributing to oversquashing. The connection leverages the existence of the reversed edge $(\sigma \to \tau)$ in addition to the edge $(\tau \to \sigma)$ over which information flows, which is not guaranteed to be the case in a generic relational message passing scheme, but is the case for undirected graphs where $k(\sigma, \tau) = k(\tau, \sigma)$. This result aligns with Theorem 4.5 from Nguyen et al. (2023) but without requiring their assumption of linearity in the message and update functions. Analyzing oversquashing for connections beyond 2-steps requires stronger assumptions, such as the influence graph $\mathcal{G}(\mathcal{S}, \mathbf{B})$ being strongly connected. These assumptions present obstructions to analyzing oversquashing in general relational structures with Ollivier-Ricci curvature.*

We provide the proof in Appendix C.2.

### B.2  REMARKS FOR SECTION 3.4

**Remark B.3** (Lipschitz Constants for MLP Message and Update Functions). *Consider the following message and update functions at layer $t$:*

*Message function:*

$$\psi_i^{(t)}\left(\mathbf{h}_\sigma^{(t)}, \mathbf{h}_{\xi_1}^{(t)}, \ldots, \mathbf{h}_{\xi_{n_i-1}}^{(t)}\right) = \mathbf{W}_i^{(t)} \begin{bmatrix} \mathbf{h}_\sigma^{(t)} \\ \mathbf{h}_{\xi_1}^{(t)} \\ \vdots \\ \mathbf{h}_{\xi_{n_i-1}}^{(t)} \end{bmatrix},$$

where $\mathbf{W}_i^{(t)}$ is a weight matrix of appropriate dimensions, and $[\cdot]$ denotes column-wise concatenation.

**Update function:**

$$\phi^{(t)}\left(\mathbf{m}_{\sigma,1}^{(t)},\ldots,\mathbf{m}_{\sigma,k}^{(t)}\right) = \mathbf{f}\left(\mathbf{W}^{(t)}\begin{bmatrix}\mathbf{g}\left(\mathbf{m}_{\sigma,1}^{(t)}\right)\\ \vdots \\ \mathbf{g}\left(\mathbf{m}_{\sigma,k}^{(t)}\right)\end{bmatrix}\right),$$

where $\mathbf{f}$ and $\mathbf{g}$ are the component-wise applications of non-linear functions $f$ and $g$ with bounded derivatives $C_f$ and $C_g$, respectively. I.e., $f, g : \mathbb{R} \to \mathbb{R}$, and $|f'(x)| \leq C_f$ and $|g'(x)| \leq C_g$ for all $x$.

Assume that the entries of all weight matrices $\mathbf{W}_i^{(t)}$ and $\mathbf{W}^{(t)}$ are bounded in absolute value by a constant $C_w > 0$. Then, the Lipschitz constants $\beta_i^{(t)}$ and $\alpha^{(t)}$ satisfy:

$$\beta_i^{(t)} \leq C_w p_{i,t},$$

*and*

$$\alpha^{(t)} \leq C_w C_f C_g p_{t+1},$$

where $p_{i,t}$ is the dimension of the message vector $\mathbf{m}_{\sigma,i}^{(t)}$, and $p_{t+1}$ is the output dimension of the update function $\phi_t$.

We provide the proof in Appendix C.4.

## C PROOFS

### C.1 PROOFS FOR SECTION 3.1

*Proof of Lemma 3.2.* First, we compute the Jacobian of Equation 3:

$$\frac{\partial \mathbf{h}_\sigma^{(s+1)}}{\partial \mathbf{h}_\tau^{(0)}} = \sum_{i=1}^k \frac{\partial \phi^{(s)}}{\partial \mathbf{m}_i^{(s)}}\frac{\partial \mathbf{m}_{\sigma,i}^{(s)}}{\partial \mathbf{h}_\tau^{(0)}}$$

$$= \sum_{i=1}^k \left(\frac{\partial \phi^{(s)}}{\partial \mathbf{m}_i^{(s)}}\right)\sum_{\boldsymbol{\xi}\in\mathcal{S}^{n_i-1}}\mathbf{A}_{\sigma,\boldsymbol{\xi}}^{R_i}\left[\frac{\partial \psi_i^{(s)}}{\partial \mathbf{h}_\sigma^{(s)}}\frac{\partial \mathbf{h}_\sigma^{(s)}}{\partial \mathbf{h}_\tau^{(0)}} + \sum_{j=1}^{n_i-1}\frac{\partial \psi_i^{(s)}}{\partial \mathbf{h}_{\xi_j}^{(s)}}\frac{\partial \mathbf{h}_{\xi_j}^{(s)}}{\partial \mathbf{h}_\tau^{(0)}}\right].$$

By the submultiplicative and additive properties of the induced 1-norm (maximum absolute column sum) and the boundedness of the Jacobians of the functions $\phi^{(s)}$ and $\psi_i^{(s)}$ (Assumption 1), we get:

$$\left\|\frac{\partial \mathbf{h}_\sigma^{(s+1)}}{\partial \mathbf{h}_\tau^{(0)}}\right\|_1 \leq \sum_{i=1}^k\left\|\frac{\partial \phi^{(s)}}{\partial \mathbf{m}_i^{(s)}}\right\|_1\sum_{\boldsymbol{\xi}\in\mathcal{S}^{n_i-1}}\mathbf{A}_{\sigma,\boldsymbol{\xi}}^{R_i}\left[\left\|\frac{\partial \psi_i^{(s)}}{\partial \mathbf{h}_\sigma^{(s)}}\right\|_1\left\|\frac{\partial \mathbf{h}_\sigma^{(s)}}{\partial \mathbf{h}_\tau^{(0)}}\right\|_1 + \sum_{j=1}^{n_i-1}\left\|\frac{\partial \psi_i^{(s)}}{\partial \mathbf{h}_{\xi_j}^{(s)}}\right\|_1\left\|\frac{\partial \mathbf{h}_{\xi_j}^{(s)}}{\partial \mathbf{h}_\tau^{(0)}}\right\|_1\right]$$

$$\leq \alpha^{(s)}\sum_{i=1}^k\beta_i^{(s)}\sum_{\boldsymbol{\xi}\in\mathcal{S}^{n_i-1}}\mathbf{A}_{\sigma,\boldsymbol{\xi}}^{R_i}\left[\left\|\frac{\partial \mathbf{h}_\sigma^{(s)}}{\partial \mathbf{h}_\tau^{(0)}}\right\|_1 + \sum_{j=1}^{n_i-1}\left\|\frac{\partial \mathbf{h}_{\xi_j}^{(s)}}{\partial \mathbf{h}_\tau^{(0)}}\right\|_1\right]$$

$$\leq \alpha^{(s)}\beta^{(s)}\sum_{i=1}^k\sum_{\boldsymbol{\xi}\in\mathcal{S}^{n_i-1}}\mathbf{A}_{\sigma,\boldsymbol{\xi}}^{R_i}\left[\left\|\frac{\partial \mathbf{h}_\sigma^{(s)}}{\partial \mathbf{h}_\tau^{(0)}}\right\|_1 + \sum_{j=1}^{n_i-1}\left\|\frac{\partial \mathbf{h}_{\xi_j}^{(s)}}{\partial \mathbf{h}_\tau^{(0)}}\right\|_1\right]$$

$$\leq \alpha^{(s)}\beta^{(s)}\left[\gamma\left\|\frac{\partial \mathbf{h}_\sigma^{(s)}}{\partial \mathbf{h}_\tau^{(0)}}\right\|_1 + \sum_{i=1}^k\sum_{\boldsymbol{\xi}\in\mathcal{S}^{n_i-1}}\mathbf{A}_{\sigma,\boldsymbol{\xi}}^{R_i}\sum_{j=1}^{n_i-1}\left\|\frac{\partial \mathbf{h}_{\xi_j}^{(s)}}{\partial \mathbf{h}_\tau^{(0)}}\right\|_1\right].$$

Here, we used that the entries of $\mathbf{A}^{R_i}$ are nonnegative, and $\sum_{i=1}^k \sum_{\boldsymbol{\xi}\in\mathcal{S}^{n_i-1}}\mathbf{A}_{\sigma,\boldsymbol{\xi}}^{R_i} \leq \gamma$.

We now prove the lemma using induction on the layer $t$. For the base case $t = 1$, we get

$$
\left\| \frac{\partial \mathbf{h}_\sigma^{(1)}}{\partial \mathbf{h}_\tau^{(0)}} \right\|_1 \leq \alpha^{(0)} \beta^{(0)} \left[ \gamma \left\| \frac{\partial \mathbf{h}_\sigma^{(0)}}{\partial \mathbf{h}_\tau^{(0)}} \right\|_1 + \sum_{i=1}^{k} \sum_{\boldsymbol{\xi} \in \mathcal{S}^{n_i - 1}} \mathbf{A}_{\sigma, \boldsymbol{\xi}}^{R_i} \sum_{j=1}^{n_i - 1} \left\| \frac{\partial \mathbf{h}_{\xi_j}^{(0)}}{\partial \mathbf{h}_\tau^{(0)}} \right\|_1 \right]
$$

$$
= \alpha^{(0)} \beta^{(0)} \left[ \gamma \mathbf{I}_{\sigma, \tau} + \sum_{i=1}^{k} \sum_{\boldsymbol{\xi} \in \mathcal{S}^{n_i - 1}} \mathbf{A}_{\sigma, \boldsymbol{\xi}}^{R_i} \sum_{j=1}^{n_i - 1} \mathbf{I}_{\xi_j, \tau} \right]
$$

$$
= \alpha^{(0)} \beta^{(0)} \left[ \gamma \mathbf{I}_{\sigma, \tau} + \sum_{i=1}^{k} \tilde{\mathbf{A}}_{\sigma, \tau}^{R_i} \right]
$$

$$
= \alpha^{(0)} \beta^{(0)} \left( \gamma \mathbf{I} + \sum_{i=1}^{k} \tilde{\mathbf{A}}^{R_i} \right)_{\sigma, \tau}
$$

$$
= \alpha^{(0)} \beta^{(0)} \left( \mathbf{B}^1 \right)_{\sigma, \tau},
$$

where we used that $\tilde{\mathbf{A}}_{\sigma, \tau}^{R_i} = \sum_{j=1}^{n_i - 1} \sum_{\boldsymbol{\xi} \in \mathcal{S}^{n_i - 2}} \mathbf{A}_{\sigma, \xi_1, \ldots, \xi_{j-1}, \tau, \xi_j, \ldots, \xi_{n_i - 2}}^{R_i}$. This proves the base case.

For the induction step, assume the bound holds for $t$. We now compute:

$$
\left\| \frac{\partial \mathbf{h}_\sigma^{(t+1)}}{\partial \mathbf{h}_\tau^{(0)}} \right\|_1 \leq \alpha^{(t)} \beta^{(t)} \left[ \gamma \left\| \frac{\partial \mathbf{h}_\sigma^{(t)}}{\partial \mathbf{h}_\tau^{(0)}} \right\|_1 + \sum_{i=1}^{k} \sum_{\boldsymbol{\xi} \in \mathcal{S}^{n_i - 1}} \mathbf{A}_{\sigma, \boldsymbol{\xi}}^{R_i} \sum_{j=1}^{n_i - 1} \left\| \frac{\partial \mathbf{h}_{\xi_j}^{(t)}}{\partial \mathbf{h}_\tau^{(0)}} \right\|_1 \right]
$$

$$
\leq \left( \prod_{\ell=0}^{t} \alpha^{(\ell)} \beta^{(\ell)} \right) \left[ \gamma \left( \mathbf{B}^t \right)_{\sigma, \tau} + \sum_{i=1}^{k} \sum_{\boldsymbol{\xi} \in \mathcal{S}^{n_i - 1}} \mathbf{A}_{\sigma, \boldsymbol{\xi}}^{R_i} \sum_{j=1}^{n_i - 1} \left( \mathbf{B}^t \right)_{\xi_j, \tau} \right]
$$

$$
= \left( \prod_{\ell=0}^{t} \alpha^{(\ell)} \beta^{(\ell)} \right) \left[ \left( \gamma \mathbf{I} \mathbf{B}^t \right)_{\sigma, \tau} + \left( \left( \sum_{i=1}^{k} \tilde{\mathbf{A}}^{R_i} \right) \mathbf{B}^t \right)_{\sigma, \tau} \right]
$$

$$
= \left( \prod_{\ell=0}^{t} \alpha^{(\ell)} \beta^{(\ell)} \right) \left( \left( \gamma \mathbf{I} + \sum_{i=1}^{k} \tilde{\mathbf{A}}^{R_i} \right) \mathbf{B}^t \right)_{\sigma, \tau}
$$

$$
= \left( \prod_{\ell=0}^{t} \alpha^{(\ell)} \beta^{(\ell)} \right) \left( \mathbf{B}^{t+1} \right)_{\sigma, \tau}.
$$

To see this, note that:

$$
\left( \left( \sum_{i=1}^{k} \tilde{\mathbf{A}}^{R_i} \right) \mathbf{B}^t \right)_{\sigma, \tau} = \sum_{i=1}^{k} \sum_{\nu \in \mathcal{S}} \tilde{\mathbf{A}}_{\sigma, \nu}^{R_i} \left( \mathbf{B}^t \right)_{\nu, \tau}
$$

$$
= \sum_{i=1}^{k} \sum_{j=1}^{n_i - 1} \sum_{\boldsymbol{\xi} \in \mathcal{S}^{n_i - 2}} \sum_{\nu \in \mathcal{S}} \mathbf{A}_{\sigma, \xi_1, \ldots, \xi_{j-1}, \nu, \xi_j, \ldots, \xi_{n_i - 2}}^{R_i} \left( \mathbf{B}^t \right)_{\nu, \tau}
$$

$$
= \sum_{i=1}^{k} \sum_{\boldsymbol{\xi} \in \mathcal{S}^{n_i - 1}} \mathbf{A}_{\sigma, \boldsymbol{\xi}}^{R_i} \sum_{j=1}^{n_i - 1} \left( \mathbf{B}^t \right)_{\xi_j, \tau}.
$$

This completes the proof. $\qquad \square$

## C.2 PROOFS FOR SECTION 3.2 AND APPENDIX B.1

*Proof of Proposition 3.4.* In the influence graph $\mathcal{G}(\mathcal{S}, \mathbf{B})$, the edge weights correspond to the entries of $\mathbf{B}$, that is, $w_{\tau \to \sigma} = \mathbf{B}_{\sigma, \tau}$. Therefore, the $(\sigma, \tau)$ entry of $\mathbf{B}^2$ is

$$
(\mathbf{B}^2)_{\sigma, \tau} = \sum_{\xi \in \mathcal{S}} \mathbf{B}_{\sigma, \xi} \mathbf{B}_{\xi, \tau} = \sum_{\xi \in \mathcal{S}} w_{\xi \to \sigma} \cdot w_{\tau \to \xi} = w_T.
$$

From Lemma 3.2 with $t = 2$, we have

$$\left\| \frac{\partial \mathbf{h}_\sigma^{(2)}}{\partial \mathbf{h}_\tau^{(0)}} \right\|_1 \leq \left( \prod_{\ell=0}^{1} \alpha^{(\ell)} \beta^{(\ell)} \right) (\mathbf{B}^2)_{\sigma,\tau} = \left( \prod_{\ell=0}^{1} \alpha^{(\ell)} \beta^{(\ell)} \right) w_T.$$

Rewriting the curvature formula to solve for $w_T$, and since $w_F \geq 0$, we get

$$w_T = \frac{1}{3} \left( \text{EFC}_\mathcal{G}(\tau, \sigma) + w_\tau^{\text{out}} + w_\sigma^{\text{in}} - 4 - 2 w_F \right)$$
$$\leq \frac{1}{3} \left( \text{EFC}_\mathcal{G}(\tau, \sigma) + w_\tau^{\text{out}} + w_\sigma^{\text{in}} - 4 \right).$$

Substituting back, we obtain the desired inequality. This completes the proof. $\qquad \square$

*Proof of Remark B.2.* Consider the influence graph $\mathcal{G}(\mathcal{S}, \mathbf{B}) = (\mathcal{S}, \mathcal{E}, w)$ derived from the matrix $\mathbf{B}$, where the edge weights correspond to the entries of $\mathbf{B}$: $w_{\xi \to \eta} = \mathbf{B}_{\eta, \xi}$. Assume that the reversed edge $(\sigma \to \tau) \in \mathcal{E}$ exists with weight $w_{\sigma \to \tau} > 0$. From Lemma 3.2 with $t = 2$, we have:

$$\left\| \frac{\partial \mathbf{h}_\sigma^{(2)}}{\partial \mathbf{h}_\tau^{(0)}} \right\|_1 \leq \left( \prod_{\ell=0}^{1} \alpha^{(\ell)} \beta^{(\ell)} \right) (\mathbf{B}^2)_{\sigma,\tau} = \left( \prod_{\ell=0}^{1} \alpha^{(\ell)} \beta^{(\ell)} \right) \sum_{\xi \in \mathcal{S}} w_{\xi \to \sigma} \cdot w_{\tau \to \xi} = \left( \prod_{\ell=0}^{1} \alpha^{(\ell)} \beta^{(\ell)} \right) w_T,$$

where $w_T = \sum_\xi w_{\tau \to \xi} \cdot w_{\xi \to \sigma}$.

We construct a *transference plan* to transport mass from $\mu_\sigma^{\text{in}}$ to $\mu_\tau^{\text{out}}$: We do not move $\frac{w_T}{w_\tau^{\text{out}} w_\sigma^{\text{in}}}$, but move the rest of the mass with a cost at most $w_{\max}^3$ from the incoming neighbors of $\sigma$ to the outgoing neighbors of $\tau$, and we get:

$$W\left( \mu_\sigma^{\text{in}}, \mu_\tau^{\text{out}} \right) \leq \left( 1 - \frac{w_T}{w_\sigma^{\text{in}} w_\tau^{\text{out}}} \right) w_{\max}^3.$$

The Ollivier-Ricci curvature of the edge $(\sigma \to \tau)$ is:

$$k(\sigma, \tau) = 1 - \frac{W\left( \mu_\sigma^{\text{in}}, \mu_\tau^{\text{out}} \right)}{w_{\sigma \to \tau}}.$$

Substituting and rearranging, we obtain:

$$w_T \leq w_\sigma^{\text{in}} w_\tau^{\text{out}} \left( 1 - \frac{w_{\sigma \to \tau}}{w_{\max}^3} (1 - k(\sigma, \tau)) \right).$$

We thus get:

$$\left\| \frac{\partial \mathbf{h}_\sigma^{(2)}}{\partial \mathbf{h}_\tau^{(0)}} \right\|_1 \leq \left( \prod_{\ell=0}^{1} \alpha^{(\ell)} \beta^{(\ell)} \right) w_T \leq \left( \prod_{\ell=0}^{1} \alpha^{(\ell)} \beta^{(\ell)} \right) w_\tau^{\text{out}} w_\sigma^{\text{in}} \left( 1 - \frac{w_{\sigma \to \tau}}{w_{\max}^3} (1 - k(\sigma, \tau)) \right).$$

This completes the proof. $\qquad \square$

### C.3    Proofs for Section 3.3

*Proof of Theorem 3.5.* We start with the bound from Lemma 3.2 and expand the right-hand side using the binomial theorem:

$$\left\| \frac{\partial \mathbf{h}_\sigma^{(r+m)}}{\partial \mathbf{h}_\tau^{(0)}} \right\|_1 \leq \left( \prod_{\ell=0}^{r+m-1} \alpha^{(\ell)} \beta^{(\ell)} \right) (\gamma \mathbf{I} + \tilde{\mathbf{A}})_{\sigma,\tau}^{r+m}$$
$$= \left( \prod_{\ell=0}^{r+m-1} \alpha^{(\ell)} \beta^{(\ell)} \right) \sum_{i=0}^{r+m} \binom{r+m}{i} \gamma^{r+m-i} \left( \tilde{\mathbf{A}}^i \right)_{\sigma,\tau}.$$

Since the combinatorial distance from $\tau$ to $\sigma$ in the graph $\mathcal{G}(\mathcal{S}, \tilde{\mathbf{A}})$ is $r$, the first $r - 1$ terms of the sum vanish:

$$\left\| \frac{\partial \mathbf{h}_\sigma^{(r+m)}}{\partial \mathbf{h}_\tau^{(0)}} \right\|_1 \leq \left( \prod_{\ell=0}^{r+m-1} \alpha^{(\ell)} \beta^{(\ell)} \right) \sum_{i=r}^{r+m} \binom{r+m}{i} \gamma^{r+m-i} \left( \tilde{\mathbf{A}}^i \right)_{\sigma, \tau}.$$

Using $\tilde{\mathbf{A}}_{\sigma,\tau}^i \leq \omega_i(\sigma, \tau) M^i \leq \omega_{r+m}(\sigma, \tau) M^i$ and letting $q = i - r$:

$$\left\| \frac{\partial \mathbf{h}_\sigma^{(r+m)}}{\partial \mathbf{h}_\tau^{(0)}} \right\|_1 \leq \left( \prod_{\ell=0}^{r+m-1} \alpha^{(\ell)} \beta^{(\ell)} \right) M^r \omega_{r+m}(\sigma, \tau) \sum_{q=0}^{m} \binom{r+m}{r+q} \gamma^{m-q} M^q.$$

We can bound $\binom{r+m}{r+q}$ as follows:

$$\binom{r+m}{r+q} = \frac{(r+m)(r-1+m)\cdots(1+m)}{(r+q)(r-1+q)\cdots(1+q)} \binom{m}{q} \leq \frac{(r+m)(r-1+m)\cdots(1+m)}{r!} \binom{m}{q}$$

$$\leq \left(1 + \frac{m}{r}\right) \cdots \left(1 + \frac{m}{1}\right) \binom{m}{q} \leq \left(1 + \frac{m}{m+1}\right)^{r-m} (1+m)^m \binom{m}{q}.$$

Substituting this bound, we get:

$$\left\| \frac{\partial \mathbf{h}_\sigma^{(r+m)}}{\partial \mathbf{h}_\tau^{(0)}} \right\|_1 \leq \left( \prod_{\ell=0}^{r+m-1} \alpha^{(\ell)} \beta^{(\ell)} \right) M^r \omega_{r+m}(\sigma, \tau) \left(1 + \frac{m}{m+1}\right)^{r-m} (1+m)^m \sum_{q=0}^{m} \binom{m}{q} \gamma^{m-q} M^q$$

$$= \left( \prod_{\ell=0}^{r+m-1} \alpha^{(\ell)} \beta^{(\ell)} \right) M^r \omega_{r+m}(\sigma, \tau) \left(1 + \frac{m}{m+1}\right)^{r-m} (1+m)^m (\gamma + M)^m$$

$$= \left( \prod_{\ell=r}^{r+m-1} \alpha^{(\ell)} \beta^{(\ell)} \right) \left( \frac{(1+m)^2}{2m+1}(\gamma + M) \right)^m \omega_{r+m}(\sigma, \tau) \left( \prod_{\ell=0}^{r-1} \alpha^{(\ell)} \beta^{(\ell)} \right) \left( \left(1 + \frac{m}{m+1}\right) M \right)^r.$$

Using $1 + \frac{m}{m+1} \leq 2$, $M \leq k$, and $\gamma \leq k$:

$$\left\| \frac{\partial \mathbf{h}_\sigma^{(r+m)}}{\partial \mathbf{h}_\tau^{(0)}} \right\|_1 \leq \left( \prod_{\ell=r}^{r+m-1} \alpha^{(\ell)} \beta^{(\ell)} \right) (2k(1+m))^m \omega_{r+m}(\sigma, \tau) \left( \prod_{\ell=0}^{r-1} \alpha^{(\ell)} \beta^{(\ell)} \right) (2M)^r.$$

Define $C = (\alpha_{\max} \beta_{\max})^m (2k(1+m))^m$. This depends only on $\alpha_{\max}$, $\beta_{\max}$, $k$, and $m$.

Finally, we can write:

$$\left\| \frac{\partial \mathbf{h}_\sigma^{(r+m)}}{\partial \mathbf{h}_\tau^{(0)}} \right\|_1 \leq C \omega_{r+m}(\sigma, \tau)(\alpha_{\max} \beta_{\max})^r (2M)^r. \tag{15}$$

This completes the proof. $\qquad\square$

### C.4 PROOFS FOR SECTION 3.4 AND APPENDIX B.2

*Proof of Remark B.3.* We derive the bounds for $\beta_i^{(t)}$ and $\alpha^{(t)}$ separately.

**Derivation of $\beta_i^{(t)}$:**

The message function is linear:

$$\psi_i^{(t)} \left( \mathbf{h}_\sigma^{(t)}, \mathbf{h}_{\xi_1}^{(t)}, \ldots, \mathbf{h}_{\xi_{n_i-1}}^{(t)} \right) = \mathbf{W}_i^{(t)} \mathbf{H}_{\sigma, \xi}^{(t)},$$

where $\mathbf{H}_{\sigma,\xi}^{(t)} = \begin{bmatrix} \mathbf{h}_{\sigma}^{(t)} \\ \mathbf{h}_{\xi_1}^{(t)} \\ \vdots \\ \mathbf{h}_{\xi_{n_i-1}}^{(t)} \end{bmatrix}$ is the concatenated feature vector.

We need to compute the Lipschitz constant $\beta_i^{(t)}$ of the message function $\psi_i^{(t)}$ with respect to each neighbor's feature vector $\mathbf{h}_{\xi_j}^{(t)}$.

The Jacobian of $\psi_i^{(t)}$ with respect to $\mathbf{h}_{\xi_j}^{(t)}$ is:

$$\frac{\partial \psi_i^{(t)}}{\partial \mathbf{h}_{\xi_j}^{(t)}} = \mathbf{W}_i^{(t)(:,\mathcal{I}_j)},$$

where $\mathbf{W}_i^{(t)(:,\mathcal{I}_j)}$ denotes the columns of $\mathbf{W}_i^{(t)}$ corresponding to $\mathbf{h}_{\xi_j}^{(t)}$.

Since the entries of $\mathbf{W}_i^{(t)}$ are bounded by $C_w$, and $\mathbf{h}_{\xi_j}^{(t)} \in \mathbb{R}^{p_t}$, the matrix $\frac{\partial \psi_i^{(t)}}{\partial \mathbf{h}_{\xi_j}^{(t)}}$ is of size $p_{i,t} \times p_t$ with entries bounded by $C_w$.

Using the induced matrix 1-norm:

$$\left\| \frac{\partial \psi_i^{(t)}}{\partial \mathbf{h}_{\xi_j}^{(t)}} \right\|_1 = \max_{1 \le l \le p_t} \sum_{k=1}^{p_{i,t}} \left| \left( \frac{\partial \psi_i^{(t)}}{\partial \mathbf{h}_{\xi_j}^{(t)}} \right)_{k,l} \right|.$$

Since each entry $\left| \left( \frac{\partial \psi_i^{(t)}}{\partial \mathbf{h}_{\xi_j}^{(t)}} \right)_{k,l} \right| \le C_w$, the sum over $k$ is bounded by $C_w p_{i,t}$. Therefore,

$$\left\| \frac{\partial \psi_i^{(t)}}{\partial \mathbf{h}_{\xi_j}^{(t)}} \right\|_1 \le C_w p_{i,t}.$$

Thus, the Lipschitz constant $\beta_i^{(t)}$ satisfies:

$$\beta_i^{(t)} \le C_w p_{i,t}.$$

**Derivation of $\alpha^{(t)}$:**

The update function is given by:

$$\phi^{(t)} \left( \mathbf{m}_{\sigma,1}^{(t)}, \ldots, \mathbf{m}_{\sigma,k}^{(t)} \right) = \mathbf{f} \left( \mathbf{W}^{(t)} \mathbf{M}_{\sigma}^{(t)} \right),$$

where $\mathbf{M}_{\sigma}^{(t)} = \begin{bmatrix} \mathbf{g} \left( \mathbf{m}_{\sigma,1}^{(t)} \right) \\ \vdots \\ \mathbf{g} \left( \mathbf{m}_{\sigma,k}^{(t)} \right) \end{bmatrix}.$

We need to compute the Lipschitz constant $\alpha^{(t)}$ of the update function $\phi^{(t)}$ with respect to each input message $\mathbf{m}_{\sigma,i}^{(t)}$.

First, compute the Jacobian of $\phi^{(t)}$ with respect to $\mathbf{m}_{\sigma,i}^{(t)}$:

$$\frac{\partial \phi^{(t)}}{\partial \mathbf{m}_{\sigma,i}^{(t)}} = \frac{\partial \phi^{(t)}}{\partial \mathbf{M}_{\sigma}^{(t)}} \frac{\partial \mathbf{M}_{\sigma}^{(t)}}{\partial \mathbf{m}_{\sigma,i}^{(t)}}.$$

Compute $\frac{\partial \mathbf{M}_\sigma^{(t)}}{\partial \mathbf{m}_{\sigma,i}^{(t)}}$:

$$\frac{\partial \mathbf{M}_\sigma^{(t)}}{\partial \mathbf{m}_{\sigma,i}^{(t)}} = \begin{bmatrix} \mathbf{0} \\ \vdots \\ \mathrm{diag}\left(g'\left(\mathbf{m}_{\sigma,i}^{(t)}\right)\right) \\ \vdots \\ \mathbf{0} \end{bmatrix},$$

where the non-zero block $\mathrm{diag}\left(g'\left(\mathbf{m}_{\sigma,i}^{(t)}\right)\right)$ is at position $i$.

Compute $\frac{\partial \boldsymbol{\phi}^{(t)}}{\partial \mathbf{M}_\sigma^{(t)}}$:

$$\frac{\partial \boldsymbol{\phi}^{(t)}}{\partial \mathbf{M}_\sigma^{(t)}} = \mathrm{diag}\left(f'\left(\mathbf{W}^{(t)}\mathbf{M}_\sigma^{(t)}\right)\right)\mathbf{W}^{(t)}.$$

Therefore,

$$\frac{\partial \boldsymbol{\phi}^{(t)}}{\partial \mathbf{m}_{\sigma,i}^{(t)}} = \mathrm{diag}\left(f'\left(\mathbf{W}^{(t)}\mathbf{M}_\sigma^{(t)}\right)\right)\mathbf{W}^{(t)(:,\mathcal{I}_i)}\mathrm{diag}\left(g'\left(\mathbf{m}_{\sigma,i}^{(t)}\right)\right),$$

where $\mathbf{W}^{(t)(:,\mathcal{I}_i)}$ denotes the columns of $\mathbf{W}^{(t)}$ corresponding to $\mathbf{m}_{\sigma,i}^{(t)}$.

The matrix $\frac{\partial \boldsymbol{\phi}^{(t)}}{\partial \mathbf{m}_{\sigma,i}^{(t)}}$ is of size $p_{t+1} \times p_{i,t}$.

Since $|f'(x)| \leq C_f$, $|g'(x)| \leq C_g$, and $|(\mathbf{W}^{(t)})_{k,l}| \leq C_w$, the entries of $\frac{\partial \boldsymbol{\phi}^{(t)}}{\partial \mathbf{m}_{\sigma,i}^{(t)}}$ are bounded by $C_w C_f C_g$.

Using the induced matrix 1-norm:

$$\left\|\frac{\partial \boldsymbol{\phi}^{(t)}}{\partial \mathbf{m}_{\sigma,i}^{(t)}}\right\|_1 = \max_{1 \leq l \leq p_{i,t}} \sum_{k=1}^{p_{t+1}} \left|\left(\frac{\partial \boldsymbol{\phi}^{(t)}}{\partial \mathbf{m}_{\sigma,i}^{(t)}}\right)_{k,l}\right|.$$

Each column $l$ sums over $k$ up to $p_{t+1}$ entries, each bounded by $C_w C_f C_g$. Therefore,

$$\left\|\frac{\partial \boldsymbol{\phi}^{(t)}}{\partial \mathbf{m}_{\sigma,i}^{(t)}}\right\|_1 \leq C_w C_f C_g p_{t+1}.$$

Thus, the Lipschitz constant $\alpha^{(t)}$ satisfies:

$$\alpha^{(t)} \leq C_w C_f C_g p_{t+1}.$$

$\square$

## D  SUPPLEMENTARY ANALYSES

### D.1  GRAPH LIFTING EXAMPLE AND CURVATURE

As shown in Section D.3, the edge curvature distribution is non-trivially impacted by the graph lifting procedure. We explore this more in this section.

One possible explanation for the general positive distribution shift is the widening of bottleneck regions as well as the addition of many nodes and edges in densely connected regions. We observe this qualitatively in Figure 3. We can see that the narrow path is now widened into two nodes instead of one. We also see that the clique regions gain a lot more nodes and edges than the path. The Ollivier-Ricci curvature becomes much more red, while the balanced Forman curvature and

augmented Forman curvature maintain a higher number of small, negatively curved edges. This example is consistent with the observations from Figure 5.

In the augmented Forman curvature plot, we can see that the curvature for edge `DI` becomes more negative. While this edge may not propagate information very well due to its negative curvature, there are now many edges for information to flow around this edge. This provides some qualitative evidence that incorporating global structure into analysis of graph lifting could present an important future direction.

### D.2 GRAPH LIFTING AND WEIGHTED CURVATURE

The addition of many nodes and edges, as well as the shift in the curvature distribution, makes direct comparisons between graphs and their corresponding lifts more challenging. While this widened bottleneck may alleviate oversquashing, measures like algebraic connectivity may not measure this effect since widened bottlenecks are counteracted by the addition of many nodes. Similarly, average curvature could be biased by the addition of many positively curved edges located in densely connected regions. We instead propose the *betweenness weighted curvature*

$$\text{wc} = \sum_{e \in E(G)} \text{bc}(e)\,\text{curv}(e), \tag{16}$$

where bc denotes betwenness centrality and curv denotes the Ollivier-Ricci curvature. This measure, originally in introduced by Münch (2022), places more weight on bottleneck edges which gives a weighted average of the curvature in the graph. This measure has the added benefit that it captures both local information propagation through curvature as well as global information propagation through betweenness centrality. To account for the influence of more positively weighted edges, we also consider the *negative betweenness weighted curvature*

$$\text{nwc} = \sum_{e \in E(G)} \text{bc}(e)\,\text{curv}(e)\chi_{\text{curv}(e)<0}, \tag{17}$$

where $\chi$ is the indicator function.

In Figure 4, we create a scatter plot of the weighted curvature of a graph and its corresponding clique complex: $(\text{wc}(G), \text{wc}(C))$. We can see that the weighted curvature generally becomes less negative after complex construction for Ollivier-Ricci and balanced Forman curvature, which may suggest that oversquashing is alleviated. However, the augmented Forman curvature makes the weighted curvature more negative for all graphs in the MUTAG dataset. This is consistent with Figure 5 which produced a small number of negatively curved edges for augmented Forman curvature. Interestingly, there is a strong linear correlation in each of the scatter plots. Further study is required to understand this trend.

### D.3 GRAPH LIFTING AND EDGE CURVATURE DISTRIBUTION

Understanding the impact on information propagation of lifting a graph to a clique complex is crucial. Graph lifting often adds numerous edges and nodes, complicating theoretical analysis, as the structural differences between the relational structures corresponding to a graph and its clique complex can be significant. As a first step, we analyze the edge curvature distribution to gain empirical insight into the clique graph lift. Figure 5 shows the kernel density estimate (KDE) of edge curvature on the MUTAG dataset. Ollivier-Ricci curvature (Nguyen et al., 2023) increases uniformly, while balanced Forman curvature (Topping et al., 2022) generally increases with some edges becoming slightly more negative. A more pronounced shift is observed with augmented Forman curvature (Fesser & Weber, 2023), but the overall effect on information propagation remains unclear, as global graph structure is not captured in this analysis.

### D.4 SYNTHETIC BENCHMARK: NEIGHBORSMATCH

The neighbors match experiment is a graph transfer task introduced by Alon & Yahav (2021) to test oversquashing in GNNs. We build on the implementation of Karhadkar et al. (2023) which adapts this benchmark to test rewiring. Each graph in the dataset is a path of cliques as shown in Figure 6. Each of the green nodes is assigned a distinct random label which is represented by a

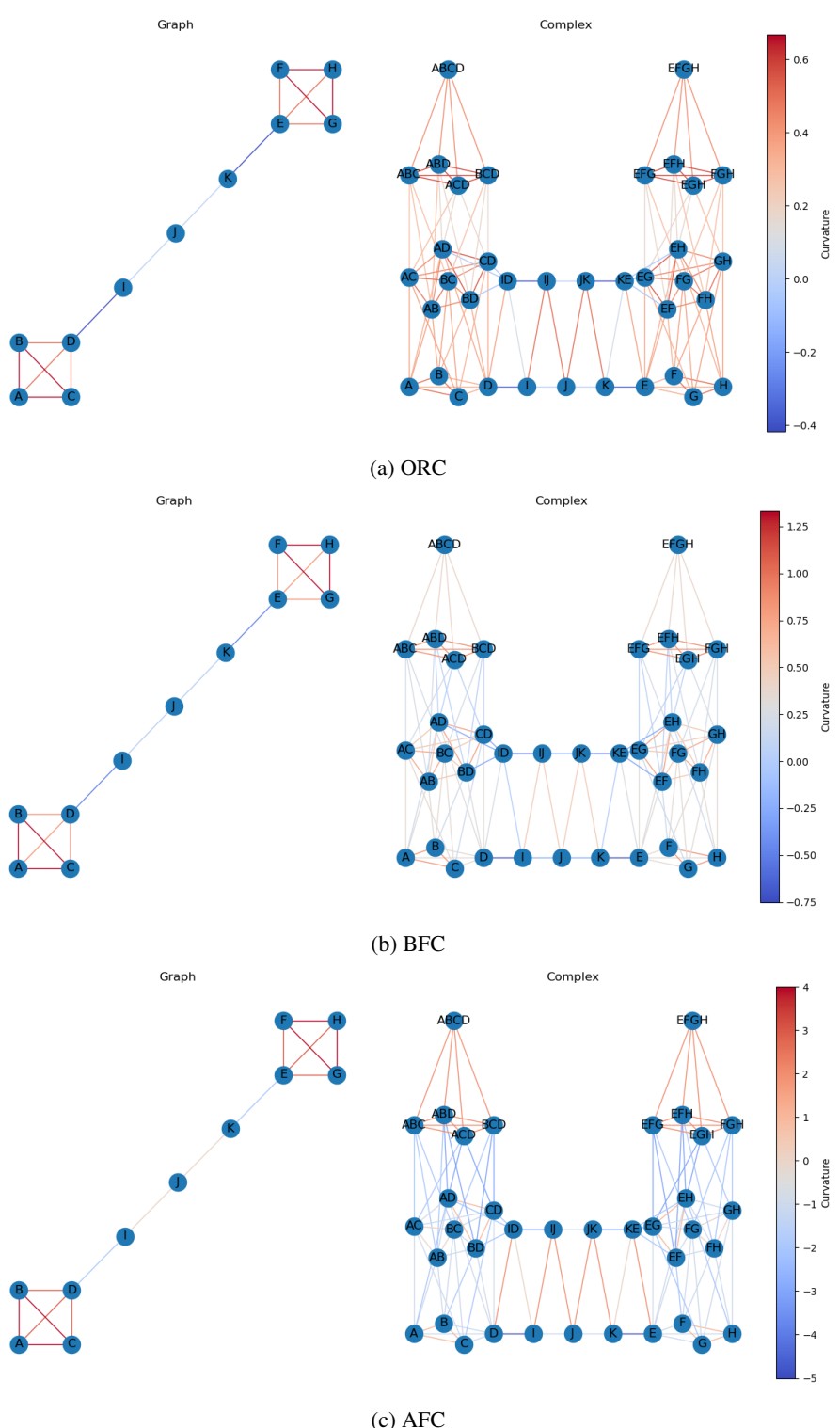

Figure 3: Long dumbbell graph before and after lifting to its clique complex. Edges are colored based on their curvature: Ollivier Ricci curvature (top), balanced Forman curvature (middle), and augmented Forman curvature (bottom). Note that this uses Boundary, Co-boundary, Lower, and Upper relations. These are presented as edges for visual clarity.

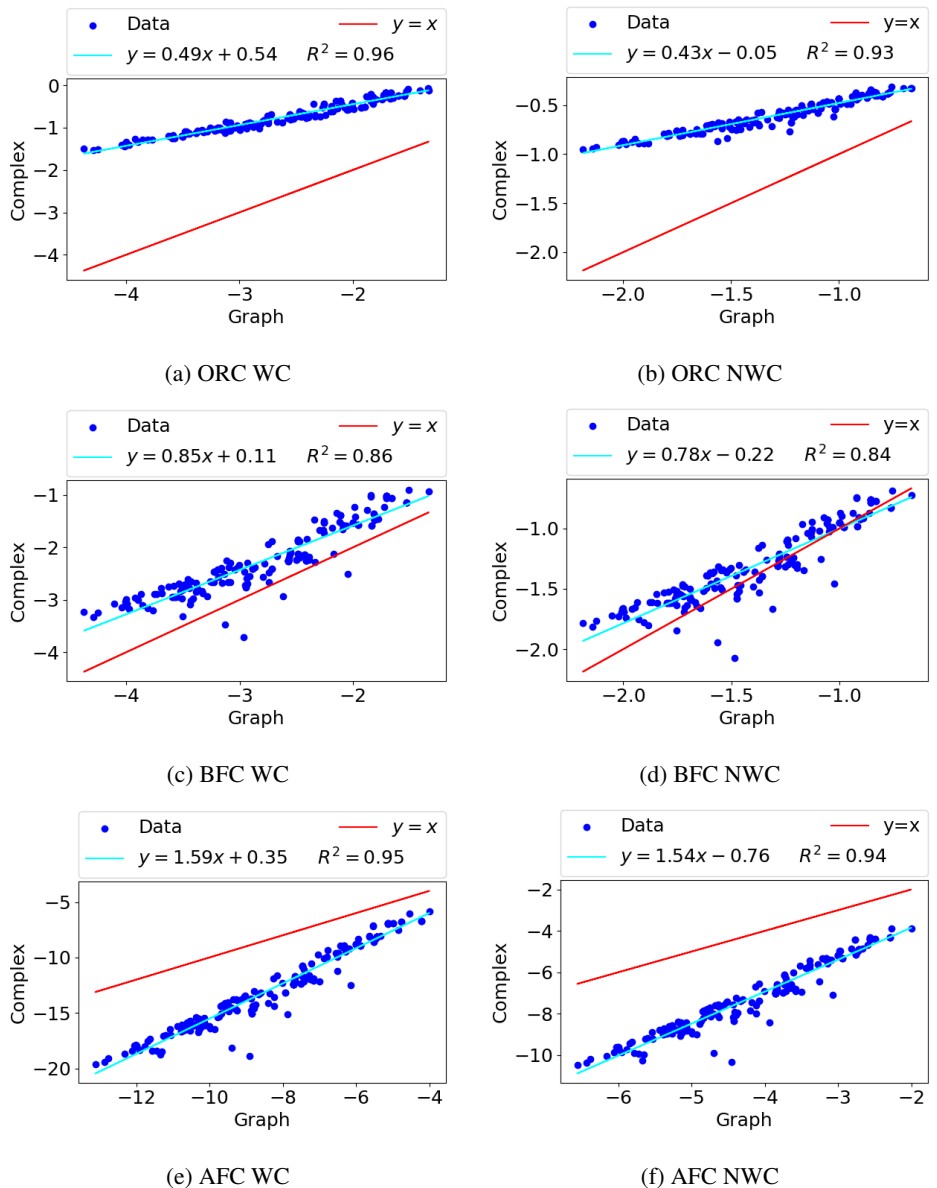

Figure 4: Weighted curvature scatter plots for MUTAG: Ollivier Ricci curvature (top), balanced Forman curvature (middle), and augmented Forman curvature (bottom); betweeness weighted curvature (left) and negative betweenness weighted curvature (right).

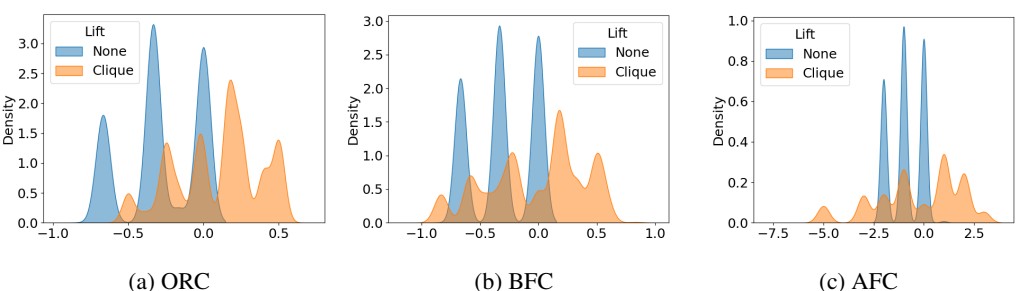

Figure 5: Edge curvature distribution across all graphs in the MUTAG dataset: Ollivier-Ricci curvature (left), Balanced Forman curvature (middle), and Augmented Forman curvature (right).

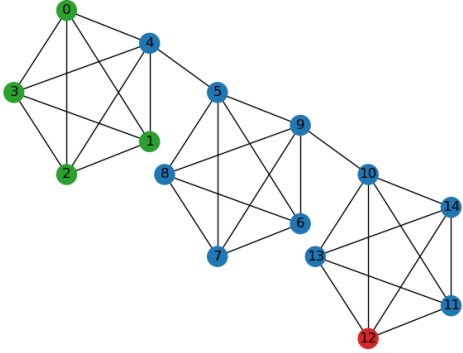

Figure 6: NeighborsMatch Graph with 3 cliques of 5 nodes.

one-hot encoding. The root node, colored red, is also assigned a random label which is encoded using a one-hot encoding. The goal is for the neural network to learn which green node has the same one-hot encoding as the root node based solely on the output of the neural network at the root node.

In this experiment, we choose the number of nodes in each clique to be 5 to avoid excessively large clique complexes when performing graph lifting. Note that we see similar trends as in the RingTransfer experiment. However, this creates a much larger clique complex which may present a greater challenge for rewiring methods. We can see in Figure 7 that the Clique and Ring lifts require more rewiring iterations to achieve the same performance as their graph counterparts. This is likely related to the lifts having many more nodes and edges than the original graph. The original graph has 15 nodes and 32 edges, whereas the clique lift has 77 nodes and 764 edges and the ring lift has 77 nodes and 854 edges.

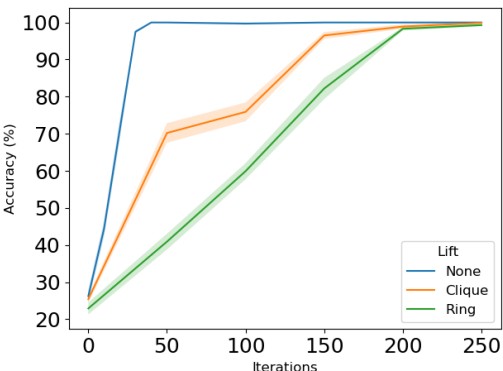

Figure 7: Neighbors Match Rewiring experiment.

We also benchmark trees and trees with cycles attached to the leaves on the NeighborsMatch benchmark (see Figure 8). We note that a tree as an acyclic connected graph has Betti numbers $b_0 = 1$ (number of connected components) and $b_1 = 0$ (number of independent loops or cycles). On the other hand, a tree with $n$ attached cycles has Betti numbers $b_0 = 1$ and $b_1 = n$. As such, the two structures are topologically very distinct when considered as 1-simplicial complexes: the tree is contractible and has a trivial fundamental group, while the tree with attached cycles is not contractible and has a non-trivial fundamental group corresponding to a free group with $n$ generators. However, their relational message passing structures are quite similar, and they demonstrate almost identical prototypical oversquashing trends (see Figure 9).

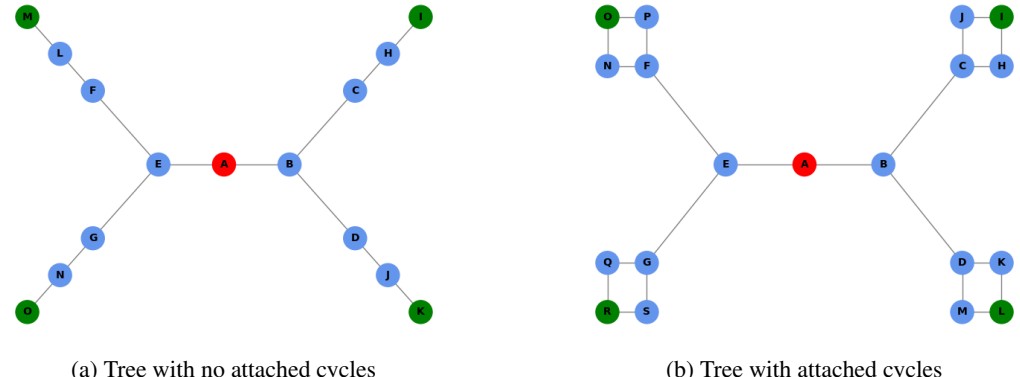

(a) Tree with no attached cycles                    (b) Tree with attached cycles

Figure 8: Tree without and with attached cycles have very distinct topologies.

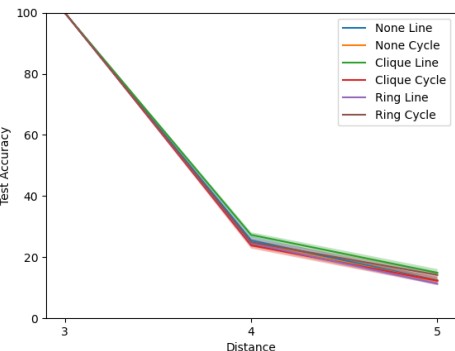

Figure 9: Neighbors Match Rewiring experiment on trees and trees with cycles, with increasing tree depths. "None", "Clique", and "Ring" indicate the graph lifting applied. "Line" and "Cycle" indicate whether lines or cycles are attached to the leaves.

### D.5   REAL-WORLD BENCHMARK: GRAPH REGRESSION (ZINC)

In this section we test our methods on the ZINC dataset. Due to increased computational demands, we do not perform a hyperparameter sweep for this experiment and instead use the hyperparameters from the TUDataset experiments. We also test this on the ring lifting procedure. The ring lift consists of adding cycles in the graph as 2-cells in the complex as in Bodnar et al. (2021a). In this work, we restrict the rings to have size at most 7. Structures like carbon rings are important in molecule datasets and prediction tasks can benefit from encoding larger rings in the lifting procedure. The results are presented in Table 2, where we see a clear improvement from the Ring lift across all models and rewirings tested for the ZINC dataset.

### D.6   REAL-WORLD BENCHMARK: NODE CLASSIFICATION

In this section, we test the lifting and rewiring methods on node classification tasks. The results of this experiment are shown in Table 3. The rewirings considered included FoSR and the pruning algorithm introduced in Appendix D.8. The best results for each dataset and model are highlighted in gold. Notably, GIN and CIN++ benefited from lifting on all datasets tested.

### D.7   ABLATION TESTS

We perform ablation studies to evaluate the effect of increasing the number of layers and rewiring iterations, as well as the effect of increasing the hidden dimensions and rewiring iterations, on the performance of CIN++ on the graph classification MUTAG dataset with FoSR rewiring. The results

| Complex | Rewiring | GIN | R-GIN | SIN | CIN++ |
|---------|----------|-----|-------|-----|-------|
| None | None | $0.405 \pm 0.001$ | $0.368 \pm 0.003$ | $0.370 \pm 0.003$ | $0.339 \pm 0.005$ |
|      | FoSR | $1.056 \pm 0.010$ | $1.485 \pm 1.045$ | $0.377 \pm 0.005$ | $0.374 \pm 0.008$ |
| Clique | None | $0.409 \pm 0.006$ | $0.326 \pm 0.003$ | $0.331 \pm 0.004$ | $0.320 \pm 0.006$ |
|        | FoSR | $0.844 \pm 0.011$ | $0.420 \pm 0.002$ | $0.369 \pm 0.003$ | $0.379 \pm 0.008$ |
| Ring | None | $0.332 \pm 0.006$ | $0.215 \pm 0.003$ | $0.223 \pm 0.004$ | $0.214 \pm 0.003$ |
|      | FoSR | $0.641 \pm 0.011$ | $0.232 \pm 0.002$ | $0.212 \pm 0.003$ | $0.211 \pm 0.002$ |

Table 2: ZINC (MAE). Lower values are better.

are presented in Tables 4 and 5. This analysis complements the experiments in Section 5.2 by testing on real world data. We note that the results of the ablation studies for the MUTAG dataset do not show a clear pattern across all rewiring iterations. However, we broadly see that rewiring too much tends to hurt performance as they begin to oversmooth information. We also generally see that increasing the hidden dimension tends to improve performance. Due to time constraints and limited compute, we could only test one dataset and perform 10 trials for each set of hyperparamters. With more trials, the standard error will decrease and make the patterns more clear. Nguyen et al. (2023) and Fesser & Weber (2024) performed similar ablations and considered 100 trials for each set of hyperparameters. As in the experiments from Section 5 and the results reported in Rusch et al. (2023) and Nguyen et al. (2023), we expect that increasing the hidden dimension will improve classification accuracy. Similarly, increasing layers will initially improve performance until the model becomes too deep and experiences oversmoothing.

### D.8 SIMPLEX PRUNING

In this section we analyze the impact of pruning edges in the influence graph. For the pruning algorithm, we consider removing the edge with the highest balanced Forman curvature. More precisely, we first convert the simplex into the influence graph and collapse multi-edges. Then we compute the balanced Forman curvature of each edge in the graph. Finally, we remove the edge with the highest curvature. This process is repeated 40 times. When removing an edge with higher arity $e = (\sigma, \tau, \delta)$, the entire higher-order edge is removed from the simplex. When an edge $(\sigma, \tau)$ with higher multiplicity is selected, all instances of that edge are removed from the graph.

The results of this experiment are shown in Table 6. In eight out of twelve experiments, the performance improves with some amount of pruning. Notably, GIN and CIN++ attain their highest performance with 40 pruning iterations. This suggests that rewiring algorithms that implement edge pruning and edge addition could be the most promising candidates for simplicial rewiring algorithms. This is in part due to the addition of many nodes and edges in the graph lifting procedure.

## E  ADDITIONAL EXPERIMENTAL DETAILS

For all experiments, we split the dataset randomly using $80\%$ for training, $10\%$ for validation, and $10\%$ for testing. For classificiation, we use the cross entropy loss. Each experiment is run with early stopping using the validation set. Also, if there is no improvement for 10 iterations, the learning rate is decreased. For all rewiring methods, we only consider adding edges and no edge removal. Each rewiring method is run for 40 iterations.

### E.1  REAL-WORLD BENCHMARK: GRAPH CLASSIFICATION ON TUDATASET

To train and evaluate these method on real world data, we use the TUDataset (Morris et al., 2020). These graph classification tasks are commonly used to benchmark message passing neural networks. We have included summary statistics for MUTAG, NCI1, ENZYMES, PROTEINS, and IMDB-BINARY in Table 7.

**(a) GIN**

| Complex | Rewiring | CORNELL | TEXAS | WISCONSIN | CITESEER | CORA |
|---|---|---|---|---|---|---|
| None | FoSR | $41.6 \pm 3.5$ | $57.9 \pm 2.7$ | $49.6 \pm 2.5$ | $69.3 \pm 0.8$ | $82.4 \pm 0.7$ |
| | Prune | $45.8 \pm 4.0$ | $55.3 \pm 4.9$ | $48.1 \pm 3.9$ | $70.6 \pm 0.4^*$ | $84.4 \pm 0.8^*$ |
| | None | $33.7 \pm 3.3$ | $60.0 \pm 5.2$ | $49.2 \pm 2.9$ | $69.9 \pm 0.8$ | $82.7 \pm 0.9$ |
| Clique | FoSR | $40.5 \pm 3.5$ | $64.7 \pm 5.3$ | $56.5 \pm 2.6$ | $68.6 \pm 0.8$ | $84.4 \pm 0.6$ |
| | Prune | $36.8 \pm 2.6$ | $61.6 \pm 3.3$ | $47.3 \pm 3.7$ | $72.0 \pm 0.8^*$ | $82.4 \pm 0.5^*$ |
| | None | $44.7 \pm 3.6$ | $58.9 \pm 2.3$ | $55.8 \pm 2.2$ | $69.0 \pm 0.8$ | $83.2 \pm 0.6$ |
| Ring | FoSR | $45.3 \pm 2.1$ | $61.1 \pm 3.3$ | $55.0 \pm 3.3$ | | |
| | Prune | $47.4 \pm 4.1$ | $65.3 \pm 4.5$ | $54.2 \pm 3.2^*$ | | |
| | None | $46.3 \pm 3.1$ | $58.9 \pm 3.3$ | $54.6 \pm 3.0$ | | |

**(b) RGIN**

| Complex | Rewiring | CORNELL | TEXAS | WISCONSIN | CITESEER | CORA |
|---|---|---|---|---|---|---|
| None | FoSR | $54.2 \pm 3.3$ | $51.1 \pm 3.3$ | $71.2 \pm 2.4$ | $73.2 \pm 0.6$ | $81.9 \pm 0.8$ |
| | Prune | $52.6 \pm 3.5$ | $65.3 \pm 3.2$ | $68.5 \pm 2.1$ | $73.2 \pm 0.6^*$ | $85.6 \pm 0.7^*$ |
| | None | $59.5 \pm 5.0$ | $67.9 \pm 3.2$ | $69.2 \pm 2.8$ | $72.2 \pm 1.0$ | $86.4 \pm 0.7$ |
| Clique | FoSR | $51.1 \pm 3.8$ | $62.1 \pm 3.7$ | $68.5 \pm 3.3$ | $73.4 \pm 1.0$ | $85.7 \pm 0.6$ |
| | Prune | $47.9 \pm 2.3$ | $64.2 \pm 3.6$ | $60.4 \pm 2.0$ | $73.3 \pm 0.6^*$ | $85.9 \pm 0.6^*$ |
| | None | $50.5 \pm 3.5$ | $67.4 \pm 3.4$ | $68.5 \pm 3.8$ | $72.1 \pm 0.9$ | $86.6 \pm 0.6$ |
| Ring | FoSR | $57.4 \pm 2.3$ | $69.5 \pm 3.6$ | $66.5 \pm 2.4$ | | |
| | Prune | $54.2 \pm 4.1$ | $59.5 \pm 4.7$ | $74.6 \pm 2.5^*$ | | |
| | None | $56.3 \pm 2.7$ | $62.1 \pm 4.1$ | $67.7 \pm 2.2$ | | |

**(c) SIN**

| Complex | Rewiring | CORNELL | TEXAS | WISCONSIN | CITESEER | CORA |
|---|---|---|---|---|---|---|
| None | FoSR | $30.5 \pm 3.5$ | $34.7 \pm 4.1$ | $30.8 \pm 3.2$ | $54.3 \pm 1.1$ | $50.6 \pm 2.1$ |
| | Prune | $44.7 \pm 4.2$ | $55.8 \pm 4.9$ | $53.1 \pm 2.9$ | $66.9 \pm 0.6^*$ | $81.7 \pm 0.8^*$ |
| | None | $42.6 \pm 2.9$ | $52.6 \pm 4.3$ | $50.4 \pm 2.8$ | $66.9 \pm 0.8$ | $81.6 \pm 0.8$ |
| Clique | FoSR | $36.3 \pm 2.7$ | $49.5 \pm 3.0$ | $47.3 \pm 3.0$ | $60.2 \pm 1.2$ | $68.2 \pm 0.9$ |
| | Prune | $40.5 \pm 2.9$ | $56.3 \pm 3.0$ | $51.5 \pm 3.1$ | $64.8 \pm 0.9^*$ | $75.6 \pm 1.2^*$ |
| | None | $36.8 \pm 2.9$ | $55.8 \pm 2.7$ | $45.8 \pm 3.8$ | $63.2 \pm 0.8$ | $77.7 \pm 0.7$ |
| Ring | FoSR | $33.7 \pm 4.1$ | $51.1 \pm 4.4$ | $53.1 \pm 3.5$ | | |
| | Prune | $42.1 \pm 3.2$ | $58.4 \pm 2.8$ | $55.0 \pm 3.3^*$ | | |
| | None | $38.9 \pm 3.3$ | $54.7 \pm 2.2$ | $61.2 \pm 3.5$ | | |

**(d) CIN++**

| Complex | Rewiring | CORNELL | TEXAS | WISCONSIN | CITESEER | CORA |
|---|---|---|---|---|---|---|
| None | FoSR | $41.6 \pm 2.0$ | $55.3 \pm 2.7$ | $51.9 \pm 4.1$ | $44.9 \pm 1.1$ | $64.0 \pm 1.8$ |
| | Prune | $42.6 \pm 3.5$ | $49.5 \pm 4.7$ | $61.2 \pm 2.3$ | $45.2 \pm 1.4^*$ | $55.7 \pm 1.8^*$ |
| | None | $43.2 \pm 4.4$ | $59.5 \pm 2.6$ | $54.2 \pm 3.8$ | $44.7 \pm 1.7$ | $63.6 \pm 2.0$ |
| Clique | FoSR | $47.9 \pm 4.5$ | $66.3 \pm 3.7$ | $64.6 \pm 1.6$ | $64.4 \pm 1.0$ | $78.6 \pm 0.8$ |
| | Prune | $52.6 \pm 3.5$ | $57.9 \pm 3.2$ | $63.8 \pm 3.5$ | $60.5 \pm 0.7^*$ | $77.0 \pm 1.0^*$ |
| | None | $50.5 \pm 2.7$ | $65.3 \pm 2.6$ | $66.2 \pm 2.8$ | $63.2 \pm 0.9$ | $79.4 \pm 0.7$ |
| Ring | FoSR | $42.1 \pm 4.1$ | $65.3 \pm 3.3$ | $62.3 \pm 1.7$ | | |
| | Prune | $47.9 \pm 3.7$ | $63.2 \pm 4.4$ | $56.9 \pm 2.5^*$ | | |
| | None | $54.2 \pm 4.2$ | $64.2 \pm 4.1$ | $63.1 \pm 3.0$ | | |

Table 3: Node classification experiments. Experiments with $*$ used 1d curvature for pruning.

In Tables 8, 9 and 10, we provide all the results for Table 1 from Section 5. We note that in Table 1, we only report the Lift=None results for FoSR, and not AFR4 or SDRF, for fair comparison with Lift=Clique and Lift=Ring, where we only report the results for FoSR. We note that for $24/32 = 75\%$ of the results on ENZYME, MUTAG, NCI1, and PROTEINS, the best performing rewiring algorithms for Lift=None and Lift=Clique agree. Similarly, the best rewiring algorithms for Lift=None and Lift=Ring agree in $17/32 = 53\%$ of the experiments. In Figure 10, we compare model performance on graphs and the corresponding clique complexes, and note how graph, relational graph, and topological models all respond similarly to relational rewiring.

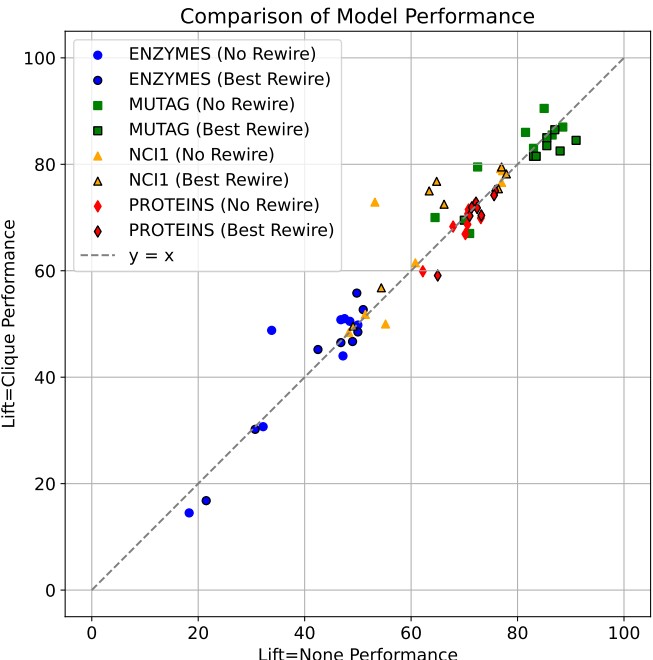

Figure 10: Comparison of model performance between graph representations (Lift=None) and corresponding clique complex representations (Lift=Clique). Each point corresponds to a specific model on a given dataset, comparing the performance with and without rewiring.

| Lift | Rewire Iterations Layers | 0 | 10 | 20 | 40 |
|------|--------------------------|---|----|----|----|
| None | 2 | $86.0 \pm 2.9$ | $85.0 \pm 2.4$ | $75.5 \pm 1.7$ | $85.5 \pm 3.1$ |
|      | 4 | $87.5 \pm 2.3$ | $81.5 \pm 2.6$ | $81.0 \pm 3.2$ | $77.5 \pm 2.9$ |
|      | 6 | $89.0 \pm 2.7$ | $82.5 \pm 2.8$ | $78.5 \pm 3.7$ | $80.0 \pm 2.8$ |
| Clique | 2 | $85.5 \pm 2.2$ | $84.0 \pm 1.8$ | $81.0 \pm 2.2$ | $83.5 \pm 3.0$ |
|      | 4 | $86.0 \pm 1.9$ | $78.5 \pm 3.6$ | $83.5 \pm 2.9$ | $79.0 \pm 2.1$ |
|      | 6 | $84.5 \pm 2.7$ | $80.5 \pm 2.0$ | $82.5 \pm 3.1$ | $80.5 \pm 2.3$ |
| Ring | 2 | $84.5 \pm 1.9$ | $86.5 \pm 2.4$ | $86.5 \pm 2.6$ | $87.0 \pm 2.5$ |
|      | 4 | $86.5 \pm 2.6$ | $84.5 \pm 2.4$ | $80.5 \pm 3.0$ | $86.5 \pm 1.7$ |
|      | 6 | $82.0 \pm 1.9$ | $86.0 \pm 1.8$ | $86.0 \pm 2.7$ | $88.0 \pm 2.5$ |

Table 4: Impact of increasing the number of layers and rewiring iterations for FoSR on the MUTAG dataset with CIN++.

| Lift | Rewire Iterations Hidden Dimension | 0 | 10 | 20 | 40 |
|------|------------------------------------|---|----|----|----|
| None | 16 | $81.5 \pm 2.5$ | $77.5 \pm 3.0$ | $78.5 \pm 3.7$ | $74.0 \pm 1.8$ |
|      | 32 | $83.0 \pm 2.9$ | $77.5 \pm 2.8$ | $84.5 \pm 2.6$ | $82.0 \pm 4.2$ |
|      | 64 | $85.0 \pm 1.8$ | $82.0 \pm 2.1$ | $83.0 \pm 1.5$ | $82.5 \pm 2.4$ |
| Clique | 16 | $87.5 \pm 3.0$ | $79.5 \pm 3.6$ | $77.0 \pm 2.6$ | $79.0 \pm 1.6$ |
|      | 32 | $83.5 \pm 2.2$ | $81.0 \pm 2.2$ | $80.5 \pm 4.1$ | $78.5 \pm 3.4$ |
|      | 64 | $89.0 \pm 2.6$ | $81.0 \pm 2.8$ | $75.5 \pm 3.8$ | $83.5 \pm 2.8$ |
| Ring | 16 | $87.0 \pm 1.7$ | $89.0 \pm 2.4$ | $85.0 \pm 1.5$ | $85.0 \pm 1.1$ |
|      | 32 | $88.5 \pm 2.8$ | $83.5 \pm 1.8$ | $89.0 \pm 2.2$ | $83.5 \pm 1.3$ |
|      | 64 | $85.5 \pm 3.0$ | $85.0 \pm 3.2$ | $86.0 \pm 1.8$ | $84.5 \pm 1.9$ |

Table 5: Impact of increasing the hidden dimensions and rewiring iterations for FoSR on the MU-TAG dataset with CIN++.

### E.2 SYNTHETIC BENCHMARK: RINGTRANSFER

The RingTransfer benchmark is a graph transfer task introduced by Bodnar et al. (2021a). Each graph in the dataset is a cycle graph of size $2k$ for some $k \geq 1$. A node $i$ is randomly selected as the *root node* and the node $j = i + k( \mod 2k$ on the opposite side of the cycle is designated as the feature node. All features are initialized to zero except for the feature node which contains a one-hot label among the 5 classes. The task is to then predict the label by reading the feature stored in the root node. This is shown for a cycle graph with 10 nodes in Figure 11, where the red node is the root node and the green node is the opposite node. Note that the size of the ring determines the number of layers necessary for a model to complete the task successfully. By increasing the size of the ring, we can test how depth and the task distance impact training performance. When increasing the ring size, the model depth is set to $2k + 1$ so that task information remains within the model's receptive field.

## F IMPLEMENTATION DETAILS

### F.1 MODEL DETAILS

Each model consists of multiple graph-convolution layers. For more details about the convolutional layers, refer to Appendix F.5. The hyperparameters are presented in Table 12. For the real world experiments, each model has a neural network head consisting of a pooling operation followed by a two layer neural network with ReLU and dropout. The only exception to this is the SGC model which uses pooling and a single linear layer for the head. For the synthetic graph transfer experiments, there is no neural network head. Instead, the neural network just returns the feature of the root node at the final layer.

| Complex | Prune Iterations Model | 0 | 10 | 20 | 40 |
|---------|------------------------|---|----|----|----|
| None | GIN | $83.0 \pm 3.1$ | $87.5 \pm 3.1$ | $89.0 \pm 3.1$ | $86.0 \pm 2.1$ |
| | RGIN | $81.5 \pm 1.7$ | $84.0 \pm 2.8$ | $91.0 \pm 2.2$ | $86.0 \pm 1.2$ |
| | SIN | $88.5 \pm 3.0$ | $83.0 \pm 2.5$ | $86.0 \pm 2.1$ | $86.5 \pm 2.5$ |
| | CIN++ | $85.0 \pm 3.4$ | $91.5 \pm 1.8$ | $88.0 \pm 2.0$ | $84.5 \pm 2.5$ |
| Clique | GIN | $83.0 \pm 2.8$ | $83.5 \pm 1.7$ | $82.5 \pm 3.1$ | $90.5 \pm 1.4$ |
| | RGIN | $86.0 \pm 2.3$ | $81.0 \pm 2.4$ | $82.5 \pm 2.5$ | $85.0 \pm 1.7$ |
| | SIN | $87.0 \pm 3.2$ | $79.5 \pm 2.2$ | $83.5 \pm 2.2$ | $84.5 \pm 3.4$ |
| | CIN++ | $90.5 \pm 2.2$ | $84.5 \pm 3.7$ | $78.5 \pm 3.2$ | $91.0 \pm 1.8$ |
| Ring | GIN | $90.0 \pm 2.2$ | $87.5 \pm 3.2$ | $88.0 \pm 1.7$ | $85.5 \pm 1.6$ |
| | RGIN | $87.0 \pm 2.3$ | $87.5 \pm 2.6$ | $86.5 \pm 2.8$ | $87.0 \pm 2.4$ |
| | SIN | $87.5 \pm 1.5$ | $87.0 \pm 2.9$ | $84.0 \pm 2.4$ | $88.0 \pm 2.3$ |
| | CIN++ | $87.5 \pm 2.6$ | $80.5 \pm 3.3$ | $88.0 \pm 2.0$ | $81.5 \pm 1.5$ |

Table 6: MUTAG pruning experiment. Removing edges with highest balanced Forman Curvature.

| Dataset | # Graphs | Classes | Avg. Nodes | Avg. Edges | Features |
|---------|----------|---------|------------|------------|----------|
| MUTAG | 188 | 2 | 17.93 | 19.79 | 7 |
| NCI1 | 4110 | 2 | 29.87 | 32.30 | 37 |
| ENZYMES | 600 | 6 | 32.63 | 62.14 | 3 |
| PROTEINS | 1113 | 2 | 39.06 | 72.82 | 3 |
| IMDB-BINARY | 1000 | 2 | 19.77 | 96.53 | 0 |

Table 7: TUDataset Statistics.

## F.2 HARDWARE

Experiments were run on a system with Intel(R) Xeon(R) Gold 6152 CPUs @ 2.10GHz and NVIDIA GeForce RTX 2080 Ti GPUs.

## F.3 LIBRARIES

Code Use: Rewiring algorithms were adapted from the works of Karhadkar et al. (2023) and Fesser & Weber (2023). We used PyTorch Geometric Benchmarks for neural network hyperparameters. We build on the work of Giusti et al. (2024) for clique graph lifting and ring transfer. The real world graph datasets were sourced from the work of Morris et al. (2020). The SGC model was adapted from the work of Wu et al. (2019). For curvature implementations, we adapt code from Nguyen et al. (2023) for Ollivier-Ricci and Fesser & Weber (2023) for BFC and AFC.

Python libraries: Torch Geometric (Fey & Lenssen, 2019). Torch (Paszke et al., 2019). Scikit-Learn (Pedregosa et al., 2011). Ray (Moritz et al., 2018). Ray Tune (Liaw et al., 2018). Seaborn (Waskom,

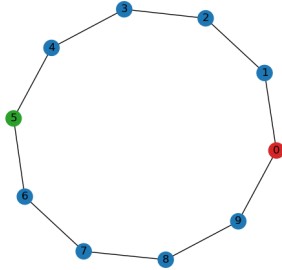

Figure 11: Graph for the RingTransfer experiment.

(a) **SGC**

| Lift | Rewiring | ENZYMES | IMDB-B | MUTAG | NCI1 | PROTEINS |
|---|---|---|---|---|---|---|
| None | None | $18.3 \pm 1.2$ | $49.5 \pm 1.5$ | $64.5 \pm 5.8$ | $55.2 \pm 1.0$ | $62.2 \pm 1.4$ |
| | AFR4 | $17.0 \pm 1.7$ | $48.6 \pm 2.3$ | $65.5 \pm 3.5$ | $54.4 \pm 0.7$ | $65.0 \pm 1.5$ |
| | FoSR | $18.2 \pm 1.7$ | $50.0 \pm 1.8$ | $52.5 \pm 6.6$ | $49.3 \pm 0.8$ | $64.8 \pm 1.1$ |
| | SDRF | $21.5 \pm 1.6$ | $49.7 \pm 1.7$ | $70.0 \pm 2.6$ | $52.4 \pm 0.7$ | $59.9 \pm 1.4$ |
| Clique | None | $14.5 \pm 1.4$ | $48.7 \pm 2.2$ | $70.0 \pm 3.3$ | $50.0 \pm 1.3$ | $59.9 \pm 1.8$ |
| | AFR4 | $16.3 \pm 1.0$ | $> 1$ day | $62.5 \pm 3.0$ | $56.8 \pm 0.8$ | $59.1 \pm 1.4$ |
| | FoSR | $16.0 \pm 1.4$ | $47.8 \pm 1.6$ | $60.5 \pm 5.8$ | $50.0 \pm 0.6$ | $50.0 \pm 4.3$ |
| | SDRF | $16.8 \pm 0.9$ | $> 12$ hours | $69.5 \pm 2.6$ | $50.8 \pm 0.7$ | $54.9 \pm 3.2$ |
| Ring | None | $16.5 \pm 1.6$ | $50.1 \pm 1.9$ | $65.5 \pm 3.6$ | $51.5 \pm 1.2$ | $44.8 \pm 2.3$ |
| | AFR4 | $19.3 \pm 1.2$ | $> 1$ day | $72.5 \pm 3.2$ | $50.6 \pm 0.6$ | $> 1$ day |
| | FoSR | $18.7 \pm 1.6$ | $49.9 \pm 1.9$ | $75.0 \pm 5.7$ | $50.3 \pm 0.8$ | $47.6 \pm 3.0$ |
| | SDRF | $19.0 \pm 1.3$ | $> 12$ hours | $70.0 \pm 2.7$ | $51.4 \pm 0.4$ | $49.3 \pm 3.6$ |

(b) **GCN**

| Lift | Rewiring | ENZYMES | IMDB-B | MUTAG | NCI1 | PROTEINS |
|---|---|---|---|---|---|---|
| None | None | $32.2 \pm 2.0$ | $49.1 \pm 1.4$ | $71.0 \pm 3.8$ | $48.3 \pm 0.6$ | $73.1 \pm 1.2$ |
| | AFR4 | $30.5 \pm 2.5$ | $48.6 \pm 1.0$ | $69.0 \pm 2.7$ | $48.4 \pm 0.6$ | $72.6 \pm 1.5$ |
| | FoSR | $27.2 \pm 2.5$ | $47.9 \pm 1.0$ | $83.0 \pm 1.5$ | $49.1 \pm 0.8$ | $75.8 \pm 1.7$ |
| | SDRF | $30.7 \pm 1.5$ | $46.7 \pm 1.2$ | $72.5 \pm 3.5$ | $48.9 \pm 0.5$ | $74.5 \pm 1.5$ |
| Clique | None | $30.7 \pm 1.2$ | $64.0 \pm 3.1$ | $67.0 \pm 3.5$ | $48.4 \pm 0.4$ | $69.9 \pm 0.6$ |
| | AFR4 | $28.2 \pm 2.2$ | $> 1$ day | $66.5 \pm 3.0$ | $48.3 \pm 0.6$ | $72.5 \pm 0.9$ |
| | FoSR | $26.0 \pm 2.1$ | $65.5 \pm 3.1$ | $81.5 \pm 2.9$ | $48.8 \pm 1.1$ | $75.0 \pm 1.4$ |
| | SDRF | $30.2 \pm 2.4$ | $> 12$ hours | $72.0 \pm 2.6$ | $49.6 \pm 0.6$ | $72.1 \pm 1.2$ |
| Ring | None | $34.8 \pm 1.3$ | $46.9 \pm 1.4$ | $72.0 \pm 2.7$ | $49.3 \pm 0.9$ | $72.2 \pm 1.3$ |
| | AFR4 | $29.0 \pm 1.5$ | $> 1$ day | $72.5 \pm 2.3$ | $49.1 \pm 0.6$ | $> 1$ day |
| | FoSR | $28.8 \pm 1.6$ | $48.0 \pm 1.2$ | $77.5 \pm 2.4$ | $47.9 \pm 0.6$ | $71.3 \pm 1.6$ |
| | SDRF | $32.0 \pm 1.4$ | $> 12$ hours | $67.0 \pm 2.8$ | $49.4 \pm 0.6$ | $72.7 \pm 1.2$ |

(c) **GIN**

| Lift | Rewiring | ENZYMES | IMDB-B | MUTAG | NCI1 | PROTEINS |
|---|---|---|---|---|---|---|
| None | None | $47.2 \pm 1.9$ | $71.7 \pm 1.5$ | $83.0 \pm 3.1$ | $77.2 \pm 0.5$ | $70.6 \pm 1.4$ |
| | AFR4 | $50.0 \pm 2.7$ | $69.7 \pm 1.8$ | $88.0 \pm 2.4$ | $77.9 \pm 0.5$ | $70.1 \pm 0.9$ |
| | FoSR | $33.3 \pm 1.7$ | $67.1 \pm 1.5$ | $75.5 \pm 1.7$ | $64.8 \pm 0.9$ | $72.2 \pm 0.8$ |
| | SDRF | $45.5 \pm 2.1$ | $65.8 \pm 2.0$ | $84.5 \pm 2.8$ | $76.3 \pm 1.0$ | $72.0 \pm 1.5$ |
| Clique | None | $44.0 \pm 1.7$ | $69.1 \pm 1.2$ | $83.0 \pm 2.8$ | $78.8 \pm 0.7$ | $68.7 \pm 1.4$ |
| | AFR4 | $48.5 \pm 2.2$ | $> 1$ day | $82.5 \pm 2.6$ | $78.2 \pm 0.6$ | $67.6 \pm 0.8$ |
| | FoSR | $39.3 \pm 1.4$ | $70.8 \pm 1.1$ | $79.0 \pm 2.6$ | $63.5 \pm 0.7$ | $72.8 \pm 1.2$ |
| | SDRF | $41.7 \pm 2.2$ | $> 12$ hours | $73.5 \pm 3.4$ | $76.9 \pm 0.7$ | $69.2 \pm 0.8$ |
| Ring | None | $46.7 \pm 2.4$ | $70.1 \pm 1.7$ | $88.0 \pm 2.1$ | $78.9 \pm 0.6$ | $69.8 \pm 1.4$ |
| | AFR4 | $47.0 \pm 1.6$ | $> 1$ day | $89.0 \pm 1.9$ | $77.5 \pm 0.9$ | $> 1$ day |
| | FoSR | $37.5 \pm 1.5$ | $73.1 \pm 1.1$ | $84.0 \pm 2.1$ | $65.6 \pm 0.8$ | $72.0 \pm 1.3$ |
| | SDRF | $41.2 \pm 1.6$ | $> 12$ hours | $79.0 \pm 2.7$ | $75.9 \pm 0.9$ | $72.1 \pm 0.9$ |

Table 8: Baseline and rewiring results for SGC, GCN, and GIN. Numbers highlighted in blue correspond to no rewiring and numbers highlighted in red are the best among the rewiring methods.

(a) **RGCN**

| Lift | Rewiring | ENZYMES | IMDB-B | MUTAG | NCI1 | PROTEINS |
|---|---|---|---|---|---|---|
| None | None | $33.8 \pm 1.6$ | $47.6 \pm 1.4$ | $72.5 \pm 2.5$ | $53.2 \pm 0.7$ | $71.9 \pm 1.6$ |
|  | AFR4 | $36.2 \pm 2.3$ | $48.1 \pm 1.0$ | $69.5 \pm 3.4$ | $54.3 \pm 1.1$ | $72.8 \pm 1.2$ |
|  | FoSR | $39.3 \pm 1.6$ | $68.0 \pm 1.3$ | $83.5 \pm 1.8$ | $62.4 \pm 0.5$ | $71.2 \pm 1.8$ |
|  | SDRF | $42.5 \pm 1.3$ | $59.1 \pm 2.2$ | $76.0 \pm 1.9$ | $63.4 \pm 1.1$ | $75.6 \pm 1.2$ |
| Clique | None | $48.8 \pm 1.2$ | $71.0 \pm 1.0$ | $79.5 \pm 1.7$ | $72.9 \pm 0.8$ | $72.4 \pm 1.6$ |
|  | AFR4 | $42.3 \pm 2.4$ | $> 1$ day | $78.0 \pm 2.1$ | $75.0 \pm 0.9$ | $74.2 \pm 1.2$ |
|  | FoSR | $39.3 \pm 2.5$ | $69.7 \pm 1.5$ | $79.0 \pm 3.2$ | $64.5 \pm 0.6$ | $70.9 \pm 1.4$ |
|  | SDRF | $45.2 \pm 1.5$ | $> 12$ hours | $81.5 \pm 3.8$ | $68.5 \pm 0.6$ | $70.6 \pm 1.6$ |
| Ring | None | $35.2 \pm 1.7$ | $71.1 \pm 1.4$ | $83.5 \pm 2.7$ | $73.9 \pm 0.5$ | $70.7 \pm 1.6$ |
|  | AFR4 | $34.8 \pm 2.1$ | $> 1$ day | $80.5 \pm 2.9$ | $73.5 \pm 0.5$ | $> 1$ day |
|  | FoSR | $38.5 \pm 1.5$ | $70.0 \pm 1.6$ | $84.0 \pm 2.1$ | $64.2 \pm 0.8$ | $71.3 \pm 1.2$ |
|  | SDRF | $45.7 \pm 1.5$ | $> 12$ hours | $79.5 \pm 3.7$ | $68.3 \pm 0.6$ | $68.8 \pm 1.1$ |

(b) **RGIN**

| Lift | Rewiring | ENZYMES | IMDB-B | MUTAG | NCI1 | PROTEINS |
|---|---|---|---|---|---|---|
| None | None | $46.8 \pm 1.8$ | $69.6 \pm 1.6$ | $81.5 \pm 1.7$ | $76.8 \pm 1.1$ | $70.8 \pm 1.2$ |
|  | AFR4 | $49.8 \pm 2.0$ | $73.2 \pm 1.4$ | $85.5 \pm 2.0$ | $77.0 \pm 0.7$ | $71.1 \pm 1.3$ |
|  | FoSR | $48.2 \pm 1.4$ | $48.9 \pm 2.9$ | $85.5 \pm 2.8$ | $55.1 \pm 2.6$ | $72.4 \pm 1.4$ |
|  | SDRF | $49.7 \pm 2.1$ | $50.4 \pm 3.2$ | $85.0 \pm 1.7$ | $52.8 \pm 2.7$ | $70.9 \pm 1.2$ |
| Clique | None | $50.8 \pm 1.5$ | $71.6 \pm 0.9$ | $86.0 \pm 2.3$ | $79.2 \pm 0.6$ | $71.5 \pm 1.5$ |
|  | AFR4 | $55.8 \pm 2.5$ | $> 1$ day | $85.0 \pm 2.4$ | $79.5 \pm 0.4$ | $71.0 \pm 1.1$ |
|  | FoSR | $46.2 \pm 1.4$ | $69.0 \pm 1.4$ | $79.0 \pm 2.4$ | $72.7 \pm 0.6$ | $71.8 \pm 1.7$ |
|  | SDRF | $45.5 \pm 1.6$ | $> 12$ hours | $83.0 \pm 3.2$ | $76.4 \pm 0.9$ | $69.1 \pm 2.4$ |
| Ring | None | $45.3 \pm 1.3$ | $68.6 \pm 1.2$ | $87.0 \pm 2.9$ | $78.4 \pm 0.7$ | $68.8 \pm 1.5$ |
|  | AFR4 | $49.2 \pm 1.5$ | $> 1$ day | $87.5 \pm 2.4$ | $79.8 \pm 0.7$ | $> 1$ day |
|  | FoSR | $47.3 \pm 1.6$ | $67.2 \pm 1.8$ | $82.5 \pm 4.2$ | $72.9 \pm 0.7$ | $71.3 \pm 1.5$ |
|  | SDRF | $47.2 \pm 2.6$ | $> 12$ hours | $81.0 \pm 3.3$ | $76.4 \pm 0.5$ | $70.0 \pm 0.9$ |

Table 9: Baseline and rewiring results for RGCN and RGIN. Numbers highlighted in blue correspond to no rewiring and numbers highlighted in red are the best among the rewiring methods.

2021). Matplotlib (Hunter, 2007). Networkx (Hagberg et al., 2008). Pandas pandas development team (2020). Numpy Harris et al. (2020). Numba (Lam et al., 2015). Gudhi (The GUDHI Project, 2020). POT (Flamary et al., 2021).

### F.4 GRAPH LIFTING

We lift graphs to clique and ring complexes for relational message passing using Algorithm 3. We refer to this graph lifting as the *Clique* and *Ring* lifts. Note that when representing a graph as a complex (e.g., max_dim = 0), we simply add the ↑ relation to the edges. We refer to this graph lifting as the *None* lift. The resulting complex has trivial cell intersections and does not contain cell unions. This is outlined in Algorithm 2. We note that whereas clique lifts introduce higher-order simplices by introducing simplices for cliques in the graph, ring lifts instead promote cycles to 2-dimensional cells.

### F.5 LAYERS

We review the implementation of the layers and update rules used in the models considered in this work. In the following, $\mathcal{B}(\cdot)$, $\mathcal{C}(\cdot)$, $\mathcal{N}_\uparrow(\cdot)$, and $\mathcal{N}_\downarrow(\cdot)$ follow the definitions in Definition 2.3. Given a relational structure $\mathcal{R} = (\mathcal{S}, R_1, \dots, R_k)$, the notation $r \in \mathcal{R}$ refers to iterating over the relation indices $r = 1, 2, \dots, k$. For a binary relation $R \subseteq \mathcal{S} \times \mathcal{S}$, $\mathcal{R}(\cdot)$ denotes the incoming neighborhood of an entity, i.e., for any $\sigma, \tau \in \mathcal{S}$, if $\tau \in \mathcal{N}_R(\sigma)$, then $(\sigma, \tau) \in R$.

**SGC** Following Wu et al. (2019), the SGC update rule is:

$$\mathbf{h}_\sigma^{(t+1)} = \sum_{r \in \mathcal{R}} \sum_{\tau \in \mathcal{N}_{R_r}(\sigma)} \mathbf{h}_\tau^{(t)},$$

where no learnable weights nor non-linearities are used, and signals are simply aggregated over neighborhoods.

(a) **SIN**

| Lift | Rewiring | ENZYMES | IMDB-B | MUTAG | NCI1 | PROTEINS |
|---|---|---|---|---|---|---|
| None | None | $47.5 \pm 2.3$ | $70.0 \pm 1.4$ | $88.5 \pm 3.0$ | $77.0 \pm 0.6$ | $70.2 \pm 1.3$ |
| | AFR4 | $44.0 \pm 2.5$ | $71.0 \pm 1.0$ | $85.5 \pm 1.7$ | $76.4 \pm 0.4$ | $69.6 \pm 1.3$ |
| | FoSR | $45.7 \pm 2.6$ | $63.0 \pm 2.7$ | $85.5 \pm 2.8$ | $61.3 \pm 2.4$ | $73.2 \pm 1.5$ |
| | SDRF | $46.8 \pm 2.1$ | $62.8 \pm 3.3$ | $80.0 \pm 2.1$ | $54.7 \pm 3.5$ | $70.2 \pm 1.8$ |
| Clique | None | $51.0 \pm 2.4$ | $53.0 \pm 1.9$ | $87.0 \pm 3.2$ | $76.6 \pm 1.3$ | $66.9 \pm 1.3$ |
| | AFR4 | $46.5 \pm 1.2$ | $> 1$ day | $83.5 \pm 1.7$ | $75.4 \pm 0.7$ | $69.6 \pm 1.0$ |
| | FoSR | $38.2 \pm 2.2$ | $64.0 \pm 2.3$ | $83.5 \pm 2.7$ | $65.2 \pm 0.7$ | $70.4 \pm 1.2$ |
| | SDRF | $44.8 \pm 1.9$ | $> 12$ hours | $80.0 \pm 3.5$ | $73.9 \pm 0.4$ | $68.7 \pm 1.2$ |
| Ring | None | $40.3 \pm 2.2$ | $50.6 \pm 1.9$ | $85.0 \pm 2.1$ | $80.0 \pm 0.8$ | $70.6 \pm 1.1$ |
| | AFR4 | $48.0 \pm 2.0$ | $> 1$ day | $88.5 \pm 2.5$ | $79.1 \pm 0.9$ | $> 1$ day |
| | FoSR | $39.8 \pm 1.6$ | $60.9 \pm 2.1$ | $86.0 \pm 2.4$ | $71.2 \pm 0.6$ | $72.1 \pm 0.7$ |
| | SDRF | $39.2 \pm 1.4$ | $> 12$ hours | $80.5 \pm 2.5$ | $75.6 \pm 0.5$ | $67.5 \pm 1.1$ |

(b) **CIN**

| Lift | Rewiring | ENZYMES | IMDB-B | MUTAG | NCI1 | PROTEINS |
|---|---|---|---|---|---|---|
| None | None | $50.0 \pm 1.9$ | $58.1 \pm 4.0$ | $86.5 \pm 1.8$ | $51.4 \pm 2.5$ | $70.7 \pm 1.0$ |
| | AFR4 | $48.2 \pm 1.9$ | $58.8 \pm 4.3$ | $87.0 \pm 2.4$ | $53.3 \pm 2.6$ | $71.0 \pm 1.4$ |
| | FoSR | $49.0 \pm 2.0$ | $58.4 \pm 2.7$ | $81.0 \pm 1.8$ | $66.2 \pm 2.0$ | $70.9 \pm 1.3$ |
| | SDRF | $44.0 \pm 1.8$ | $55.3 \pm 2.5$ | $82.5 \pm 3.0$ | $59.0 \pm 4.3$ | $69.5 \pm 1.5$ |
| Clique | None | $49.8 \pm 1.9$ | $52.6 \pm 2.4$ | $85.5 \pm 2.8$ | $51.8 \pm 2.3$ | $70.7 \pm 1.2$ |
| | AFR4 | $46.7 \pm 1.3$ | $> 1$ day | $86.5 \pm 2.6$ | $49.0 \pm 0.5$ | $69.2 \pm 1.3$ |
| | FoSR | $37.2 \pm 1.8$ | $68.1 \pm 1.6$ | $82.5 \pm 3.0$ | $66.9 \pm 0.9$ | $70.3 \pm 0.8$ |
| | SDRF | $42.0 \pm 2.3$ | $> 12$ hours | $78.5 \pm 4.1$ | $72.5 \pm 0.8$ | $67.9 \pm 1.8$ |
| Ring | None | $47.5 \pm 2.0$ | $48.6 \pm 1.6$ | $93.5 \pm 2.1$ | $51.6 \pm 3.2$ | $68.7 \pm 1.4$ |
| | AFR4 | $49.5 \pm 2.0$ | $> 1$ day | $92.5 \pm 1.5$ | $58.2 \pm 4.2$ | $> 1$ day |
| | FoSR | $37.7 \pm 2.1$ | $66.1 \pm 2.0$ | $95.0 \pm 1.3$ | $70.4 \pm 0.8$ | $68.5 \pm 1.6$ |
| | SDRF | $38.2 \pm 2.2$ | $> 12$ hours | $88.0 \pm 2.4$ | $76.5 \pm 0.5$ | $65.6 \pm 1.3$ |

(c) **CIN++**

| Lift | Rewiring | ENZYMES | IMDB-B | MUTAG | NCI1 | PROTEINS |
|---|---|---|---|---|---|---|
| None | None | $48.5 \pm 1.9$ | $66.6 \pm 3.7$ | $85.0 \pm 3.4$ | $60.8 \pm 3.8$ | $67.9 \pm 1.9$ |
| | AFR4 | $51.0 \pm 1.5$ | $61.0 \pm 4.1$ | $91.0 \pm 2.3$ | $54.8 \pm 3.7$ | $70.7 \pm 1.3$ |
| | FoSR | $50.5 \pm 1.8$ | $56.0 \pm 3.9$ | $82.0 \pm 2.5$ | $64.8 \pm 3.1$ | $71.4 \pm 1.4$ |
| | SDRF | $45.8 \pm 2.2$ | $53.3 \pm 2.3$ | $87.5 \pm 2.3$ | $58.0 \pm 4.4$ | $69.3 \pm 1.6$ |
| Clique | None | $50.5 \pm 2.1$ | $62.8 \pm 3.8$ | $90.5 \pm 2.2$ | $61.5 \pm 4.6$ | $68.3 \pm 1.3$ |
| | AFR4 | $52.7 \pm 1.6$ | $> 1$ day | $84.5 \pm 3.3$ | $60.6 \pm 4.8$ | $66.7 \pm 1.3$ |
| | FoSR | $43.0 \pm 2.1$ | $64.7 \pm 1.5$ | $78.5 \pm 2.4$ | $72.9 \pm 0.5$ | $71.9 \pm 1.0$ |
| | SDRF | $47.7 \pm 1.5$ | $> 12$ hours | $81.0 \pm 3.0$ | $76.8 \pm 0.4$ | $70.0 \pm 1.7$ |
| Ring | None | $47.5 \pm 1.7$ | $66.0 \pm 1.4$ | $85.5 \pm 2.0$ | $56.8 \pm 4.5$ | $68.1 \pm 1.2$ |
| | AFR4 | $46.3 \pm 1.8$ | $> 1$ day | $90.0 \pm 2.7$ | $59.5 \pm 4.1$ | $> 1$ day |
| | FoSR | $46.0 \pm 2.8$ | $67.8 \pm 1.3$ | $85.0 \pm 1.7$ | $71.9 \pm 0.8$ | $70.1 \pm 1.2$ |
| | SDRF | $43.2 \pm 1.5$ | $> 12$ hours | $81.5 \pm 3.1$ | $76.0 \pm 0.6$ | $69.1 \pm 1.0$ |

Table 10: Baseline and rewiring results for SIN, CIN, and CIN++. Numbers highlighted in blue correspond to no rewiring and numbers highlighted in red are the best among the rewiring methods.

| Parameter | Value |
|---|---|
| Number of Graphs | 1000 |
| Trials | 10 |
| Default Neighbors Match Cliques | 3 |
| Default Neighbors Match Clique Size | 5 |
| Default Ring Transfer Nodes | 10 |

Table 11: Synthetic Parameters

| Parameter | Value |
|---|---|
| Layers | 4 |
| Hidden | 64 |
| Dropout | 0.5 |
| Pooling | mean |
| Multidimensional | False |
| Max Dimension | 2 |

Table 12: Model Parameters

| Parameter | Value |
|---|---|
| Optimizer | Adam |
| Batch Size | 64 |
| Learning Rate | 0.001 |
| Max Epochs | 500 |
| Stop Criteria | Validation |
| Stop Factor | 1.01 |
| Stop Patience | 100 |

Table 13: Optimization Parameters

---

**Algorithm 2** Graph Lifting (None)

---

**Input:** Unweighted undirected graph $G = (V, E)$, node features $(\mathbf{x}_v)_{v \in V}$
**Output:** Relational structure $(\mathcal{S}, \{R_{\text{upper}}\})$ and features $(\mathbf{x}_\sigma)_{\sigma \in \mathcal{S}}$

1: Initialize $\mathcal{S} = \emptyset$ $\hspace{2cm}$ ▷ Entities
2: **for** each node $v \in V$ **do**
3: $\hspace{0.5cm} \sigma \leftarrow \{v\}$
4: $\hspace{0.5cm} \mathcal{S} \leftarrow \mathcal{S} \cup \{\sigma\}$
5: $\hspace{0.5cm} \mathbf{x}_\sigma \leftarrow \mathbf{x}_v$ $\hspace{2cm}$ ▷ Features
6: **end for**
7: Initialize $R_{\text{upper}} = \emptyset$ $\hspace{2cm}$ ▷ Relations
8: **for** each edge $\{u, v\} \in E$ **do**
9: $\hspace{0.5cm} \sigma, \tau \leftarrow u, v$
10: $\hspace{0.5cm} R_{\text{upper}} \leftarrow R_{\text{upper}} \cup \{(\sigma, \tau), (\tau, \sigma)\}$
11: **end for**
12: **return** $(\mathcal{S}, \{R_{\text{upper}}\})$ and $(\mathbf{x}_\sigma)_{\sigma \in \mathcal{S}}$

---

**Algorithm 3** Graph Lifting (Complex)

**Input:** Unweighted undirected graph $G = (V, E)$, node features $(\mathbf{x}_v)_{v \in V}$, relation types $\mathcal{A} \subseteq$ {boundary, coboundary, lower, upper}, max simplex dimension max_dim
**Output:** Relational structure $(\mathcal{S}, \{R_r\}_{r \in \mathcal{A}})$ and features $(\mathbf{x}_\sigma)_{\sigma \in \mathcal{S}}$

1: Initialize $\mathcal{S} = \emptyset$          ▷ Entities
2: **for** each non-empty clique $\sigma \subseteq V$ with $|\sigma| \leq$ max_dim $+ 1$ **do**
3:      $\mathcal{S} \leftarrow \mathcal{S} \cup \{\sigma\}$
4:      $\mathbf{x}_\sigma \leftarrow \frac{1}{|\sigma|} \sum_{v \in \sigma} \mathbf{x}_v$          ▷ Features
5: **end for**
6: **for** each relation type $r \in \mathcal{A}$ **do**
7:      Initialize $R_r = \emptyset$          ▷ Relations
8:      **for** each entity $\sigma \in \mathcal{S}$ **do**
9:          **if** $r =$ boundary **then**
10:              **for** each $\tau \in \mathcal{B}(\sigma)$ **do**
11:                  $R_r \leftarrow R_r \cup \{(\sigma, \tau)\}$
12:              **end for**
13:          **else if** $r =$ coboundary **then**
14:              **for** each $\tau \in \mathcal{C}(\sigma)$ **do**
15:                  $R_r \leftarrow R_r \cup \{(\sigma, \tau)\}$
16:              **end for**
17:          **else if** $r =$ lower **then**
18:              **for** each $\tau \in \mathcal{N}_\downarrow(\sigma)$ **do**
19:                  $R_r \leftarrow R_r \cup \{(\sigma, \tau, \sigma \cap \tau)\}$
20:              **end for**
21:          **else if** $r =$ upper **then**
22:              **for** each $\tau \in \mathcal{N}_\uparrow(\sigma)$ **do**
23:                  $R_r \leftarrow R_r \cup \{(\sigma, \tau, \sigma \cup \tau)\}$
24:              **end for**
25:          **end if**
26:      **end for**
27: **end for**
28: **return** $(\mathcal{S}, \{R_r\}_{r \in \mathcal{A}})$ and $(\mathbf{x}_\sigma)_{\sigma \in \mathcal{S}}$

**GCN** Following Kipf & Welling (2017), the GCN update rule is:

$$\mathbf{h}_\sigma^{(t+1)} = \mathbf{f}\left(\mathbf{W}^{(t)}\left(\sum_{r\in\mathcal{R}}\sum_{\tau\in\mathcal{N}_{R_i}(\sigma)\cup\{\sigma\}}\frac{1}{\sqrt{\hat{d}_\sigma\hat{d}_\tau}}\mathbf{h}_\tau^{(t)}\right)\right),$$

where $\mathbf{f}$ is a point-wise non-linearity (e.g., ReLU), $\mathbf{W}^{(t)}$ is a learnable weight matrix, and $\hat{d}_\tau = \deg(\tau) + 1$.

**GIN** Following Xu et al. (2019), the GIN update rule is:

$$\mathbf{h}_\sigma^{(t+1)} = \text{MLP}^{(t)}\left((1+\epsilon)\mathbf{h}_\sigma^{(t)} + \sum_{r\in\mathcal{R}}\sum_{\tau\in\mathcal{N}_{R_i}(\sigma)}\mathbf{W}^{(t)}\mathbf{h}_\tau^{(t)}\right),$$

where $\mathbf{W}^{(t)}$ is a learnable weight matrix and $\epsilon$ is a learnable scalar. Here, MLP consists of two layer fully connected layers, each followed by a point-wise non-linearity (e.g., ReLU) and a BatchNorm layer.

**R-GCN** Following Schlichtkrull et al. (2018), the R-GCN update rule is:

$$\mathbf{h}_\sigma^{(t+1)} = \mathbf{f}\left(\mathbf{W}_{\text{root}}^{(t)}\mathbf{h}_\sigma^{(t)} + \sum_{r\in\mathcal{R}}\sum_{j\in\mathcal{N}_r(i)}\frac{1}{|\mathcal{N}_r(i)|}\mathbf{W}_r^{(t)}\mathbf{h}_\sigma^{(t)}\right),$$

where $\mathbf{f}$ is a component-wise non-linearity (e.g., ReLU), and $\mathbf{W}_{\text{root}}^{(t)}$ and $\mathbf{W}_r^{(t)}$ are learnable weight matrices.

**R-GIN** Following Xu et al. (2019) and Schlichtkrull et al. (2018), the R-GIN update rule is

$$\mathbf{h}_\sigma^{(t+1)} = \mathbf{W}_{\text{root}}^{(t)}\mathbf{h}_\sigma^{(t)} + \sum_{r\in\mathcal{R}}\text{MLP}_r^{(t)}\left((1+\epsilon_r)\mathbf{h}_\sigma^{(t)} + \sum_{\tau\in\mathcal{N}_r(\sigma)}\mathbf{h}_\tau^{(t)}\right),$$

where $\mathbf{W}_{\text{root}}^{(t)}$ is a learnable weight matrix and each $\epsilon_\bullet$ is a learnable scalar. Here, each MLP consists of two layer fully connected layers, each followed by a point-wise non-linearity (e.g., ReLU) and a BatchNorm layer. and $\text{MLP}_r^{(t)}$ is a relation-specific MLP.

**SIN** Following Bodnar et al. (2021b), the SIN update rule is:

$$\mathbf{h}_\sigma^{(t+1)} = \text{MLP}_{U,p}^{(t)}\left[\begin{array}{c} \text{MLP}_{B,p}\left((1+\epsilon_\mathcal{B})\mathbf{h}_\sigma^{(t)} + \sum_{\delta\in\mathcal{B}(\sigma)}\mathbf{h}_\delta^{(t)}\right) \\ \text{MLP}_{\uparrow,p}\left((1+\epsilon_\uparrow)\mathbf{h}_\sigma^{(t)} + \sum_{\delta\in\mathcal{N}_\uparrow(\sigma)}\mathbf{h}_\delta^{(t)}\right) \\ \textit{additionally, when rewiring:} \\ \overline{\left[\text{MLP}_{R_{new},p}\left((1+\epsilon_{R_{new}})\mathbf{h}_\sigma^{(t)} + \sum_{\delta\in\mathcal{N}_{R_{new}}(\sigma)}\mathbf{h}_\delta^{(t)}\right)\right]} \end{array}\right],$$

where $[\cdot]$ denotes column-wise concatenation and each $\epsilon_\bullet$ is a learnable scalar. Here, each MLP consists of two layer fully connected layers, each followed by a point-wise non-linearity (e.g., ReLU) and a BatchNorm layer. and $\text{MLP}_r^{(t)}$ is a relation-specific MLP.

**CIN** Following Bodnar et al. (2021a), the CIN update rule is:

$$
\mathbf{h}_\sigma^{(t+1)} = \mathrm{MLP}_{U,p}^{(t)} \begin{bmatrix} \mathrm{MLP}_{B,p}\left( (1+\epsilon_\mathcal{B})\,\mathbf{h}_\sigma^{(t)} + \sum_{\tau\in\mathcal{B}(\sigma)} \mathbf{h}_\tau^{(t)} \right) \\ \mathrm{MLP}_{\uparrow,p}\left( (1+\epsilon_\uparrow)\,\mathbf{h}_\sigma^{(t)} + \sum_{\substack{\tau\in\mathcal{N}_\uparrow(\sigma) \\ \delta\in C(\sigma,\tau)}} \mathrm{MLP}_{M,p}\begin{bmatrix} \mathbf{h}_\tau^{(t)} \\ \mathbf{h}_\delta^{(t)} \end{bmatrix} \right) \\ \textit{additionally, when rewiring:} \\ \boxed{\mathrm{MLP}_{R_{new},p}\left( (1+\epsilon_{R_\mathrm{new}})\,\mathbf{h}_\sigma^{(t)} + \sum_{\delta\in\mathcal{N}_{R_\mathrm{new}}(\sigma)} \mathbf{h}_\delta^{(t)} \right)} \end{bmatrix},
$$

where $[\cdot]$ denotes column-wise concatenation, each $\epsilon_\bullet$ is a learnable scalar, and $\mathrm{MLP}_{M,p}$ is a single fully connected layer projecting back to the dimension of $\mathbf{h}_\sigma^{(t)}$. The other MLPs consists of two layer fully connected layers, each followed by a point-wise non-linearity (e.g., ReLU) and a BatchNorm layer. When entity unions aren't contained in the relational structure due to dimension constraints, we instead use $\mathbf{h}_\delta^{(t)} := \mathbf{0}$.

**CIN++** Following Giusti et al. (2024), the CIN++ update rule is:

$$
\mathbf{h}_\sigma^{(t+1)} = \mathrm{MLP}_{U,p}^{(t)} \begin{bmatrix} \mathrm{MLP}_{B,p}\left( (1+\epsilon_\mathcal{B})\,\mathbf{h}_\sigma^{(t)} + \sum_{\tau\in\mathcal{B}(\sigma)} \mathbf{h}_\tau^{(t)} \right) \\ \mathrm{MLP}_{\uparrow,p}\left( (1+\epsilon_\uparrow)\,\mathbf{h}_\sigma^{(t)} + \sum_{\substack{\tau\in\mathcal{N}_\uparrow(\sigma) \\ \delta\in C(\sigma,\tau)}} \mathrm{MLP}_{M,p}\begin{bmatrix} \mathbf{h}_\tau^{(t)} \\ \mathbf{h}_\delta^{(t)} \end{bmatrix} \right) \\ \mathrm{MLP}_{\downarrow,p}\left( (1+\epsilon_\downarrow)\,\mathbf{h}_\sigma^{(t)} + \sum_{\substack{\tau\in\mathcal{N}_\downarrow(\sigma) \\ \delta\in B(\sigma,\tau)}} \mathrm{MLP}_{M',p}\begin{bmatrix} \mathbf{h}_\tau^{(t)} \\ \mathbf{h}_\delta^{(t)} \end{bmatrix} \right) \\ \textit{additionally, when rewiring:} \\ \boxed{\mathrm{MLP}_{R_\mathrm{new},p}\left( (1+\epsilon_{R_\mathrm{new}})\,\mathbf{h}_\sigma^{(t)} + \sum_{\delta\in\mathcal{N}_{R_\mathrm{new}}(\sigma)} \mathbf{h}_\delta^{(t)} \right)} \end{bmatrix},
$$

where $[\cdot]$ denotes column-wise concatenation, each $\epsilon_\bullet$ is a learnable scalar. $\mathrm{MLP}_{M,p}$ and $\mathrm{MLP}_{M',p}$ are single fully connected layers projecting back to the dimension of $\mathbf{h}_\sigma^{(t)}$. The other MLPs consists of two layer fully connected layers, each followed by a point-wise non-linearity (e.g., ReLU) and a BatchNorm layer. When entity unions or intersections aren't contained in the relational structure due to dimension constraints, we instead use $\mathbf{h}_\delta^{(t)} := \mathbf{0}$.

## G  HIGHER-ORDER GRAPHS ARE RELATIONAL STRUCTURES

Although we do not focus on higher-order graph neural networks ($k$-GNNs) (Morris et al., 2019) in this work, we briefly illustrate how they naturally fit into our relational framework. In $k$-GNNs, graphs are lifted into higher-order graphs following the procedure outlined in Algorithm 4. In our framework, these higher-order graphs can be viewed as relational structures, where each entity represents a set of nodes of size $n = k$. The relations between these entities are of two types: *local* relations, which connect entities differing by one node and where the differing nodes are adjacent

in the original graph, and *global* relations, where the differing nodes are not adjacent in the original graph. From this perspective, $k$-GNN message passing on higher-order graphs is equivalent to R-GCN message passing on the corresponding relational structures. We leave the theoretical and empirical study of these relational structures for future work.

---

**Algorithm 4** Graph Lifting (Higher-Order Graph)

---

**Input:** Unweighted undirected graph $G = (V, E)$, node features $(\mathbf{x}_v)_{v \in V}$, entity size $n$
**Output:** Relational structure $(\mathcal{S}, \{R_{\texttt{local}}, R_{\texttt{global}}\})$ and features $(\mathbf{x}_\sigma)_{\sigma \in \mathcal{S}}$

1: Initialize $\mathcal{S} = \emptyset$          ▷ Entities
2: **for** each subset $\sigma \subseteq V$ with $|\sigma| = n$ **do**
3:     $\mathcal{S} \leftarrow \mathcal{S} \cup \{\sigma\}$
4:     $\mathbf{x}_\sigma \leftarrow \frac{1}{|\sigma|} \sum_{v \in \sigma} \mathbf{x}_v$          ▷ Features
5: **end for**
6: Initialize $R_{\texttt{local}} = \emptyset$, $R_{\texttt{global}} = \emptyset$          ▷ Relations
7: **for** each pair $\sigma, \tau \in \mathcal{S}$ **do**
8:     **if** $|\sigma \cap \tau| = n - 1$ **then**
9:         Let $\{s_\sigma\} = \sigma \setminus \tau$ and $\{t_\tau\} = \tau \setminus \sigma$
10:         **if** $\{s_\sigma, t_\tau\} \in E$ **then**
11:             $R_{\texttt{local}} \leftarrow R_{\texttt{local}} \cup \{(\sigma, \tau), (\tau, \sigma)\}$
12:         **else**
13:             $R_{\texttt{global}} \leftarrow R_{\texttt{global}} \cup \{(\sigma, \tau), (\tau, \sigma)\}$
14:         **end if**
15:     **end if**
16: **end for**
17: **return** $(\mathcal{S}, \{R_{\texttt{local}}, R_{\texttt{global}}\})$ and $(\mathbf{x}_\sigma)_{\sigma \in \mathcal{S}}$

---

## H    WORKED EXAMPLE: RELATIONS AND OPERATORS FOR A PATH GRAPH

In this section, we illustrate, via a small example, the definitions of a simplicial complex, relational structure, and influence graph.

**Simplicial Complex.**    Consider the simplicial complex $\mathcal{K}$ corresponding to the path graph $i - j - k$. The complex $\mathcal{K}$ consists of

- 0-simplices (vertices): $\{i\}, \{j\}, \{k\}$, and
- 1-simplices (edges): $\{i, j\}, \{j, k\}$.

**Relational Structure.**    The relations on the set of entities $\mathcal{S} = \mathcal{K}$ are as follows:

- Relation $R_1$ (identity):

$$
\begin{aligned}
R_1 &= \{(\sigma) \mid \sigma \in \mathcal{K}\} \\
&= \{(\{i\}), (\{j\}), (\{k\}), (\{i, j\}), (\{j, k\})\}.
\end{aligned}
$$

- Relation $R_2$ (boundary):

$$
\begin{aligned}
R_2 &= \{(\sigma, \tau) \mid \sigma \in \mathcal{K}, \tau \in \mathcal{B}(\sigma)\} \\
&= \left\{ \begin{array}{l} (\{i, j\}, \{i\}), (\{i, j\}, \{j\}), \\ (\{j, k\}, \{j\}), (\{j, k\}, \{k\}) \end{array} \right\}.
\end{aligned}
$$

- Relation $R_3$ (co-boundary):

$$
\begin{aligned}
R_3 &= \{(\sigma, \tau) \mid \sigma \in \mathcal{K}, \tau \in \mathcal{C}(\sigma)\} \\
&= \left\{ \begin{array}{l} (\{i\}, \{i, j\}), (\{j\}, \{i, j\}), \\ (\{j\}, \{j, k\}), (\{k\}, \{j, k\}) \end{array} \right\}.
\end{aligned}
$$

- Relation $R_4$ (lower adjacency):

$$R_4 = \{(\sigma, \tau, \delta) \mid \sigma \in \mathcal{K}, \tau \in \mathcal{N}_\uparrow(\sigma), \delta = \sigma \cap \tau\}$$
$$= \left\{ \begin{array}{l} (\{i,j\}, \{j,k\}, \{j\}), \\ (\{j,k\}, \{i,j\}, \{j\}) \end{array} \right\}.$$

- Relation $R_5$ (upper adjacency):

$$R_5 = \{(\sigma, \tau, \delta) \mid \sigma \in \mathcal{K}, \tau \in \mathcal{N}_\downarrow(\sigma), \delta = \sigma \cup \tau\}$$
$$= \left\{ \begin{array}{l} (\{i\}, \{j\}, \{i,j\}), \ (\{j\}, \{i\}, \{i,j\}), \\ (\{j\}, \{k\}, \{j,k\}), \ (\{k\}, \{j\}, \{j,k\}) \end{array} \right\}.$$

If we assume, for the sake of example, that all relations are involved in message passing, this gives rise to the relational structure

$$\mathcal{R}(\mathcal{K}) = (\mathcal{S}, R_1, R_2, R_3, R_4, R_5).$$

Of course, one could restrict message passing to a smaller subset of the relations $\{R_1, R_2, R_3, R_4, R_5\}$.

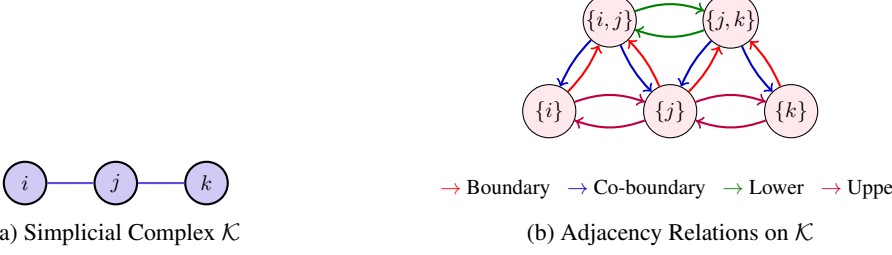

$\rightarrow$ Boundary   $\rightarrow$ Co-boundary   $\rightarrow$ Lower   $\rightarrow$ Upper

(a) Simplicial Complex $\mathcal{K}$        (b) Adjacency Relations on $\mathcal{K}$

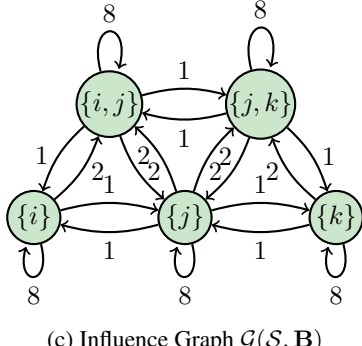

(c) Influence Graph $\mathcal{G}(\mathcal{S}, \mathbf{B})$

Figure 12: (a) The simplicial complex $\mathcal{K}$ consisting of nodes $i$, $j$, $k$, and edges $\{i,j\}$, $\{j,k\}$. (b) The adjacency relations on $\mathcal{K}$, showing boundary, co-boundary, lower, and upper relations. (c) The influence graph $\mathcal{G}(\mathcal{S}, \mathbf{B})$ representing the information flow between the entities.

**Shift Operators.** In what follows, we index the entries in matrices as follows:

| Index | Simplex |
|-------|---------|
| 1 | $\{i\}$ |
| 2 | $\{j\}$ |
| 3 | $\{k\}$ |
| 4 | $\{i,j\}$ |
| 5 | $\{j,k\}$ |

For instance, the 14th entry in a matrix would correspond to the connection $(\{i\}, \{i, j\})$. We assume, for the sake of example, that all shift operators correspond to the adjacency tensors, i.e., the entries with indices corresponding to entities that are related by a relation are equal to $1$, and the remaining entries are all $0$. Of course, normalized adjacency tensors and others could be similarly considered. As such, we get the following:

- Shift operators $\mathbf{A}^{R_1}$ and $\tilde{\mathbf{A}}^{R_1}$ (identity relation):

$$\mathbf{A}^{R_1} = \tilde{\mathbf{A}}^{R_1} = \begin{pmatrix} 1 & 0 & 0 & 0 & 0 \\ 0 & 1 & 0 & 0 & 0 \\ 0 & 0 & 1 & 0 & 0 \\ 0 & 0 & 0 & 1 & 0 \\ 0 & 0 & 0 & 0 & 1 \end{pmatrix}.$$

- Shift operators $\mathbf{A}^{R_2}$ and $\tilde{\mathbf{A}}^{R_2}$ (boundary relation):

$$\mathbf{A}^{R_2} = \tilde{\mathbf{A}}^{R_2} = \begin{pmatrix} 0 & 0 & 0 & 0 & 0 \\ 0 & 0 & 0 & 0 & 0 \\ 0 & 0 & 0 & 0 & 0 \\ 1 & 1 & 0 & 0 & 0 \\ 0 & 1 & 1 & 0 & 0 \end{pmatrix}.$$

- Shift operators $\mathbf{A}^{R_3}$ and $\tilde{\mathbf{A}}^{R_3}$ (co-boundary relation):

$$\mathbf{A}^{R_3} = \tilde{\mathbf{A}}^{R_3} = \begin{pmatrix} 0 & 0 & 0 & 1 & 0 \\ 0 & 0 & 0 & 1 & 1 \\ 0 & 0 & 0 & 0 & 1 \\ 0 & 0 & 0 & 0 & 0 \\ 0 & 0 & 0 & 0 & 0 \end{pmatrix}.$$

- Shift operator $\mathbf{A}^{R_4}$ and $\tilde{\mathbf{A}}^{R_4}$ (lower adjacency relation):
  The non-zero entries of $\mathbf{A}^{R_4}$ are

$$\mathbf{A}^{R_4} : \begin{cases} \mathbf{A}^{R_4}_{\{i,j\},\{j,k\},\{j\}} = 1 \\ \mathbf{A}^{R_4}_{\{j,k\},\{i,j\},\{j\}} = 1 \end{cases}.$$

  The aggregated influence for $\tilde{\mathbf{A}}^{R_4}$:

$$\tilde{\mathbf{A}}^{R_4} = \begin{pmatrix} 0 & 0 & 0 & 0 & 0 \\ 0 & 0 & 0 & 0 & 0 \\ 0 & 0 & 0 & 0 & 0 \\ 0 & 1 & 0 & 0 & 1 \\ 0 & 1 & 0 & 1 & 0 \end{pmatrix}.$$

- Shift Operator $\mathbf{A}^{R_5}$ and $\tilde{\mathbf{A}}^{R_5}$ (upper adjacency relation):
  The non-zero entries of $\mathbf{A}^{R_5}$ are

$$\mathbf{A}^{R_5} : \begin{cases} \mathbf{A}^{R_5}_{\{i\},\{j\},\{i,j\}} = 1 \\ \mathbf{A}^{R_5}_{\{j\},\{i\},\{i,j\}} = 1 \\ \mathbf{A}^{R_5}_{\{j\},\{k\},\{j,k\}} = 1 \\ \mathbf{A}^{R_5}_{\{k\},\{j\},\{j,k\}} = 1 \end{cases}.$$

  The aggregated influence for $\tilde{\mathbf{A}}^{R_5}$:

$$\tilde{\mathbf{A}}^{R_5} = \begin{pmatrix} 0 & 1 & 0 & 1 & 0 \\ 1 & 0 & 1 & 1 & 1 \\ 0 & 1 & 0 & 0 & 1 \\ 0 & 0 & 0 & 0 & 0 \\ 0 & 0 & 0 & 0 & 0 \end{pmatrix}.$$

.

Lastly, the aggregated influence matrix $\tilde{\mathbf{A}}$ arising from including all relations $R_1$ to $R_5$:

$$
\begin{aligned}
\tilde{\mathbf{A}} &= \tilde{\mathbf{A}}^{R_1} + \tilde{\mathbf{A}}^{R_2} + \tilde{\mathbf{A}}^{R_3} + \tilde{\mathbf{A}}^{R_4} + \tilde{\mathbf{A}}^{R_5} \\
&= \begin{pmatrix}
1 & 1 & 0 & 2 & 0 \\
1 & 1 & 1 & 2 & 2 \\
0 & 1 & 1 & 0 & 2 \\
1 & 2 & 0 & 1 & 1 \\
0 & 2 & 1 & 1 & 1
\end{pmatrix}.
\end{aligned}
$$

The maximum row sum of $\tilde{\mathbf{A}}$ is $\gamma = 7$, giving rise to the augmented adjacency matrix

$$
\begin{aligned}
\mathbf{B} &= \gamma \mathbf{I} + \tilde{\mathbf{A}} \\
&= \begin{pmatrix}
8 & 1 & 0 & 2 & 0 \\
1 & 8 & 1 & 2 & 2 \\
0 & 1 & 8 & 0 & 2 \\
1 & 2 & 0 & 8 & 1 \\
0 & 2 & 1 & 1 & 8
\end{pmatrix}.
\end{aligned}
$$

The influence graph is finally constructed from $\mathbf{B}$ as follows:

- Nodes: Each simplex in $\mathcal{K}$.
- Edges: For each $(\sigma, \tau)$, if $\mathbf{B}_{\sigma,\tau} > 0$, there is a directed edge from $\tau$ to $\sigma$ with weight $\mathbf{B}_{\sigma,\tau}$.

## I  COMPUTATIONAL COMPLEXITY

It is well known in the TDL community that topological methods can require much more computation due to directly modeling higher-order structures. This typically incurs combinatorial complexity in the algorithms. A common strategy is to truncate the dimensionality of the objects considered to maintain polynomial complexity. Another option is to use a sparse complex construction method. As noted earlier, more work is needed in understanding graph lifting and which method is preferred in a given setting. In this section, we analyze the running times required for TDL with clique complexes.

### I.1  CLIQUE COMPLEX SIZE

The exact computational requirements of performing clique lifting depend on the graph in consideration. However, there are a few cases where we can provide precise estimates.

**Clique lifting of dimension 1**  In this case, the highest dimensional cell is the edge cell. Consider a graph $G = (V, E)$ with $|V| = n$ nodes, $|E| = m$ edges, and maximum node degree $d$. The corresponding clique complex $\mathcal{K}$ has:

1. $n + m$ cells
2. $2m$ boundary relations
3. $2m$ co-boundary relations
4. $\sum_{v \in V} 2\binom{\deg(v)}{2} \leq 2md$ lower relations
5. $2m$ upper relations

Note that we need to count the upper and lower relations for each direction of the edge so we get a factor of 2.

**Clique lifting of dimension 2**  If $G$ has $|F| = k$ faces, then the clique $\mathcal{K}$ contains

1. $n + m + k$ cells
2. $2m + 3k$ boundary relations
3. $2m + 3k$ co-boundary relations

4. $\sum_{v \in V} 2\binom{\deg(v)}{2} + \sum_{\sigma \in F} \sum_{\tau \in F} \chi(\sigma \cap \tau)$ lower relations
5. $2m + 6k$ upper relations

In dense graphs, the number of lower relations can be very large. Depending on the dataset under consideration, one may opt for a model like CIN where lower relations are ignored.

**Complete Graph**  We can provide precise estimates when we have a complete graph. This can help us understand performance in the limiting case that the graph is very dense or contains large cliques. Consider a complete graph $K_n$ on $n$ nodes and the cell $\sigma = \{1, 2, ..., m\}$ for $m \geq 1$. This is the boundary of $n - m$ cells and is the co-boundary of $m$ cells. It is upper adjacent to $(n - m)m$ cells. It is lower adjacent to $m(n - m)$ cells. Counting all the cells of dimension $d = m - 1$, we get

1. $n_d = \binom{n}{d+1}$ cells
2. $n_d(n - d - 1)$ boundary relations (if $d > 0$)
3. $n_d(d + 1)$ co-boundary relations
4. $n_d(d + 1)(n - d - 1)$ lower relations (if $d > 0$)
5. $n_d(d + 1)(n - d - 1)$ upper relations

, where the $d > 0$ condition comes from the fact that there isn't a cell $\{\}$ in the complex so there aren't lower or co-boundary relations for zero dimensional cells. Computing the total number of cells gives $2^n - 1$. The total number of edges is

$$-n - n(n-1) + \sum_{d=0}^{n-1} \binom{n}{d+1}(n + (n - d - 1)(d + 1))$$

$$= \frac{n}{2}(2^n n + 2^n - 2) - n^2 = O(n^2 2^n)$$

This bound is prohibitive for large $n$. If we restrict to two dimensional cells, we get the total number of cells is

$$n + \binom{n}{2} + \binom{n}{3} = \frac{n^3 + 5n}{6} = O(n^3)$$

and the total number of relations is

$$\frac{n}{6}(-24 + 41n - 24n^2 + 7n^3) = O(n^4)$$

## I.2 CLIQUE LIFT COMPLEXITY

Our main use case is computing complexes for two-dimensional cells. In this section, we compute the complexity of clique lifting. Note that computing the 0 and 1-dimensional cells is fast since they are encoded directly in the graph as nodes and edges, respectively. The boundary and coboundary relations encode edge incidence to a node and are also easily obtained. The upper relations on the 0-cells are just the edges in the graph. The lower relations on the edges are all pairs of edges in the graph sharing a node. This can be efficiently computed by storing the edge incidences in a dictionary in $O(m)$ time and then computing all pairs in the neighborhood of each node. This requires $O\left(n + m + \sum_{v \in V} \binom{\deg(v)}{2}\right)$ computation. For connected graphs, this reduces to $O(md)$.

Computing the two-dimensional cells can also be done efficiently. By storing the neighborhood dictionary, we just need to loop over pairs of edges $(u, v), (u, w)$ in the neighborhood of $u$ and check if $(v, w) \in G$. This can be done in $O\left(\sum_{v \in V} \binom{\deg(v)}{2}\right) = O(md)$ time. The boundary, co-boundary, and upper adjacencies can be easily computed in $O(k)$ time. The lower adjacency computation for 2-cells can require more time. For a given edge $(u, v)$, consider the 2-cells $\sigma$ such that $(u, v) \prec \sigma$. Each pair of these gets a lower adjacency and can be efficiently added. This results in

$$O\left(\sum_{e \in E} \binom{|\mathcal{C}(e)|}{2}\right)$$

computation. We can bound this as

$$\sum_{e \in E} \binom{|\mathcal{C}(e)|}{2} \leq \frac{1}{2} \sum_{e \in E} |\mathcal{C}(e)|^2 \leq \frac{1}{2} \max_e |\mathcal{C}(e)| \sum_{e \in E} |\mathcal{C}(e)| = \frac{3k}{2} \max_e |\mathcal{C}(e)|$$

and we get the complexity bound

$$O\left(k \max_e |\mathcal{C}(e)|\right)$$

This is analogous to the result with lower adjacency of 1-cells that relied on the maximum node degree in $G$. The maximum co-boundary degree $\max_e |\mathcal{C}(e)|$ is the analogue of node degree for 2-cells. So, the clique complex construction up to dimension 2 has time complexity:

$$O(md + k \max_e |\mathcal{C}(e)|)$$

### I.3    RUNNING TIMES

| Lift | None | Clique | Ring |
|------|------|--------|------|
| ENZYMES | 2.12 | 6.89 | 11.81 |
| IMDB-B | 3.51 | 45.04 | 101.98 |
| MUTAG | 1.60 | 1.22 | 1.48 |
| NCI1 | 4.70 | 20.26 | 26.62 |
| PROTEINS | 2.99 | 16.07 | 30.73 |
| ZINC | 63.47 | 94.70 | 106.37 |
| TEXAS | 6.02 | 2.64 | 0.65 |
| WISCONSIN | 6.09 | 2.78 | 0.74 |
| CORNELL | 5.05 | 2.76 | 0.62 |
| CORA | 5.77 | 8.29 | 45.71 |
| CITESEER | 7.05 | 7.65 | NA |

Table 14: Time to perform graph lifting and download datasets from Torch Geometric.

