# OpenReview forum: "Demystifying Topological Message-Passing with Relational Structures: A Case Study on Oversquashing in Simplicial Message-Passing"
_ICLR.cc/2025/Conference — ICLR 2025 Poster_

### Official Review · Reviewer_SPF6 · 2024-10-29

**Soundness:** 2
**Presentation:** 2
**Contribution:** 2
**Rating:** 6
**Confidence:** 4

**Summary:**

This paper proposes a unified axiomatic framework, aiming at the over-squashing phenomenon and the lack of theoretical analysis in graph topological message passing. The framework examines simplicial complexes and their message-passing schemes from the perspective of relational structures, closely combines graphs and topological message passing, and extends them to high-order structures, enhancing the theoretical analysis ability and effectively alleviating the over-squeezing problem.

**Strengths:**

1. The message-passing theory of simple complexes and their relational structures is comprehensively and intensely studied, and a detailed explanation and rigorous proof are provided.

2. The article has a clear structure and a smooth and natural expression, which is easy for readers to understand and grasp.

**Weaknesses:**

1. The literature review on relational graph neural networks mainly focuses on research in 2020 and before, which limits the comprehensive grasp of the latest progress in this field. It is necessary to include more recent research results to enhance the timeliness and persuasiveness of the review.

2. Although the comparison method includes a baseline for 2024, more is needed in terms of SOTA (state-of-the-art) comparison in the past two years to ensure the comprehensiveness and superiority of the evaluation.

3. Given that this study aims to explore the over-squeezing problem, unfortunately, the paper fails to show the results of the application of this method in deeper models, thus failing to verify and strengthen the conclusions obtained in a more in-depth manner. The lack limits the comprehensiveness and persuasiveness of the study.

4. Although the paper mentions the impact of local complexity, model depth, and hidden dimension on performance, unfortunately, it fails to provide corresponding ablation experiments to verify the impact and effectiveness of these factors from an experimental level, which to some extent weakens the experimental support for the conclusions.

5. Many of the formulas in the paper are not original, so it is recommended that the author provide corresponding references for these equations.

6. Why does the article take up so much space on the simplicial complex and its influence on over-squashing? Although this part is also essential, its level of detail exceeds the description of the solution part far, giving people a slightly unbalanced impression. The content distribution can be appropriately adjusted to make the solution part more detailed to ensure the focus is highlighted.

**Questions:**

Please refer to weaknesses.

---

> ### Author Response · Authors · 2024-11-21
> **Author Response (1/2)**
>
> We thank the reviewer for their evaluation, and for the very valuable questions and suggestions that will improve the manuscript.
>
> (W1) At the reviewer’s request, we have expanded the related work section (Appendix A) to include more recent research on relational graph neural networks. If there are additional recent references we should also include, we kindly ask the reviewer to let us know, so that no important recent contributions are overlooked. We are committed to addressing any potential gaps or oversights in our review.
>
> (W2) Our work focuses on oversquashing as a failure mode of the message passing paradigm. To study this phenomenon, we followed prior work in evaluating oversquashing using synthetic and real-world benchmarks with prototypical message passing models such as GCN, R-GCN, GIN, and R-GIN, as our primary focus is on understanding message passing behavior itself. For example, we compare our work in this regard to [1,2,3,4,6]. Additionally, we benchmark these prototypical graph message passing models with analogous topological message passing frameworks, namely SIN, CIN, and CIN++. We note that earlier models like these are also commonly used in recent studies to benchmark related phenomena, such as oversmoothing (see, e.g., [5]).
>
> If the reviewer has suggestions for additional SOTA models or comparisons that would further strengthen our evaluation, we would greatly appreciate their suggestions. Due to the limited duration of the discussion period and computational resources, we can include some of those evaluations in the final version, or cite them as directions for future work.
>
> (W3) We expanded the experiments with node classification on 5 datasets in Appendix D6 and the ZINC dataset in Appendix D5. We also added the ring lift to the ZINC experiment in order to better show the impact of different lifting techniques. Due to the limited duration of the discussion period and computational resources, the new experiments are limited in scope. The empirical results will be fully expanded in the final version of the manuscript.

---

> ### Author Response · Authors · 2024-11-21
> **Author Response (2/2)**
>
> (W4) We acknowledge that while we validated our theoretical findings on the impact of local geometry, depth, and width using a synthetic benchmark, we did not conduct ablation studies for the real-world benchmarks. In response to the reviewer’s feedback, we have run additional ablation tests on the impact of model depth and width, with the results to be expanded in the final version and report the results in Appendix D7. It is important to note that local geometry in real-world datasets is intrinsic to each graph and not a property of the model that can be directly ablated. Therefore, our additional experiments focus on model depth and width. We hope this addresses the reviewer’s concerns.
>
> (W5) While we added references for the mathematical concepts and results we used or extended to the best of our ability, it is possible we may have unintentionally missed providing references for some equations. If the reviewer could kindly provide examples of equations that need references, we will ensure those and any similar equations are properly cited. We are committed to addressing any oversights or gaps in attribution.
>
> (W6) Our work is a necessary first step towards addressing questions on oversquashing in the topological message passing community, and we propose leveraging concepts and methods from other communities (e.g., relational message passing) to attack those questions. As such, we aimed to contextualize our work and provide an introduction that would resonate with researchers from both communities. As such, the first half of Section 2 (“Simplicial Complexes are Relational Structures”) is dedicated to simplicial complexes, and the second half to relational structures. Section 2 in total occupies around 2 pages of the main paper. Sections 3 (theory) and 4 (rewiring heuristic) are not specific to simplicial complexes and apply to all relational message passing schemes covered by the model from Definition 2.6. Sections 3 and 4 together occupy around 3.5 pages in the main paper. Our presentation can definitely be improved, however. If the reviewer could comment on the kind of details we should add and where, we will try to accommodate that request as best as we can during the limited duration of the discussion period.
>
> We hope our responses and updates address the reviewer’s concerns and requests. We welcome any further comments or suggestions.
>
> References
>
> [1] Uri Alon and Eran Yahav. On the bottleneck of graph neural networks and its practical implications. In International Conference on Learning Representations (ICLR), 2021.
>
> [2] Kedar Karhadkar, Pradeep Kr. Banerjee, and Guido Montufar. FoSR: First-order spectral rewiring for addressing oversquashing in GNNs. In International Conference on Learning Representations (ICLR), 2023.
>
> [3] Khang Nguyen, Nong Minh Hieu, Vinh Duc Nguyen, Nhat Ho, Stanley Osher, and Tan Minh Nguyen. Revisiting over-smoothing and over-squashing using Ollivier-Ricci curvature. In International Conference on Machine Learning (ICML), 2023.
>
> [4] Jake Topping, Francesco Di Giovanni, Benjamin Paul Chamberlain, Xiaowen Dong, and Michael M. Bronstein. Understanding over-squashing and bottlenecks on graphs via curvature. In International Conference on Learning Representations (ICLR), 2022.
>
> [5] T. Konstantin Rusch, Michael M. Bronstein, and Siddhartha Mishra. A survey on oversmoothing in graph neural networks. arXiv preprint, arXiv:2303.10993, 2023.
>
> [6] Lukas Fesser and Melanie Weber. Mitigating over-smoothing and over-squashing using augmentations of Forman-Ricci curvature. In Learning on Graphs Conference (LoG), PMLR, 2024.

---

### Official Review · Reviewer_Qn33 · 2024-10-30

**Soundness:** 3
**Presentation:** 3
**Contribution:** 2
**Rating:** 6
**Confidence:** 3

**Summary:**

This paper addresses a well-known issue in Graph Neural Networks known as over-squashing. It introduces a framework called the Relational Message Passing Model, which unifies various higher-order message-passing approaches. This unification is leveraged to theoretically examine over-squashing within the context of higher-order message-passing.

**Strengths:**

This paper addresses a critical gap in the literature on over-squashing. Over-squashing remains underexplored in the context of higher-order message-passing architectures like simplicial neural networks and CW networks. This paper takes an essential first step toward addressing this gap within Topological Deep Learning. By presenting a unifying, axiomatic framework, a relational perspective on simplicial and cellular complexes is provided. This unified approach facilitates the analysis of over-squashing in topological message-passing networks.

**Weaknesses:**

The paper appears to be a relatively straightforward extension of Topping et al. (2022). This impression arises primarily from Definition 2.6, which is positioned as the core contribution. This definition raises several concerns:

1. The substitution of $R_i$ with $\mathcal{S}^{n_i}$
  seems to have minimal impact on the overall definition, effectively diminishing the role of relations $R_i$ within the definition.

2. The shift operator $A^{R_i}$, associated with relation $R_i$, quantifies the signal passed from $(\xi_1, \cdots, \xi_{n_i-1})$ to $\sigma$ when $(\sigma, \xi_1, \cdots, \xi_{n_i-1}) \in R_i$. However, it does not account for the signal passed from $(\sigma, \xi_2, \cdots, \xi_{n_i-1})$ to $\xi_1$. Consequently, $A^{R_i}$ may not serve as an adequate representation of $R_i$.

3. The update rule and message functions $m^{(t)}_{\sigma,i}$ can be viewed as updated versions of the update rule and message function that appear in Equation 1 from Topping et al. (2022), adapted for dimensions $k$ and $n_i$. These functions exhibit a behavior similar to the update rule and message functions in Equation 1 from Topping et al. (2022) when computing the Jacobian in Lemma 3.2. Consequently, this lemma appears to be a straightforward extension of Lemma 1 from Topping et al., lacking significant innovation or depth.


Topping et al. "Understanding Over-Squashing and Bottlenecks on Graphs via Curvature.", 2022.

Minor Comments:

Line 34: "Questions 2 and 9" should be changed to "research directions 2 and 9."

Definition 2.3: The symbol $\succ$ is used but not defined. This definition may be updated by replacing $\succ$ with $\prec$.

Definition 2.4: Is this definition necessary for the paper’s main arguments?

Definition 3.1: In the expression "$Q \in \tilde{A}, B$", it would be clearer to write "$Q \in \lbrace \tilde{A}, B \rbrace$."

**Questions:**

Please refer to the weaknesses.

Suggestion:

It is observed in line 167 that certain projections exist from $ R_4(\sigma) $ to $ R_2(\sigma) $ and from $R_2(\sigma) $ to $ R_1(\sigma) $ (similarly, from $ R_5(\sigma) $ to $ R_3(\sigma) $ and from $ R_3(\sigma) $ to $R_1(\sigma) $). The relational structure could be further enriched by treating the set of relations as a directed set. Considering the direct limit of this directed set may provide a comprehensive summary of all the relations within the given relational structure. This could serve as a terminal object for defining message functions, potentially enhancing the model’s representational capacity.

---

> ### Author Response · Authors · 2024-11-21
> **Author Response (1/2)**
>
> We thank the reviewer for their evaluation, and for the very valuable questions and suggestions that will improve the manuscript.
>
> > The paper appears to be a relatively straightforward extension of Topping et al. (2022). This impression arises primarily from Definition 2.6, which is positioned as the core contribution.
>
> > The update rule and message functions mσ,i(t) can be viewed as updated versions of the update rule and message function that appear in Equation 1 from Topping et al. (2022), adapted for dimensions k and ni. These functions exhibit a behavior similar to the update rule and message functions in Equation 1 from Topping et al. (2022) when computing the Jacobian in Lemma 3.2. Consequently, this lemma appears to be a straightforward extension of Lemma 1 from Topping et al., lacking significant innovation or depth.
>
> We would like to emphasize that we carefully phrased Definition 2.6 to highlight the connections between topological message passing (e.g., Bodnar et al.) and graph message passing schemes (e.g., Topping et al.). These connections are crucial to extending the theoretical understanding of oversquashing from graphs to higher-order domains. However, it is important to note that Topping et al. and similar works do not directly apply to these settings. For example, it is not immediately clear what would play the role of adjacency matrices for simplicial complexes—would it be a Hodge Laplacian or something else? What about hypergraphs or knowledge graphs? This ambiguity highlights the need to systematically identify appropriate analogs of key objects and quantities such as adjacency matrices, curvature, and other structural quantities when extending results to relational structures.
>
> In our work, these objects—such as the aggregated influence matrix—emerge naturally from deriving extensions of key results in the graph neural network literature. This process of teasing out appropriate analogs ensures that tools from GNNs can be meaningfully applied to higher-order settings. As a result, our work establishes a systematic guideline for importing concepts and theoretical tools from graph neural networks into relational message-passing frameworks, including topological deep learning.
>
> As far as we can tell, this is the first work to take explicit steps toward analyzing oversquashing in topological message passing. By providing one concrete formulation of how this analysis can be performed, we aim to not only bridge a critical gap in the literature but also offer a perspective that can guide future research in that area. Our framework provides clarity on how to approach this understudied area.
>
> > The substitution of Ri with Sni seems to have minimal impact on the overall definition, effectively diminishing the role of relations Ri within the definition.
>
> We kindly ask the reviewer to clarify. Having separate relations with possibly different arities is crucial for modeling message passing over various objects that fit the relational structure framework, ranging from simplicial complexes and hypergraphs to knowledge graphs and k-order graphs. Also, having relations with possibly different arities makes it possible to account for the different ways entities can exchange messages. For instance, in the CIN and CIN++ architectures, if a triplet $(\sigma, \tau, \delta)$ is related by upper adjacency, messages will pass from $\tau$ and $\delta$ to $\sigma$. If that does not address the reviewer’s question, kindly let us know, and we will update our response.
>
> > The shift operator ARi, associated with relation Ri, quantifies the signal passed from (ξ1,⋯,ξni−1) to σ when (σ,ξ1,⋯,ξni−1)∈Ri. However, it does not account for the signal passed from (σ,ξ2,⋯,ξni−1) to ξ1. Consequently, ARi may not serve as an adequate representation of Ri.
>
> We here emphasize the distinction between the relations in the relational structure as a computation graph modeling message passing over an object, and the relations of the underlying object itself. If the underlying object has a relation that would require messages to pass from $(\xi_1, \xi_2)$ to $\sigma$ and also from $(\sigma, \xi_2)$ to $\xi_1$, then those would be modeled by a relational structure that includes $(\sigma, \xi_1, \xi_2)$ and $(\xi_1, \sigma, \xi_2)$. This is evident in how we pass from a simplicial complex with boundary, coboundary, upper adjacency, and lower adjacency, to a relational structure with relations $R_1$, $R_2$, … We added a worked out example to the appendix to further clarify the construction of the relational structure. We also updated the description of the relational structure arising from simplicial message passing starting around Line 170, replacing the neighborhood description we used earlier with a direct definition of the relation.

---

> ### Author Response · Authors · 2024-11-21
> **Author Response (2/2)**
>
> > It is observed in line 167 that certain projections exist from R4(σ) to R2(σ) and from R2(σ) to R1(σ) (similarly, from R5(σ) to R3(σ) and from R3(σ) to R1(σ)). The relational structure could be further enriched by treating the set of relations as a directed set. Considering the direct limit of this directed set may provide a comprehensive summary of all the relations within the given relational structure. This could serve as a terminal object for defining message functions, potentially enhancing the model’s representational capacity.
>
> We greatly appreciate this suggestion, as the theoretical study of relational structures is of significant interest to us. Based on our understanding, the directed set of relations appears to have a unique maximum, $R_1$, and as a consequence, the directed limit of the set of relations would be isomorphic to $R_1$. That said, we recognize the possibility of having misunderstood the suggestion. If the reviewer could kindly elaborate or provide further clarification, we would be glad to address this point and respond accordingly.
>
> > Minor Comments:
> > Line 34: "Questions 2 and 9" should be changed to "research directions 2 and 9."
> > Definition 2.3: The symbol ≻ is used but not defined. This definition may be updated by replacing ≻ with ≺.
> > Definition 2.4: Is this definition necessary for the paper’s main arguments?
> > Definition 3.1: In the expression "Q∈A\~,B", it would be clearer to write "Q∈{A\~,B}."
>
> We thank the reviewer. We addressed these comments and absorbed Definition 2.4 into the following paragraph in the revision.
>
> We hope our responses and updates address the reviewer’s concerns and requests. We welcome any further comments or suggestions.

---

> > ### Comment · Reviewer_Qn33 · 2024-11-25
> >
> > I thank the authors for their explanation. While I acknowledge the effort and depth of your work, I remain unconvinced about the validity of Definition 2.5. Therefore, I keep my rating.

---

> ### Author Response · Authors · 2024-11-25
>
> We sincerely thank the reviewer for their response and for acknowledging our effort and the depth of our work.
>
> We understand that the reviewer has reservations regarding Definition 2.5. To better understand and address those reservations, could the reviewer kindly clarify what specific aspects of Definition 2.5 they find problematic? For example, are there logical inconsistencies in the definition?
>
> The goal of Definition 2.5 is to provide a unified framework that subsumes, for the sake of our analysis, many (seemingly) separate message-passing schemes, thereby making it possible to make new connections and take significant first steps toward analyzing these schemes together in one fell swoop. For example:
> 1. The first paragraph immediately after Definition 2.5 demonstrates how, for instance, simplicial message passing (Bodnar et al. 2021) fits the relational framework.
> 2. In Appendix F.5, we show how the definitions of standard relational graph neural networks, such as R-GCN (Schlichtkrull et al. 2018) and R-GIN (Xu et al. 2019), fit the framework.
> 3. In Appendix G, we briefly demonstrate how higher-order graph neural networks, namely k-GNN (Morris et al., 2019), also fit the framework.
> 4. Many other topological deep learning architectures [1] fit this framework as well.
>
> That said, we recognize that there is room for improvement. If the reviewer could provide some details on how the definition is invalid or actionable suggestions on how the definition could be improved, we will accordingly make revisions before the end of the discussion period and/or plan further changes for the final version.
>
> Once again, we thank the reviewer for their time and feedback.
>
> References
>
> [1] Papillon, M., Sanborn, S., Hajij, M., & Miolane, N. (2023). Architectures of Topological Deep Learning: A Survey of Message-Passing Topological Neural Networks. arXiv preprint arXiv:2304.10031.

---

> > ### Comment · Reviewer_Qn33 · 2024-11-25
> >
> > Thank you for your response. In this definition, for each relation $R_i$, there is a corresponding matrix, referred to as the shift operator. Could you please clarify the role of relation $R_i$ in determining its shift operator? I was unable to identify a clear logical connection between a relation $R_i$ and its shift operator. This is why I mentioned that $ A^{R_i} $ may not adequately serve as a representation. Ideally, the association should be precise, similar to how a graph is represented by its adjacency matrix.

---

> ### Author Response · Authors · 2024-11-25
>
> Thank you so much for the clarification!
>
> The relations $ R_i $ are **combinatorial**: they specify how entities are connected—essentially defining the existence and structure of connections between entities (e.g., whether there is a connection between entities $\sigma$ and $\tau$ in case of binary relations).
>
> The shift operators $ A^{R_i} $ are **numerical representations** of these relations: they assign weights or strengths to the connections defined by $ R_i $. The shift operators can take different forms, such as:
>
> - **Binary (0/1) tensors**: For instance, the entries are $ 1 $ if there is a connection (as defined by $ R_i $) and $ 0 $ otherwise. This is often seen in unweighted adjacency matrices in graphs.
> - **Weighted or normalized tensors**: The values in $ A^{R_i} $ can also represent different strengths or properties of the connection, for example, normalized by dividing by the indegree of an entity or the root of the product of degrees of connected entities.
>
> The choice of the shift operator of a relation is part of the design of a message passing scheme, and there are some guiding principles for making those choices, mostly having to do with numerical stability or similar.
>
> For example, in Appendix F5, we note the following two graph layers:
> - In the **Graph Isomorphism Network (GIN) layer**, the shift operator is unweighted—it is a $ 0/1 $ matrix, where each connection is treated equally.
> - In the **Graph Convolutional Network (GCN) layer**, the shift operator is **normalized** by a factor of $ 1 / \sqrt{d_\sigma d_\tau} $, where $ d_\sigma $ and $ d_\tau $ are the degrees of nodes $\sigma$ and $\tau$, respectively. This normalization helps to prevent numerical instability and overemphasizing nodes with high connectivity.
>
> In our work, we formulated the definition in full generality to allow for various types of shift operators, including unweighted, weighted, normalized, and even more sophisticated schemes, depending on the needs of the specific message-passing architecture. In Appendix H, we choose binary 0/1 tensors as an example, but perhaps we should rethink that particular choice of shift operators in the worked example.

---

> ### Author Response · Authors · 2024-11-25
>
> In response to the reviewer's comment, we will add a definition similar to the following to the revision of the manuscript (along with appropriate mathematical references):
>
> **Definition (Shift Operator)**: A real-valued tensor $A^{R_i} \in \mathbb{R}^{|\mathcal{S}|^{n_i}}$ with non-negative entries is called a shift operator for an $n_i$-ary relation $R_i \subseteq \mathcal{S}^{n_i}$ if it satisfies $A_{\zeta_1, \zeta_2, \dots, \zeta_{n_i}}^{R_i} = 0$ for any combination $(\zeta_1, \zeta_2, \dots, \zeta_{n_i}) \in \mathcal{S}^{n_i}$ where the relation $R_i$ does not hold among the entities $\zeta_1, \zeta_2, \dots, \zeta_{n_i}$ (i.e., $(\zeta_1, \zeta_2, \dots, \zeta_{n_i})
> \notin R_i$).
>
> We will also clarify this definition further with examples, emphasize the connection between a relation and its shift operator throughout the manuscript, and rewrite the worked example in Appendix H.
>
> **EDIT**: To further clarify, when we write $A_{(\sigma, \boldsymbol{\xi})}^{R_i}$ where $\boldsymbol{\xi} = (\xi_1, \dots, \xi_{n_i-1})$, we are abusing/overloading the notation. That is, $A_{\sigma, \boldsymbol{\xi}}^{R_i} = A_{\sigma, \xi_1, \dots, \xi_{n_i - 1}}^{R_i}$.

---

> > ### Comment · Reviewer_Qn33 · 2024-11-26
> >
> > This definition is relatively satisfactory and provides an acceptable representation of relations. The effects of this new definition should be considered within the context of the theory presented in the paper. I acknowledge your effort in addressing the comments carefully, and I will raise my score to 6.

---

### Official Review · Reviewer_MJXt · 2024-10-31

**Soundness:** 3
**Presentation:** 2
**Contribution:** 2
**Rating:** 6
**Confidence:** 4

**Summary:**

This paper provides both theoretical and experimental insights into oversquashing within Topological Deep Learning (TDL), a research direction recently highlighted as important for the TDL community in a recent position paper [1]. The authors encode topological structures, specifically simplicial complexes, as relational structures, allowing these to be transformed into weighted graphs via shift operators. Using this framework, they extend existing work on graph oversquashing and adapt popular rewiring algorithms to topological domains. The paper then presents experimental evidence supporting the theoretical findings and evaluates the combined performance of rewiring and TDL methods on TUD graph benchmarks.

[1] Theodore Papamarkou, Tolga Birdal, Michael M Bronstein, Gunnar E Carlsson, Justin Curry, Yue Gao, Mustafa Hajij, Roland Kwitt, Pietro Lio, Paolo Di Lorenzo, et al. Position: Topological deep learning is the new frontier for relational learning. In Forty-first International Conference on Machine Learning, 2024.

**Strengths:**

1. The paper successfully extends key results on graph oversquashing to topological domains, providing a comprehensive theoretical foundation for this adaptation.

2. The paper tackles questions that have been of interest to the TDL community, as highlighted in the recent position paper [1].

[1] Theodore Papamarkou, Tolga Birdal, Michael M Bronstein, Gunnar E Carlsson, Justin Curry, Yue Gao, Mustafa Hajij, Roland Kwitt, Pietro Lio, Paolo Di Lorenzo, et al. Position: Topological deep learning is the new frontier for relational learning. In Forty-first International Conference on Machine Learning, 2024.

**Weaknesses:**

1. The novelty of the paper is somewhat limited. It reduces simplicial complexes to an "influence graph" and then re-derives several previously established results on graph oversquashing. The observation that simplicial complexes can be encoded as graphs has been explored in prior work e.g. [1,2,4] (which i would quote if not quoted already), and many of the paper’s results on oversquashing closely mirror existing studies.

2.  An important question regarding oversquashing and TDL is understanding the effects of oversquashing in topological domains compared to standard graphs. Specifically, it remains unclear whether lifting graphs to higher-order domains could mitigate some effects of oversquashing. The paper does not address this question theoretically, and its empirical analysis only begins to explore this comparison. Instead, it primarily focuses on extending existing results on graph oversquashing to TDL, leaving a key question raised in the position paper [3] only partially addressed.

3. A minor critique is that the paper quickly reduces simplicial message passing to a relational setting, which is more combinatorial than topological. It would be interesting to see an analysis of oversquashing that leverages properties of the underlying topology of the simplicial complexes, such as topological curvature.

4. The experimental section has several issues: First, the real-world benchmarks are limited to TUD datasets, which are somewhat outdated, with a small number of samples and problematic evaluation. It would strengthen the results to include newer benchmarks, such as the Zinc and OGB datasets. These datasets were also used in the papers introducing  CIN and CIN++ which are baselines used in this paper. Additionally, since recent TDL models such as CIN and CIN++ use cyclic lifting rather than clique lifting, it would have been beneficial to include cyclic lifting across all experiments. Lastly, the effects of rewiring on real-world datasets appear inconclusive, making it difficult to determine when rewiring should be applied with TDL models.

[1] Yam Eitan, Yoav Gelberg, Guy Bar-Shalom, Fabrizio Frasca, Michael Bronstein, and Haggai Maron. Topological blind spots: Understanding and extending topological deep learning through the lens of expressivity. arXiv preprint arXiv:2408.05486, 2024.

[2] Mustafa Hajij, Ghada Zamzmi, Theodore Papamarkou, Nina Miolane, Aldo Guzm´an-S´aenz, Karthikeyan Natesan Ramamurthy, Tolga Birdal, Tamal K Dey, Soham Mukherjee, Shreyas N Samaga, et al. Topological deep learning: Going beyond graph data. arXiv preprint arXiv:2206.00606, 2022.

[3] Theodore Papamarkou, Tolga Birdal, Michael M Bronstein, Gunnar E Carlsson, Justin Curry, Yue Gao, Mustafa Hajij, Roland Kwitt, Pietro Lio, Paolo Di Lorenzo, et al. Position: Topological deep learning is the new frontier for relational learning. In Forty-first International Conference on Machine Learning, 2024.

[4] Mathilde Papillon, Guillermo Bern´ardez, Claudio Battiloro, and Nina Miolane. Topotune: A framework for generalized combinatorial complex neural networks. arXiv preprint arXiv:2410.06530, 2024.

**Questions:**

Is there significance in using multiple relations rather than a single one, other then for clarity? Could tighter bounds,  e.g. in equation (8)  be achieved by considering matrices  $B^{R_i}$ rather then a single $B$? Conversely, is it possible to combine all relations into a single “union relation”?

---

> ### Author Response · Authors · 2024-11-21
> **Author Response (1/2)**
>
> We thank the reviewer for their evaluation, and for the very valuable questions and suggestions that will improve the manuscript.
>
> > The novelty of the paper is somewhat limited. It reduces simplicial complexes to an "influence graph" and then re-derives several previously established results on graph oversquashing.
>
> We would like to clarify that, in addition to framing simplicial complexes as relational structures, our principled reduction of relational structures to influence graphs extends to a range of message passing schemes not restricted to topological message passing. Extending or generalizing previously established results on graph oversquashing in relational message passing becomes crucial for, e.g., identifying principled analogs of quantities such as curvature that are important for studying oversquashing in these previously unexplored setups. This approach exemplifies letting the problem guide the theory and lays a solid foundation for future work. In this sense, our philosophy aligns with the broader principles of geometric and topological deep learning, and as far as we can tell, this is the first work that takes steps towards studying oversquashing in these settings.
>
> > The observation that simplicial complexes can be encoded as graphs has been explored in prior work e.g. [1,2,4] (which i would quote if not quoted already), and many of the paper’s results on oversquashing closely mirror existing studies.
>
> We sincerely thank the reviewer for sharing those works. The perspectives of augmented Hasse diagrams and relational structures align very well, with the latter offering additional insights that extend connections to work from other machine learning communities. The works [1,2,4] reinforce our confidence that our approach is a step in the right direction. We have included citations to the suggested works after the definition of relational structures (Definition 2.4), in Remark 2.6 at the end of Section 2, and in Appendix A (Related Work).
>
> > An important question regarding oversquashing and TDL is understanding the effects of oversquashing in topological domains compared to standard graphs. Specifically, it remains unclear whether lifting graphs to higher-order domains could mitigate some effects of oversquashing. The paper does not address this question theoretically, and its empirical analysis only begins to explore this comparison. Instead, it primarily focuses on extending existing results on graph oversquashing to TDL, leaving a key question raised in the position paper [3] only partially addressed.
>
> We thank the reviewer for raising this important question. We acknowledge that our work is a principled first step toward studying oversquashing in TDL, laying the groundwork for future research. It shows that the structure of the computation graph, where messages are exchanged (and which we model with relational structures and relational message passing), is responsible for oversquashing and similar behaviors. This generalization allows graph-theoretic techniques to be extended to relational structures, as has been done in other works to study expressive power and blind spots in topological message passing.
>
> We demonstrated, through synthetic examples, that computation graphs and relational structures corresponding to graph lifts exhibit prototypical oversquashing trends with respect to local geometry, depth, and width, similarly to ordinary graphs. We also added an example showing that trees and trees with attached cycles exhibit nearly identical oversquashing behavior despite having different topologies. From the computation graph and relational structure perspective, this similarity becomes evident, as both have very similar computation graphs.
>
> > A minor critique is that the paper quickly reduces simplicial message passing to a relational setting, which is more combinatorial than topological. It would be interesting to see an analysis of oversquashing that leverages properties of the underlying topology of the simplicial complexes, such as topological curvature.
>
> We appreciate the reviewer’s critique regarding the reduction of simplicial message passing to a relational setting. This reduction is indeed a key part of our work, as it makes it possible to take concrete first steps towards fully describing oversquashing and similar phenomena in topological deep learning. Relational structures serve as a combinatorial model for computation graphs and, in the case of topological deep learning, are directly derived from the topology of the underlying spaces. Specifically, the relations within the relational structures and the edges in the influence graphs encode the topological information relevant to topological message passing. Consequently, the values of curvature on the influence graph are affected by the topology to the extent that topology affects message passing.
>
> (continued in comment 2/2)

---

> ### Author Response · Authors · 2024-11-21
> **Author Response (2/2)**
>
> (continued from comment 1/2)
>
> Deriving relevant general curvature notions (or similar quantities) is therefore a principled first step toward comprehensively studying oversquashing and similar phenomena in topological message passing and other message passing schemes that fit the relational framework. The next step is to study how the general quantities derived from our work depend on the topology in the specific context of topological message passing.
>
> > The experimental section has several issues: First, the real-world benchmarks are limited to TUD datasets, which are somewhat outdated, with a small number of samples and problematic evaluation. It would strengthen the results to include newer benchmarks, such as the Zinc and OGB datasets. These datasets were also used in the papers introducing CIN and CIN++ which are baselines used in this paper. Additionally, since recent TDL models such as CIN and CIN++ use cyclic lifting rather than clique lifting, it would have been beneficial to include cyclic lifting across all experiments. Lastly, the effects of rewiring on real-world datasets appear inconclusive, making it difficult to determine when rewiring should be applied with TDL models.
>
> We expanded the experiments with Appendix D5, where we added an experiment on ZINC and included cyclic lifting since ring structure is important in modeling molecules. We also added experiments for node classification in Appendix D6. Due to the limited duration of the discussion period and computational resources, the new experiments are restricted to a subset of the models and rewiring algorithm candidates. The empirical results will be fully expanded in the final version of the manuscript.
>
> We also note that:
> * In synthetic benchmarks, both graph and topological message passing demonstrate very similar (at times nearly identical) prototypical oversquashing behavior and trends. Both also respond conclusively and similarly to rewiring, validating the theoretical predictions.
> * In real-world benchmarks, graph and topological message passing also respond similarly to rewiring. However, performance improvements can be highly sensitive to training conditions and hyperparameter choices, as observed in prior works [1]. In that regard, rewiring is best viewed as a preprocessing step or heuristic to mitigate oversquashing, rather than a universally guaranteed performance booster.
>
> > Could tighter bounds, e.g. in equation (8) be achieved by considering matrices $B^{R_i}$ rather then a single $B$? Conversely, is it possible to combine all relations into a single “union relation”?
>
> From Eq. 6 and Eq. 7, we can write $\mathbf{B} = \lambda \mathbf{I} + \sum_{i=1}^k \tilde{\mathbf{A}}^{R_i}$, with $\lambda$ being the maximum row of $ \sum_{i=1}^k \tilde{\mathbf{A}}^{R_i}$. By looking at the proof in Appendix C1, it becomes clear one would also use $\mathbf{B}^\prime = \sum_{i=1}^k \mathbf{B}^{R_i}$, where $\mathbf{B}^{R_i} = \lambda_i \mathbf{I} + \tilde{\mathbf{A}}^{R_i}$ and $\lambda_i$ is the maximum row sum of $\tilde{\mathbf{A}}^{R_i}$. However, since the maximum of sums is at most the sum of maxima, we have $\lambda \leq \sum_{i=1}^k \lambda_i$, and $\mathbf{B}^\prime$ does not provide a tighter bound than $\mathbf{B}$.
>
> Each aggregated influence matrix $\tilde{\mathbf{A}}^{R_i}$ summarizes the connections between pairs of entities arising from the relation $R_i$, and as such, the augmented influence matrix $\mathbf{B} = \lambda \mathbf{I} + \sum_{i=1}^k \tilde{\mathbf{A}}^{R_i}$ is a summary of all the relations, in a sense acting as a weighted union of all the relations. To clarify this point, we added a worked example in Appendix H. There, Figure 12 illustrates how the influence graph collapses all relations into one.
>
> We hope our responses and updates address the reviewer’s concerns and requests. We welcome any further comments or suggestions.
>
> References
>
> [1] Tortorella, D., & Micheli, A. (2023). Is Rewiring Actually Helpful in Graph Neural Networks?. arXiv preprint arXiv:2305.19717.

---

> > ### Comment · Reviewer_MJXt · 2024-11-23
> > **Response**
> >
> > I thank the authors for their detailed and thoughtful response and address it below.
> >
> > 1.  I agree with the authors that the proposed reduction of relational structures to influence graphs is principled and represents a significant step toward studying oversquashing in these settings. However, I still find the paper's novelty somewhat limited (though I emphasize that the paper introduces some novel components). As mentioned earlier, the reduction of topological data to graphs has been explored in several prior works, and the theoretical analysis of oversquashing presented here closely parallels existing analyses conducted on graphs.
> >
> > 2. I appreciate the authors inclusion of the suggested papers!
> >
> > 3. I agree with the authors that their proposed method “…allows graph-theoretic techniques to be extended to relational structures, as has been done in other works...”. However, I feel that the impact of lifting graphs into higher-order relational domains on oversquashing could benefit from further exploration, and the justification for oversquashing mitigation in these instances might be more thoroughly developed. From a theoretical perspective, there are currently no results, either negative or positive, regarding the impact of graph lifting on oversquashing. Furthermore, while the authors present synthetic experiments indicating that certain relational structures suffer from oversquashing, I feel that these examples may not fully reflect the characteristics of "real-world graphs". In practical scenarios, the addition of multi-relational structures (e.g., incorporating cycles) often enhances connectivity and may naturally mitigate oversquashing effects without the need for the adaptation of graph rewiring methods.
> >
> > 4. I thank the reviewer for their insightful response.
> >
> > 5. I appreciate the authors efforts to extend their experimental section!
> >
> > 6. I thank the authors for their clarification.

---

### Official Review · Reviewer_Sfm7 · 2024-11-01

**Soundness:** 4
**Presentation:** 3
**Contribution:** 3
**Rating:** 8
**Confidence:** 4

**Summary:**

The authors make three main contributions.

First, they propose a general message-passing scheme for relational structures and show that it generalizes R-GCNs, simplicial neural nets, and higher-order GNNs. This interesting connection will hopefully prevent different communities with different terminologies from working in parallel on the same type of problems.

In Definition 2.6, you should make sure all variables are properly introduced. For instance, $p_t$ is the size of the feature representation of the \sigma s at time t. This size variable does not occur in the previous simplicial message-passing scheme. Making the terminology consistent or at least introducing each with a sentence will help readers to more easily understand the terminology.

This approach of reformulating higher-order structures as relational structures allows insights about over-squashing from graph neural networks (GNNs) to be translated to a broader class of relational message-passing systems. Introducing the aggregated influence matrix and influence graph as core tools makes it possible to study how factors like local geometry, network depth, and hidden dimension size impact message passing in these systems, much like in traditional GNNs.

Finally, the authors use the proposed framework and connection to generalize rewiring methods from prior work to the relational setting. Here, it would be interesting to understand better how computationally complex the rewiring techniques are regarding the number of relations, their arity, and elements in S. This would make the trade-off between these heuristics and the gains in the experiments more explicit.

The paper makes important theoretical contributions. Most notably, I appreciate the connection between relational GNNs and topological GNNs. Bridging one or more communities that are thinking about similar problems and making them aware of each other is important. The paper is generally well-written and easy to follow. As mentioned above, there is some minor room for improvement in the definition of the terminology. Some more explanations in natural language when defining terminology could make it a bit easier for the reader.
The main weaknesses of the paper are the “real-world” datasets used in the experiments and the missing discussion of the complexity–accuracy trade-off by omitting a discussion of the computational complexity of the rewiring techniques. This should be an easy fix for the latter by adding another section to the appendix and referring to it in the main paper. Regarding the former, I believe no graph ML method should only be evaluated with the TUD dataset. This provides very limited empirical evidence for a method. Several other datasets (e.g., those about atomistic structures such as QM9, MD17, the peptides datasets, and many more) could have been used. The method could have also been tested on node and link prediction problems, which is the main area of use for relational GNNs. All of these datasets and evaluation setups have shortcomings, of course, but a broader set of problems would make the empirical results much more convincing. This is a very strong theoretical paper with weak empirical evidence that the proposed method is helpful in practice and worth the overhead.

**Strengths:**

- Strong theoretical contribution, establishing connections between different classes of GNNs.
- Theoretical machinery for understanding over squashing and other phenomena in relational and topological GNNs
- New rewiring heuristics for relational and topological GNNs

**Weaknesses:**

- Missing discussion of the computational complexity of rewiring
- Weak experimental results due to very limited set of problems the approach was evaluated with

**Questions:**

- Why did you not run experiments on realistic datasets beyond TUD?
- Why did you not consider entity (node) and link prediction experiments, which are typically the main targets of relational GNNs

---

> ### Author Response · Authors · 2024-11-21
> **Author Response**
>
> We thank the reviewer for their positive evaluation and for the valuable questions and suggestions that will improve the manuscript.
>
> > In Definition 2.6, you should make sure all variables are properly introduced.
>
> We thank the reviewer for pointing this out. We edited the definition accordingly.
>
> > Here, it would be interesting to understand better how computationally complex the rewiring techniques are regarding the number of relations, their arity, and elements in S. This would make the trade-off between these heuristics and the gains in the experiments more explicit.
>
> We added a brief discussion of computational complexity to Appendix I at the reviewer’s request. This section shows the complexity for clique lifting as well as the resulting influence graph size.
>
> > The main weaknesses of the paper are the “real-world” datasets used in the experiments and the missing discussion of the complexity–accuracy trade-off by omitting a discussion of the computational complexity of the rewiring techniques. This should be an easy fix for the latter by adding another section to the appendix and referring to it in the main paper. Regarding the former, I believe no graph ML method should only be evaluated with the TUD dataset. This provides very limited empirical evidence for a method. Several other datasets (e.g., those about atomistic structures such as QM9, MD17, the peptides datasets, and many more) could have been used. The method could have also been tested on node and link prediction problems, which is the main area of use for relational GNNs.
>
> We thank the reviewer for these suggestions. We added some experiments to Appendix D, namely node prediction tasks in Appendix D6 and ZINC in Appendix D5. We also added the ring lift to the ZINC experiment in order to better show the impact of different lifting procedures on performance. These experiments were conducted on limited numbers of configurations and datasets due to the limited duration of the discussion period and availability of computational resources. In the final revision, we will expand all the experiments.
>
> We hope our responses and updates address the reviewer’s concerns and requests. We welcome any further comments or suggestions.

---

> > ### Comment · Reviewer_Sfm7 · 2024-11-25
> >
> > I read the responses of the authors. The experimental results are inconclusive on node classification and ZINC; sometimes, rewiring works, and sometimes, it doesn't.  The strength of the paper is in establishing the connection between various classes of methods and extending known theory to some new more complex classes. Hence, I keep my rating.

---

### Official Review · Reviewer_HYLT · 2024-11-04

**Soundness:** 4
**Presentation:** 4
**Contribution:** 3
**Rating:** 8
**Confidence:** 4

**Summary:**

The authors present a generalisation of theoretical results on oversquashing in graph learning to topological deep learning by formulating these within the framework of relational structures. This framework more generally presents an interesting approach for extending theoretical results from graphs to higher order structures.

**Strengths:**

- The paper is well written and easy to follow.
- Relational structures seem to provide a general unified framework that includes not only simplicial message passing but also other higher order structures.
- Propositions and theorem are well defined and proofs are mathematically sound.

**Weaknesses:**

- The experimental settings where the combination of lifting and rewiring leads to performance improvements seems to be limited.

**Questions:**

- Is there a correspondence between the message passing graph introduced  by Bodnar et al. and the aggregated influence matrix introduced in Eq.6?
- The rewiring algorithms considered in the paper are limited to the addition of edges however sometimes lifting graphs to clique complexes can introduce redundancies therefore could maybe rewiring algorithms that also prune edges be a better fit this context?
- The rewiring algorithms considered in the paper are limited to binary edges however in general rewiring algorithms that also include the modification of higher order edges might further improve performance. Do the results presented in the paper provide insights into how one could proceed to formulate such higher order rewiring algorithms?

---

> ### Author Response · Authors · 2024-11-21
> **Author Response (1/2)**
>
> We thank the reviewer for their evaluation, and for the very valuable questions and suggestions that will improve the manuscript.
>
> > The experimental settings where the combination of lifting and rewiring leads to performance improvements seems to be limited.
>
> We expanded the experiments with node classification experiments in Appendix D6, ZINC in Appendix D5, and simplex pruning in Appendix D8. Due to the limited duration of the discussion period and computational resources, the new experiments are carried out on a limited scope and without hyperparameter tuning. The empirical results will be fully expanded in the final version of the manuscript.
>
> We also note that:
> * In synthetic oversquashing benchmarks, both graph and topological message passing demonstrate very similar (at times nearly identical) prototypical oversquashing behavior and trends. Both also respond conclusively and similarly to rewiring, validating the theoretical predictions.
> * In real-world benchmarks, graph and topological message passing also respond similarly to rewiring. However, performance improvements can be highly sensitive to training conditions and hyperparameter choices, as observed in prior works [1]. In that regard, rewiring might best be viewed as a preprocessing step or heuristic to mitigate oversquashing, rather than a universally guaranteed performance booster.
>
> > Is there a correspondence between the message passing graph introduced by Bodnar et al. and the aggregated influence matrix introduced in Eq.6?
>
> Given any message passing scheme that fits the relational message passing framework, the scheme defines the relations and shift operators in the relational structure, which in turn determine the aggregated influence matrix for that scheme. This applies not only to the simplicial message passing scheme from Bodnar et al. but also to other frameworks such as relational graph neural networks and k-graph neural networks. That is, the framework is general, and one gets distinct relational structures and influence graphs for the message passing schemes that fit the framework. To clarify this correspondence, we provide a worked example in Appendix H.

---

> ### Author Response · Authors · 2024-11-21
> **Author Response (2/2)**
>
> > The rewiring algorithms considered in the paper are limited to the addition of edges however sometimes lifting graphs to clique complexes can introduce redundancies therefore could maybe rewiring algorithms that also prune edges be a better fit this context?
> > The rewiring algorithms considered in the paper are limited to binary edges however in general rewiring algorithms that also include the modification of higher order edges might further improve performance. Do the results presented in the paper provide insights into how one could proceed to formulate such higher order rewiring algorithms?
>
> Our work provides insight into pruning and formulating higher-order rewiring algorithms. For context, we distinguish between, e.g., a simplicial complex and the corresponding computation graph, which we model as a relational structure. Since message passing takes place in the relational structure, it is the connections of the relational structure that are responsible for oversquashing. In this work, we identify sources of oversquashing in the relational structure and mitigate them by introducing new connections between entities. If necessary, one could also directly remove connections between entities in the structure. This presents a very promising direction and we provide an experiment on MUTAG showing that edge pruning can improve accuracy in Appendix D8.
>
> Another possible alternative would be identifying connections to add or remove from the relational structure by first translating the problem to the underlying simplicial complex. This requires adding or removing simplices in the simplicial complex to introduce or eliminate the desired connections in the relational structure. The resulting changes are then reflected in the relational structure by adding or removing the corresponding entities and connections. For instance, if a higher-order rewiring algorithm required adding or removing a higher-order edge modeled by a simplex from a simplicial complex, one would update the relational structure by adding or removing the corresponding entity, along with all related entities and connections necessitated by this change.
>
> Motivated by the reviewer’s question, we report new rewiring experiments in Appendix D8, where we perform pruning according to the first alternative we suggest, and observe promising improvements. We used model- and dataset-agnostic choices for the number of added and removed connections due to the limited duration of the discussion period and limited computational resources. As finetuning these rewiring parameters is standard practice in the literature, we will update the results with finetuned parameters in the final version.
>
> We hope our responses and updates address the reviewer’s concerns and requests. We welcome any further comments or suggestions.
>
> References
>
> [1] Tortorella, D., & Micheli, A. (2023). Is Rewiring Actually Helpful in Graph Neural Networks?. arXiv preprint arXiv:2305.19717.

---

> > ### Comment · Reviewer_HYLT · 2024-11-27
> >
> > I would like to thank the authors for their extensive reply and clarifications. The modifications have significantly improved the manuscript and I have updated my rating accordingly.

---

> ### Author Response · Authors · 2024-11-26
>
> We would like to follow up and kindly ask if our response and manuscript revisions have addressed the reviewer's questions. We would be very grateful to hear if there are any remaining concerns or necessary edits before the final manuscript revision deadline on Wednesday. Thank you! - The Authors

---

### Author Response · Authors · 2024-11-21
**Common Response (Revision 1)**

We thank all the reviewers for their evaluations and constructive feedback. We summarize the main updates to the manuscript below:

**Appendix D4: NeighborsMatch on trees and trees with attached cycles**

We benchmark trees and trees with attached cycles on NeighborsMatch to analyze the impact of topology on a prototypical example of oversquashing.

**Appendix D5: Real-World Benchmark - graph classification on ZINC (12K)**

We evaluate the impact of adding 40 edges to the graphs of the ZINC dataset (12K) and impact of ring lift on graph classification performance.

**Appendix D6: Real-World Benchmark - node classification on Cornell, Texas, Wisconsin, CiteSeer, and Cora**

We independently evaluate the impact of adding 40 edges to and pruning 40 edges from the graphs of real world dataset on on node classification performance.

**Appendix D7: Ablation test on the impact of network width and depth, and number of rewiring iterations**

We perform ablation tests to assess the influence of network width, depth, and the number of rewiring iterations on oversquashing and model performance.

**Appendix D8: Simplex pruning**

We describe how to incorporate edge pruning into our framework and report experiments on the impact of pruning edges from the MUTAG dataset on graph classification performance.

**Appendix H: Worked out example - relational structure and influence graph for a path graph**

We provide a detailed worked example illustrating the construction of the relational structure and influence graph for a simple path graph to clarify the definitions introduced in the manuscript.

**Appendix I: Computational complexity analysis**

We add a computational complexity analysis to explicitly discuss the trade-offs of the proposed methods in terms of relations, arity, and the size of the underlying structures.

Due to the limited duration of the discussion period and available computational resources, the new experiments were carried out on a limited scope and without tuning the rewiring hyperparameters. In the final version, we will expand the experimental settings and descriptions of the new experiments, and tune the rewiring (incl. pruning) hyperparameters.

All modifications or updates in the manuscript taking care of the reviewer suggestions are highlighted in purple.

Kindly let us know if you have any further questions or suggestions. We welcome and value your comments and feedback!

---

> ### Author Response · Authors · 2024-11-28
> **Revision 2**
>
> **Description**:
>
> This revision implements feedback from reviewers Qn33 and SPF6 by clarifying a definition, re-emphasizing an existing reference in the original submission as the source of the simplicial message passing scheme we review, adding some contextual comments, and including additional references to recent works on graph rewiring. These updates do **not** affect the theoretical or empirical results, which remain **unchanged** from the previous revision. **We also reaffirm our commitment to implementing all the additional changes and expanded experiments previously promised for the final revision.** All the updates in this revision are marked in **blue** for clarity.
>
> **Summary of Changes**:
> * **Section 2 - Simplicial Complexes Are Relational Structures**:
>   * Added a paragraph before Equations 1 and 2 to emphasize that the Bodnar et al. (2021b) paper cited in the original submission before these equations is the source of the equations, and cross-referenced relevant parts from the appendix (Appendix A - Related Work and Appendix F5 - Layers) to provide additional context. **(SPF6)**
>   * Clarified the relationship between relations and their shift operators in Definition 2.5, introduced a new remark (Remark 2.6), and included three references to prior works on graph shift operators to provide further context. **(Qn33)**
>
> * **Section 6 - Discussion and Conclusion**:
>   * Cross-referenced newly added references in Appendix A - Related Work as candidates for future empirical evaluations of our framework’s heuristic for extending graph rewiring to higher-order settings. **(SPF6)**
>
> * **Appendix A - Related Work**:
>   * Added references to recent graph rewiring works and suggested these as potential candidates for future research on higher-order rewiring. **(SPF6)**
> * **Appendix D.7 - Ablation Tests**:
>   * Added contextual comments on the ablation tests that were added in Revision 1. **(SPF6)**

---

### Author Response · Authors · 2024-11-25

Dear Reviewers,

As the author-reviewer discussion period is ending soon, we kindly ask if you could review our responses to your comments. If you have further questions or comments, we will do our best to address them before the discussion period ends. If our responses have resolved your concerns, we would greatly appreciate it if you could update your evaluation of our work accordingly.

Thank you once again for your valuable time and thoughtful feedback.

Best regards,

The Authors

---

### Public Comment · ~Diaaeldin_Taha1 · 2025-02-25
**Camera-Ready Revision**

Dear Reviewers and Chairs,

We sincerely appreciate your valuable feedback and careful review of our work. Since the rebuttal, we have expanded the experiments in Sections 5.1 and 5.2, as well as Appendices D4, D6, D7, D8, and E1. We have now uploaded the camera-ready version of the manuscript.

Looking forward to seeing you in Singapore!

The Authors

---

### Meta-Review · Area_Chair_zxLF · 2024-12-20

**Metareview:**

This paper unifies theoretical work on over-squashing in graph neural networks and more general topological message-passing algorithms. All reviews for this paper are positive,. The reviewers felt that the paper was well-presented overall and that the theoretical contributions were interesting and helped provide a good starting point for understanding over squashing in topological deep learning. The reviewers felt that the experimental results were limited, and the authors did address this concern by adding more experiments during the rebuttal. But ultimately the reviewers still did not feel that the strength of the paper lied in its empirical contributions.

**Additional Comments On Reviewer Discussion:**

The rebuttal discussion was useful and caused some reviewers to raise their scores after their questions/concerns were addressed. The reviewers added experiments during the rebuttal and said that they would expand these experiments (e.g., properly tune hyperparameters and expand settings experimented in) for the camera ready. We hope that they do so, as one of the major weaknesses pointed out by the reviewers pre-rebuttal was non-convinving experimental results.

---

### Decision · Program_Chairs · 2025-01-22

Accept (Poster)